# Riemannian MeanFlow

**Dongyeop Woo**[1]  **Marta Skreta**[2,3]  **Seonghyun Park**[1]  **Kirill Neklyudov**[2,3,4][*]  **Sungsoo Ahn**[1][*]

## Abstract

Diffusion and flow models have become the dominant paradigm for generative modeling on Riemannian manifolds, with successful applications in protein backbone generation and DNA sequence design. However, these methods require tens to hundreds of neural network evaluations at inference time, which can become a computational bottleneck in large-scale scientific sampling workflows. We introduce Riemannian MeanFlow (RMF), a framework for learning flow maps directly on manifolds, enabling high-quality generations with as few as one forward pass. We derive three equivalent characterizations of the manifold average velocity (Eulerian, Lagrangian, and semigroup identities), and analyze parameterizations and stabilization techniques to improve training on high-dimensional manifolds. In promoter DNA design and protein backbone generation settings, RMF achieves comparable sample quality to prior methods while requiring up to $10\times$ fewer function evaluations. Finally, we show that few-step flow maps enable efficient reward-guided design through reward look-ahead, where terminal states can be predicted from intermediate steps at minimal additional cost.

## 1. Introduction

Many scientific data types possess intrinsic geometric structure that is not faithfully captured by Euclidean representations. For example, protein backbones are naturally described as sequences of rigid-body transformations, where each residue frame encodes both position and orientation (Jumper et al., 2021; Watson et al., 2022; Yim et al., 2023a;b; Bose et al., 2023). DNA and RNA sequences are distributions over nucleotides constrained to the probability

---

[*]Equal advising [1]Korea Advanced Institute of Science and Technology (KAIST) [2]Mila - Quebec AI Institute [3]Université de Montréal [4]Institut Courtois. Correspondence to: Dongyeop Woo <dongyeop.woo@kaist.ac.kr>.

*Proceedings of the $43^{rd}$ International Conference on Machine Learning*, Seoul, South Korea. PMLR 306, 2026. Copyright 2026 by the author(s).

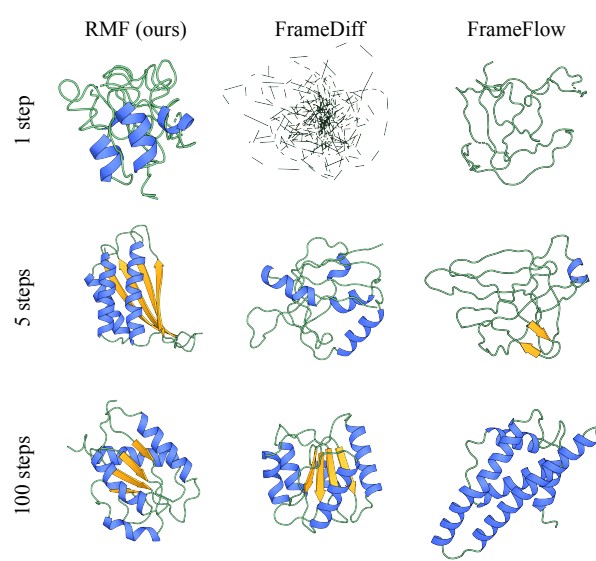

*Figure 1.* Protein backbone samples from RMF (ours), FrameDiff, and FrameFlow for different inference budgets. RMF produces well-formed structures in one step, while baselines require more.

simplex (Stark et al., 2024; Davis et al., 2024; Cheng et al., 2024), while molecular conformations are parameterized by torsion angles lying on circles (Jing et al., 2022). Naively embedding such data in $\mathbb{R}^d$ ignores these constraints, leading to invalid samples, e.g., unnormalized token probabilities or discontinuous angles, and inefficient learning.

Riemannian geometry provides a natural mathematical framework for modeling such data. By defining generative models directly on the appropriate manifold, such as $\mathrm{SE}(3)^N$ for protein backbones or the simplex $\Delta^{d-1}$ for sequences, geometric constraints are satisfied by construction. Building on the success of diffusion models and flow matching in Euclidean settings (Song et al., 2020; Lipman et al., 2022; Albergo et al., 2023), recent work has extended these continuous-time generative models to Riemannian manifolds, learning vector fields that transport noise to data along geodesic paths (Huang et al., 2022; De Bortoli et al., 2022; Chen & Lipman, 2023). These geometric generative models have achieved notable success in protein backbone generation (Yim et al., 2023a;b; Bose et al., 2023) and DNA sequence design (Davis et al., 2024; Cheng et al., 2024).

Despite this progress, the inference cost of manifold genera-

tive models remains a critical bottleneck. Sampling requires numerically integrating an ODE or SDE along the manifold, often demanding tens to hundreds of neural network evaluations (Song et al., 2020; Lipman et al., 2022). This computational burden is particularly problematic in scientific design pipelines, where generative models serve not as one-off samplers, but as a proposal mechanism within iterative loops, e.g., for property-guided optimization (Yang et al., 2020; Pacesa et al., 2024; Han et al., 2025). When each proposal requires hundreds of forward passes, the scope of exploration can become limited.

In Euclidean settings, this challenge has driven extensive research into *few-step* generation. Consistency models (Song et al., 2023; Song & Dhariwal, 2023) enforce self-consistency along trajectory paths, while flow map methods (Boffi et al., 2024; Geng et al., 2025; Zhou et al., 2025; Guo et al., 2025; Boffi et al., 2025) learn to transport between arbitrary time points via average-velocity regression. These approaches achieve high-quality generation with far fewer function evaluations, narrowing the gap with multi-step methods. However, these methods remain underexplored in the Riemannian manifold setting, limiting their applicability in scientific domains.

**Contributions.** To address this gap, we present Riemannian MeanFlow (RMF), a framework for few-step generation on Riemannian manifolds. Our contributions are:

1. **Riemannian MeanFlow identities and objectives:** We derive Riemannian generalizations of the Euclidean flow map identities (Boffi et al., 2025), obtaining theoretically equivalent characterizations (Eulerian, Lagrangian, and semigroup); we note that Davis et al. (2025) independently derived analogous characterizations. Crucially, our formulation allows for a different placement of the stop-gradient operator, which completely eliminates the need to evaluate higher-order derivatives in the training objectives. This distinction has significant implications for scalability: it reduces memory usage by nearly 2× and enables more stable training in high-dimensional settings (App. D).

2. **Scalable parameterization:** We propose $x_1$-prediction for manifold flow maps, where the network predicts a manifold-valued endpoint rather than a tangent vector. In our experiments, $x_1$-prediction performs comparably or better than $v$-prediction and scales well to high dimensions (up to $D = 2048$), making it compatible with existing scientific architectures that output manifold-valued points. We identify that the best combination for stable training on high-dimensional manifolds is the semigroup objective with $x_1$-prediction.

3. **Applications in scientific design:** Recent works have generalized consistency models (Cheng et al., 2025) and

flow-map learning (Davis et al., 2025) to manifolds, but validation remains limited to low-dimensional benchmarks ($\mathbb{S}^2$, $SO(3)$, torsion angles). We demonstrate that RMF can scale to real scientific tasks, showing that few-step generation is viable in high dimensions. On DNA promoter design (simplex in $\mathbb{R}^{1024 \times 4}$) and protein backbone generation ($SE(3)^N$ with $N$ up to 128), RMF matches the performance of the state-of-the-art multi-step generative models (Yim et al., 2023a; Davis et al., 2024) with up to $10\times$ fewer function evaluations. We further demonstrate reward-guided generation via reward look-ahead, which has not been previously shown on manifolds.

## 2. Riemannian MeanFlow Identities

### 2.1. Background on Riemannian Geometry

We consider a smooth, connected Riemannian manifold $\mathcal{M}$ and its Riemannian metric $g$. At each point $x \in \mathcal{M}$, the tangent space $T_x\mathcal{M}$ is a vector space of velocities with inner product $\langle \cdot, \cdot \rangle_g$ and norm $\| \cdot \|_g$. The disjoint union of all tangent spaces forms a tangent bundle $T\mathcal{M} := \bigsqcup_{x \in \mathcal{M}} T_x\mathcal{M}$. For manifolds embedded in ambient $d$-dimensional Euclidean space $\mathcal{M} \subset \mathbb{R}^d$, we let $\mathrm{Proj}_x : \mathbb{R}^d \to T_x\mathcal{M}$ denote tangential projection of vectors from the ambient space onto the tangent space at $x$. We provide a self-contained tutorial on Riemannian geometry in App. A and summarize key concepts needed for our framework below.

**Exponential and logarithmic maps.** The exponential map $\exp_x : T_x\mathcal{M} \to \mathcal{M}$ takes a tangent vector $v$ and returns the endpoint of the unit-time geodesic (locally shortest path) starting at $x$ with initial velocity $v$. Its local inverse, the logarithmic map $\log_x : \mathcal{M} \to T_x\mathcal{M}$, returns the initial velocity needed to reach a target point.

**Covariant derivatives.** To realize differentiation of vector fields on the manifold, we use the Levi-Civita connection $\nabla$. For a vector field $V(t)$ along a curve $\gamma(t)$, the covariant derivative $D_t V \coloneqq \nabla_{\dot{\gamma}} V$ measures how the vector field $V$ changes along the curve $\gamma$ while accounting for the manifold's curvature. For a manifold embedded in Euclidean space, this corresponds to projecting the standard Euclidean derivative onto the tangent space: $D_t V = \mathrm{Proj}_{\gamma(t)}(\frac{d}{dt}V)$.

**Derivatives of the logarithmic map.** Since $\log : \mathcal{M} \times \mathcal{M} \to T\mathcal{M}$ is a function of two arguments, we distinguish its partial derivatives: $\nabla^1_v \log_x(y)$ denotes the covariant derivative with respect to the first argument $x$ in direction $v$, while $d(\log_x)_y[w]$ denotes the differential with respect to the second argument $y$ in direction $w$. These quantities appear in our training objectives and have closed-form expressions for manifolds of interest (spheres, simplices, $SE(3)$); see App. A for details.

## 2.2. Flow Maps and Average Velocity on Manifolds

We now define the central objects of our framework. Our goal is to learn a generative model that transports a prior distribution $p_0$ to a data distribution $p_1$. This transport is governed by the ODE $\frac{dx_t}{dt} = v_t(x_t)$, where $v_t : \mathcal{M} \to T\mathcal{M}$ is a time-dependent vector field. We assume a chosen interpolant determines a family of intermediate marginals $\{p_t\}_{t \in [0,1]}$, where $p_t$ denotes the distribution at time $t$. In this setting, rather than learning $v_t$ and integrating it at inference time (as in flow matching), we directly learn the *flow map* that transports from one time point to another.

**Definition 2.1** (Integral curve). Given a time-dependent vector field $v : [0,1] \times \mathcal{M} \to T\mathcal{M}$, an integral curve is a smooth path $x : [0,1] \to \mathcal{M}$ satisfying $\frac{d}{dt} x_t = v_t(x_t)$. We use the notation $x_s, x_t, x_r$ to denote points on the same integral curve at times $s, t, r$ respectively.

**Definition 2.2** (Flow map). The flow map $\Phi_{s,t} : \mathcal{M} \to \mathcal{M}$ of a vector field $v_t$ is the mapping that transports points along integral curves: $\Phi_{s,t}(x_s) = x_t$ for any integral curve $(x_t)_{t \in [0,1]}$. The flow map satisfies the semigroup property $\Phi_{r,t} \circ \Phi_{s,r} = \Phi_{s,t}$, which states that flowing from $s$ to $r$ and then from $r$ to $t$ agrees with the direct flow from $s$ to $t$.

In Euclidean space, the flow map can be parameterized through the *average velocity*: the constant velocity that, if maintained from time $s$ to $t$, would transport $x_s$ to the same final point $x_t$. On manifolds, constant-velocity motion corresponds to traveling along geodesics. The Euclidean difference $x_t - x_s$ generalizes to the logarithmic map $\log_{x_s} x_t$, leading to the following definition:

**Definition 2.3** (Average velocity). The average velocity $u_{s,t} : \mathcal{M} \to T\mathcal{M}$ for a vector field $v_t$ is defined as

$$u_{s,t}(x_s) = \begin{cases} \dfrac{1}{t-s} \log_{x_s} x_t, & t \neq s, \\ v_s(x_s), & t = s, \end{cases} \quad (1)$$

for any integral curve $(x_t)_{t \in [0,1]}$ and times $s, t \in [0,1]$.

Geometrically, $u_{s,t}(x_s)$ is the constant velocity that would transport $x_s$ to $x_t$ over time of $t - s$ along a geodesic. The flow map can be recovered via the exponential map:

$$\Phi_{s,t}(x) = \exp_x\big((t-s)\, u_{s,t}(x)\big), \quad \forall\, s, t \in [0,1]. \quad (2)$$

### 2.3. Riemannian MeanFlow Identities

We present three equivalent characterizations of the average velocity. Each identity provides a necessary and sufficient condition: any vector field satisfying the identity must be the true average velocity. These identities form the basis of our training objectives in Sec. 3.1. The first two identities are obtained from differentiating the defining relation:

$$(t-s)\, u_{s,t}(x_s) = \log_{x_s} x_t \quad (3)$$

with respect to either $s$ or $t$. Following conventions in fluid mechanics, we call differentiation with respect to the source time $s$ the *Eulerian* perspective, and differentiation with respect to the target time $t$ the *Lagrangian* perspective.

**Proposition 2.1** (Eulerian RMF). *A vector field $u_{s,t} : \mathcal{M} \to T\mathcal{M}$ is the average velocity associated with $v_t$ **if and only if** it satisfies*

$$u_{s,t}(x_s) = (t-s)\, D_s u_{s,t}(x_s) - \nabla^1_{v_s} \log_{x_s} x_t, \quad (4)$$

*for any integral curve $(x_t)_{t \in [0,1]}$ and any $s, t \in [0,1]$.*

*Proof sketch.* Differentiating Eq. (3) with respect to $s$ gives

$$-u_{s,t}(x_s) + (t-s)\, D_s u_{s,t}(x_s) = D_s(\log_{x_s} x_t), \quad (5)$$

where covariant derivatives appear due to differentiating vector fields along the integral curve at $x_s$. The right-hand side equals $\nabla^1_{v_s} \log_{x_s} x_t$ by the covariant chain rule, where $v_s = \frac{d}{ds} x_s$ is the velocity along the curve. Rearranging yields Eq. (4). See App. B.1 for the complete proof. $\square$

**Proposition 2.2** (Lagrangian RMF). *A vector field $u_{s,t} : \mathcal{M} \to T\mathcal{M}$ is the average velocity associated with $v_t$ **if and only if** it satisfies*

$$u_{s,t}(x_s) = d(\log_{x_s})_{x_t}[v_t] - (t-s)\, \partial_t u_{s,t}(x_s), \quad (6)$$

*for any integral curve $(x_t)_{t \in [0,1]}$ and any $s, t \in [0,1]$.*

*Proof sketch.* Differentiating Eq. (3) with respect to $t$ gives

$$u_{s,t}(x_s) + (t-s)\, \partial_t u_{s,t}(x_s) = d(\log_{x_s})_{x_t}[v_t], \quad (7)$$

where the right-hand side is the differential of $\log_{x_s}$ at $x_t$ applied to $v_t = \frac{d}{dt} x_t$. Rearranging yields Eq. (6). For the complete proof, refer to App. B.1. $\square$

The third identity is algebraic rather than differential, following directly from the semigroup property $\Phi_{r,t} \circ \Phi_{s,r} = \Phi_{s,t}$. We provide the proof in App. B.1.

**Proposition 2.3** (Semigroup RMF). *A vector field $u_{s,t} : \mathcal{M} \to T\mathcal{M}$ is the average velocity associated with $v_t$ **if and only if** the following two conditions hold:*

*(i) Boundary condition: $u_{s,s}(x) = v_s(x), \ \forall x \in \mathcal{M}$;*

*(ii) Semigroup consistency: for any $s \neq t$ and $r \in [s, t]$,*

$$u_{s,t}(x_s) = \frac{1}{t-s} \log_{x_s} \Phi_{r,t}\big(\Phi_{s,r}(x_s)\big), \quad (8)$$

*where $\Phi_{s,t}(x) := \exp_x\big((t-s)\, u_{s,t}(x)\big)$ is the flow map induced by $u_{s,t}$.*

# 3. Flow Map Learning with Riemannian MF

## 3.1. Training Objectives

To learn a flow map transporting a prior $p_0$ to data distribution $p_1$, we parameterize the average velocity via a neural network $u_{s,t}^\theta : \mathcal{M} \to T\mathcal{M}$. The induced flow map is then:

$$\Phi_{s,t}^\theta(x) := \exp_x\big((t-s)u_{s,t}^\theta(x)\big). \tag{9}$$

**From identities to objectives.** Each identity from Sec. 2.3 yields a training objective by converting the consistency condition into a regression target.

> **Proposition 3.1** (Riemannian MeanFlow objectives). *Let $u_{s,t}^\theta : \mathcal{M} \to T\mathcal{M}$ be a parameterized average velocity with induced flow map $\Phi_{s,t}^\theta$ as in Eq. (9). The following objectives are valid for learning the average velocity:*
>
> 1. **Eulerian RMF:** *Sample $x_s \sim p_s$ and $s,t \sim p(s,t)$. The objective is*
>
> $$\mathcal{L}_{\mathrm{EMF}}(\theta) = \mathbb{E}_{x_s,s,t}\Big[\big\| u_{s,t}^\theta(x_s) - \mathrm{sg}(\hat{u}_{\mathrm{tgt}}) \big\|_g^2\Big], \tag{10}$$
>
> *where $\hat{u}_{\mathrm{tgt}} = (t-s)\, D_s u_{s,t}^\theta(x_s) - \nabla_{v_s}^1 \log_{x_s} \Phi_{s,t}^\theta(x_s)$.*
>
> 2. **Lagrangian RMF:** *Sample $x_t \sim p_t$ and $s,t \sim p(s,t)$. Let $\hat{x}_s = \Phi_{t,s}^\theta(x_t)$. The objective is*
>
> $$\mathcal{L}_{\mathrm{LMF}}(\theta) = \mathbb{E}_{x_t,s,t}\Big[\big\| u_{s,t}^\theta(\hat{x}_s) - \mathrm{sg}(\hat{u}_{\mathrm{tgt}}) \big\|_g^2\Big] + \mathcal{L}_{\mathrm{cyc}}(\theta), \tag{11}$$
>
> *where $\hat{u}_{\mathrm{tgt}} = \mathrm{d}(\log_{\hat{x}_s})_{x_t}[v_t] - (t-s)\, \partial_t u_{s,t}^\theta(\hat{x}_s)$, and the cycle-consistency regularizer is defined as:*
>
> $$\mathcal{L}_{\mathrm{cyc}}(\theta) = \mathbb{E}_{x_t,s,t}\big[d_g(\Phi_{s,t}^\theta(\Phi_{t,s}^\theta(x_t)), x_t)^2\big], \tag{12}$$
>
> *where $d_g$ denotes the geodesic distance.*
>
> 3. **Semigroup RMF:** *Sample $x_s \sim p_s$ and times $s,r,t \sim p(s,r,t)$. The objective is*
>
> $$\mathcal{L}_{\mathrm{SMF}}(\theta) = \mathbb{E}_{x_s,s,r,t}\Big[\big\| u_{s,t}^\theta(x_s) - \mathrm{sg}(\hat{u}_{\mathrm{tgt}}) \big\|_g^2\Big], \tag{13}$$
>
> *where $\hat{u}_{\mathrm{tgt}} = \frac{1}{t-s}\log_{x_s} \Phi_{r,t}^\theta\big(\Phi_{s,r}^\theta(x_s)\big)$ for $t \neq s$, and $\hat{u}_{\mathrm{tgt}} = v_s$ for $t = s$.*
>
> *Here, $\mathrm{sg}(\cdot)$ denotes the stop-gradient operator.*

Full proofs are in App. B.2. The key step in deriving our objectives lies in converting the average velocity characterizations from Sec. 2.3 into self-consistent regression targets. Specifically, the Eulerian characterization requires evaluation at $x_t$, necessitating integration over the unknown $v_t$ starting from $x_s$. Instead, we replace $x_t$ with the current model prediction $\Phi_{s,t}^\theta(x_s)$. The Lagrangian case is analo-

gous, with $(x_s, x_t, \Phi_{s,t}^\theta(x_s))$ replaced by $(x_t, x_s, \Phi_{t,s}^\theta(x_t))$. For Lagrangian RMF, we add a cycle-consistency loss to encourage weak invertibility, as the regression input is model-predicted. Conversely, the semigroup identity is inherently self-consistent and directly yields a regression target.

**Stop-gradient and bias.** The stop-gradient operator $\mathrm{sg}(\cdot)$ treats the target as a constant during backpropagation, preventing gradients from flowing through $\hat{u}_{\mathrm{tgt}}$. This avoids expensive higher-order derivatives (e.g., gradients through Jacobian–vector products (JVPs)) and stabilizes optimization. Importantly, this is unbiased at convergence: when $u_{s,t}^\theta$ matches the true average velocity, the underlying identity is satisfied, and the gradient of the loss vanishes, regardless of whether gradients are propagated through the target.

**Approximating the marginal velocity.** In practice, the marginal velocity $v_s$ or $v_t$ is intractable. As in flow matching, we replace it with a tractable conditional velocity. Importantly, in all of our objectives, the velocity appears only through linear differential operators. As a result, taking the expectation over the conditioning variable commutes with these operators, so this replacement does not affect the objective in expectation. We show this proof in App. B.2.

**Computation of differential terms.** The covariant derivative $D_s u_{s,t}^\theta$ and partial derivative $\partial_t u_{s,t}^\theta$ in the differential objectives can be computed efficiently using forward-mode automatic differentiation via JVPs, which adds less than 20% overhead compared with a standard forward pass (Geng et al., 2025). For embedded manifolds, the covariant derivative can be obtained by computing the JVP in the ambient space and projecting the result onto the tangent space: $D_s u_{s,t}^\theta(x_s) = \mathrm{Proj}_{x_s}\big(\frac{d}{ds} u_{s,t}^\theta(x_s)\big)$. The differential quantities $\nabla^1 \log$ and $\mathrm{d}(\log_{x_s})$ admit closed-form expressions for manifolds of interest. In practice, we implement these operations using automatic differentiation, enabling efficient and flexible evaluation across different manifold choices.

**Objective-level distinction from GFM.** Generalised Flow Map (GFM) (Davis et al., 2025) independently derives flow-map identities on Riemannian manifolds and proposes objectives structurally similar to ours. We compare the Eulerian objectives as a representative example. GFM minimizes an Eulerian consistency residual written in terms of the learned flow map $\Phi_{s,t}^\theta$:

$$\mathbb{E}_{x_s,s,t}\left[\big\| \partial_s \Phi_{s,t}^\theta(x_s) + \mathrm{sg}\big(d\Phi_{s,t}^\theta[v_s^\theta(x_s)]\big) \big\|_g^2\right], \tag{14}$$

whereas RMF learns the average velocity $u_{s,t}^\theta$ via stopped-target regression:

$$\mathbb{E}_{x_s,s,t}\left[\big\| u_{s,t}^\theta(x_s) - \mathrm{sg}(\hat{u}_{\mathrm{tgt}}) \big\|_g^2\right]. \tag{15}$$

This difference in objective form also leads to a different stop-gradient placement. In GFM, $\partial_s \Phi_{s,t}^\theta(x_s)$ remains outside the stop-gradient, so backpropagation differentiates

through this derivative term and requires higher-order derivatives. In RMF, derivative-dependent quantities are inside the stopped target $\mathrm{sg}(\hat{u}_{\mathrm{tgt}})$; JVPs are computed to form the target but not differentiated through. This avoids higher-order derivatives by construction, reducing training memory by nearly $2\times$ and improving stability in our empirical comparison; see App. D.

## 3.2. Parameterization of the Flow Map

We consider three parameterizations for the average velocity $u_{s,t}^\theta$: prediction of $v$, $x_t$, or $x_1$, and identify $x_1$-prediction as the best-suited for manifold settings.

$v$**-prediction.** The most direct approach parametrizes the average velocity:

$$u_{s,t}^\theta(x_s) = \mathrm{Proj}_{x_s}\big(\mathtt{net}^\theta(x_s, s, t)\big) \in T_{x_s}\mathcal{M}, \qquad (16)$$

where $\mathrm{Proj}_{x_s}$ projects the network output onto the tangent space. The instantaneous velocity is recovered as $v_s^\theta(x_s) = u_{s,s}^\theta(x_s)$, and the flow map follows from Eq. (9). This parameterization is conceptually simple and commonly adopted in prior work on Euclidean flow maps.

$x_t$**-prediction.** An alternative is to directly model the flow map: $\Phi_{s,t}^\theta(x_s) = \mathtt{net}^\theta(x_s, s, t)$. The average velocity is then recovered via the logarithmic map. However, training requires enforcing the boundary condition $\Phi_{s,s}^\theta(x_s) = x_s$ and matching the instantaneous velocity $\frac{d}{dt}\Phi_{s,t}^\theta(x_s)\big|_{t=s} = v_s(x_s)$, which requires differentiating through the network at every step. This introduces significant computational overhead and instability.

$x_1$**-prediction.** We propose an $x_1$-prediction scheme that inherits the benefits of endpoint prediction from flow matching while accommodating the two-time structure of flow maps. The network predicts a point on the manifold, which we interpret as an estimate of the trajectory endpoint:

$$\hat{x}_1^\theta(x_s, s, t) = \mathtt{net}^\theta(x_s, s, t), \qquad (17)$$

$$u_{s,t}^\theta(x_s) = \frac{1}{1-s}\log_{x_s}\hat{x}_1^\theta(x_s, s, t). \qquad (18)$$

Unlike standard $x_1$-prediction in flow matching, the predicted endpoint depends on both times $s$ and $t$, since $u_{s,t}$ is a function of both time variables. A practical advantage of this parameterization is compatibility with existing architectures that output manifold-valued points. For protein backbones, models such as FrameDiff (Yim et al., 2023b), FrameFlow (Yim et al., 2023a) and FoldFlow (Bose et al., 2023) naturally predict $SE(3)$ frames. Thus, $x_1$-prediction can reuse such architectures without modification, which is convenient for repurposing pre-trained flow matching models as flow maps, whereas $v$-prediction would require modifying the output to produce tangent vectors.

**Numerical stability near** $s = 1$**.** The factor $(1-s)^{-1}$ in Eq. (18) diverges as $s \to 1$, which can destabilize training. To mitigate this issue, we reweight the per-sample error inside the norm using a factor $w(s)$:

$$w(s) = \frac{1-s}{\max(1-s, \epsilon)}, \qquad (19)$$

where we set $\epsilon \in \{0.05, 0.1\}$ in our experiments, following common practices in flow matching with $x_1$-prediction (Yim et al., 2023a; Bose et al., 2023; Li & He, 2025). In practice, this weighting stabilizes $x_1$-prediction in all tested settings.

## 3.3. Stabilizing Riemannian MF Training

We identify several sources of optimization difficulty and present practical stabilization techniques below. App. G.1 provides empirical support for these choices.

**Time sampling distribution.** We find that stable optimization requires different time-sampling schemes across the three objectives. For Eulerian MF, while the objective is agnostic to time ordering, we sample ordered time pairs with $s \le t$, covering half the unit square $(s, t) \in [0, 1]^2$. For Lagrangian MF, we sample time pairs in both directions, including both $s \le t$ and $t \le s$. Finally, semigroup MF also samples an intermediate time $r$ such that $s \le r \le t$.

**Adaptive loss weighting.** Differential objectives, particularly Eulerian MF, can suffer from high variance in derivative-dependent regression targets. To stabilize training, we adopt adaptive loss weighting:

$$\mathcal{L} = \mathrm{sg}(w)\,\|\Delta\|_g^2, \quad w = (\|\Delta\|_g^2 + c)^{-p}, \qquad (20)$$

with $p = 0.5$. This substantially improves sample quality while maintaining stability, consistent with prior observations (Song & Dhariwal, 2023; Geng et al., 2025).

**Time-derivative control.** We find that bounding the time derivative of the network output is crucial for differential objectives. Using low-frequency time embeddings (e.g., $\omega = 0.02$) significantly stabilizes training compared to high-frequency embeddings (e.g., $\omega = 30$), which is in-line with consistency-model literature (Lu & Song, 2024).

## 3.4. Reward-guided Inference with Flow Maps

Finally, we study reward-guided inference on manifolds to steer generations toward downstream objectives without model retraining (Skreta et al., 2025; Hasan et al., 2026). A common strategy is to use gradients of a differentiable reward to bias the generative dynamics during inference. Concretely, we directly incorporate this signal into each inference step by perturbing the learned flow map with a guidance vector $\zeta_t \in T_{x_t}\mathcal{M}$ and guidance scale $\lambda$:

$$x_{t+\Delta t} = \exp_{x_t}\big(\Delta t\,(u_{t, t+\Delta t}^\theta(x_t) + \lambda \cdot \zeta_t)\big). \qquad (21)$$

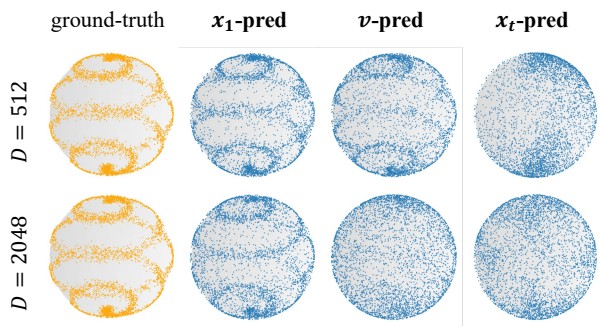

*Figure 2.* Parameterization choices: One-step generation results from models trained with different parameterizations ($x_1$-, $v$-, and $x_t$-pred) across ambient dimensions $D \in \{512, 2048\}$.

In practice, the performance of the guided generation significantly depends on the choice of $\zeta_t$. The naive approach is to define the guidance vector as the Riemannian gradient of the reward $\nabla_{x_t} r(x_t)$ evaluated at the current state $x_t$. However, evaluating the reward on $x_t$ is suboptimal, especially at small values of $t$, since the state is only partially denoised.

Sabour et al. (2025a), using their flow map model, proved that evaluating the reward via an $x_1$ look-ahead generates samples from the product density $\propto p_1(x) \exp(r(x))$. Similarly, we demonstrate that leveraging our flow map model in the manifold setting for reward evaluation is beneficial for guidance. In our setting, we use the guidance vector $\zeta_t = \nabla_{x_t} r(\Phi_{t,1}^\theta(x_t))$ as a simple heuristic.

## 4. Experiments

We first conduct an empirical analysis of key design choices to establish practical guidelines for flow map learning. Using these findings, we then evaluate RMF on two biological tasks: promoter DNA design and protein backbone generation. Finally, we show that we can do flow map-based reward-guided inference for both applications.

### 4.1. Ablation of Design Choices

We empirically study key design choices in flow map learning, focusing on parameterization and training objectives. Ablations for stabilization techniques are in App. G.1.

**Task description.** To emulate the manifold hypothesis, we use a synthetic 2D spherical helix embedded in a high-dimensional space. Points in the spherical helix $x \in \mathbb{S}^2 \subset \mathbb{R}^3$ are mapped to $y = Ux \in \mathbb{S}^{D-1} \subset \mathbb{R}^D$ via an unknown fixed column-orthogonal matrix $U \in \mathbb{R}^{D \times 3}$. For visualization, samples are projected back via $\tilde{x} = U^\top y \in \mathbb{R}^3$ and normalized as $x = \tilde{x}/\|\tilde{x}\|_2 \in \mathbb{S}^2$. Performance is evaluated across $D \in \{512, 2048\}$ using a 256-wide, 5-layer MLP.

**Parameterization choices.** We compare $x_1$-, $x_t$-, and $v$-prediction under the semigroup RMF objective. As shown in Fig. 2, $x_1$-prediction remains stable across

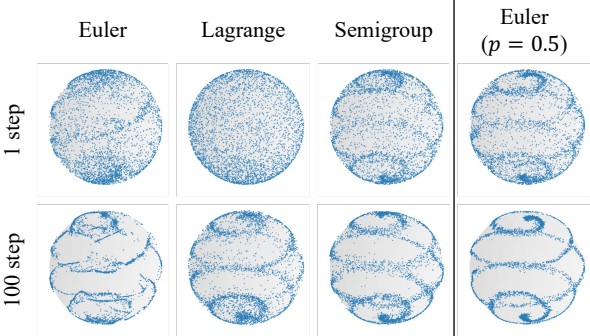

*Figure 3.* Objective choices: (Left) Samples generated by models trained with different objectives using 1-step (top) or 100-step (bottom) sampling on $D = 512$. (Right) Adaptive loss weighting for Eulerian RMF substantially improves sample quality.

both dimensions, while $x_t$-prediction completely fails. $v$-prediction degrades as $D$ increases; $x_1$-prediction performs well even at $D = 2048$, despite under-parameterization (i.e., network width $\ll D$).

**Objective choices.** We compare Eulerian, Lagrangian, and semigroup RMF objectives at ambient dimension $D = 512$ under the $v$-prediction. The semigroup objective yields the most consistent training behavior and the highest sample quality in our experiments (Fig. 3). While Eulerian alone is unstable and produces poor one-step samples, applying adaptive loss weighting substantially mitigates target variance and enables it to better capture the data distribution.

### 4.2. Promoter DNA Design

We evaluate RMF for generating human promoter sequences of length 1,024 conditioned on target transcription signal profiles. We train on 88,470 sequences from FANTOM5 (Hon et al., 2017) and compare against Dirichlet FM (Stark et al., 2024) and Fisher FM (Davis et al., 2024). We report (i) the mean squared error (MSE) between the signal profiles predicted by a pre-trained Sei model (Chen et al., 2022) for the generated and target human promoter sequences, and (ii) $k$-mer correlation ($k = 6$) between generated and real sequence distributions at different numbers of function evaluations (NFEs). A full task description and evaluation details are provided in Apps. E and F.

**Results.** Table 1 shows that all RMF variants using one-step generation match the 100-step performance of Fisher FM while consistently outperforming Dirichlet FM. Performance remains robust across NFEs (Fig. 4). Furthermore, $x_1$-prediction matches $v$-prediction across objectives; we provide a detailed analysis of its advantages in App. G.3.

**Reward guidance results.** We extend the DNA design setting to evaluate our reward guidance approach on the following task: given a target regulatory behavior, can we refine samples to better match it? At each inference step, we

*Table 1.* Promoter DNA sequence generation results on the test set averaged over three runs. Fisher FM is reproduced in our setup; the remaining baseline results follow Davis et al. (2024).

| Method | | NFE | MSE ($\downarrow$) | $k$-mer corr. ($\uparrow$) |
|---|---|---|---|---|
| Dirichlet FM | | 100 | $0.034_{\pm 0.004}$ | N/A |
| Fisher FM | | 100 | $0.030_{\pm 0.001}$ | $0.96_{\pm 0.01}$ |
| Eulerian RMF | $x_1$-pred | 1 | $0.030_{\pm 0.000}$ | $0.96_{\pm 0.01}$ |
| | $v$-pred | 1 | $0.031_{\pm 0.001}$ | $0.96_{\pm 0.00}$ |
| Lagrangian RMF | $x_1$-pred | 1 | $0.027_{\pm 0.001}$ | $0.88_{\pm 0.00}$ |
| | $v$-pred | 1 | $0.027_{\pm 0.001}$ | $0.85_{\pm 0.01}$ |
| Semigroup RMF | $x_1$-pred | 1 | $0.030_{\pm 0.001}$ | $0.84_{\pm 0.03}$ |
| | $v$-pred | 1 | $0.030_{\pm 0.001}$ | $0.93_{\pm 0.02}$ |

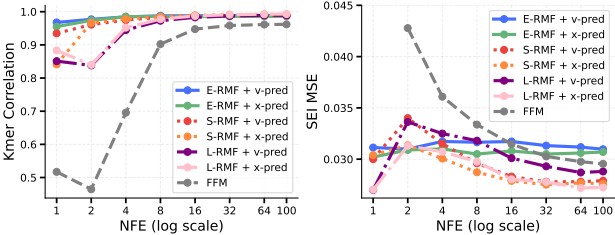

*Figure 4.* Performance vs. NFE. RMF variants outperform Fisher FM (FFM) in $k$-mer correlation ($k = 6$) and MSE. RMF achieves high accuracy at 1 NFE, whereas FFM requires $\geq 32$ steps for comparable performance.

compute the MSE between the Sei profiles of the $x_1$ look-ahead and the target sequence, and use its gradient to steer the samples according to Eq. (21). In Table 2, we report the MSE of the final sequence profiles to the test targets in Hon et al. (2017). Reward-guided sampling consistently reduces MSE compared to unguided generation, even for one-step sampling. Furthermore, naive guidance on the current state $x_t$ does worse than using the $x_1$ look-ahead. We show an ablation over guidance scales and reward details in App. E.2.

### 4.3. Protein Backbone Design

Finally, we evaluate our method on unconditional *de novo* protein backbone generation. We train on the SCOPe dataset (Chandonia et al., 2022) and benchmark three state-of-the-art baselines: GENIE (Lin & AlQuraishi, 2023), FrameDiff (Yim et al., 2023b), and FrameFlow (Yim et al., 2023a). We report standard metrics: *designability*, *novelty*, and *diversity*. For additional details, see Apps. E and F.

We find that using the semigroup RMF objective with $x_1$-prediction leads to the most stable training dynamics of the protein backbone generative model. We attribute this to the fact that the semigroup objective minimizes the number of differential operations required for the optimization. Furthermore, the $x_1$-prediction scalability agrees with the results in Sec. 4.1 and allows for out of the box usage of the standard protein backbone architectures used in prior work (Yim et al., 2023a;b; Bose et al., 2023).

*Table 2.* Reward-guided inference improves alignment between Sei signal profiles of generated and reference promoter sequences, measured by mean squared error $\pm$ standard deviation across 60 batches of 128 samples. Lower MSE is better. "—" is no guidance.

| NFE | — | $\nabla_{x_t} r(x_t)$ | $\nabla_{x_t} r(\Phi_{t,1}^\theta(x_t))$ |
|---|---|---|---|
| 1 | $0.033_{\pm 0.015}$ | $0.033_{\pm 0.015}$ | $0.025_{\pm 0.011}$ |
| 5 | $0.031_{\pm 0.014}$ | $0.017_{\pm 0.009}$ | $0.013_{\pm 0.005}$ |
| 10 | $0.031_{\pm 0.013}$ | $0.008_{\pm 0.002}$ | $0.008_{\pm 0.003}$ |

*Table 3.* Protein backbone generation results. Rows are grouped by inference regime (NFE). We highlight in **bold** the best designability ($<2$Å) within each regime, our primary metric. We mark not applicable (N/A) when no designable samples are generated or when metrics are not reported in prior work.

| Model | NFE | Designability | | Diversity | | Novelty |
|---|---|---|---|---|---|---|
| | | $<2$Å ($\uparrow$) | scRMSD ($\downarrow$) | Max. Cluster ($\uparrow$) | Pairwise scTM ($\downarrow$) | Max. scTM ($\downarrow$) |
| *Many-step regime (NFE $\geq 100$)* | | | | | | |
| | 1000 | 0.22 | N/A | 0.76 | N/A | 0.54 |
| GENIE | 750 | 0.11 | N/A | 0.79 | N/A | 0.51 |
| | 500 | 0.00 | N/A | N/A | N/A | N/A |
| FrameDiff | 500 | 0.80 | 1.63 | 0.36 | 0.34 | 0.68 |
| FrameDiff | 100 | 0.74 | 1.78 | 1.74 | 0.34 | 0.51 |
| FrameFlow | 100 | 0.93 | 1.16 | 0.41 | 0.30 | 0.77 |
| **RMF (Ours)** | 100 | **0.94** | 1.01 | 0.55 | 0.27 | 0.89 |
| *Moderate regime ($10 \leq$ NFE $< 100$)* | | | | | | |
| FrameDiff | 10 | 0.47 | 3.32 | 0.42 | 0.28 | 0.52 |
| FrameFlow | 10 | 0.61 | 2.34 | 0.54 | 0.26 | 0.67 |
| **RMF (Ours)** | 10 | **0.87** | 1.25 | 0.55 | 0.27 | 0.87 |
| *Few-step regime (NFE $< 10$)* | | | | | | |
| FrameFlow | 5 | 0.04 | 6.53 | 0.68 | 0.22 | 0.74 |
| FrameDiff | 5 | 0.09 | 6.19 | 0.54 | 0.24 | 0.96 |
| **RMF (Ours)** | 5 | **0.82** | 1.54 | 0.54 | 0.27 | 0.85 |
| FrameFlow | 2 | 0.00 | N/A | N/A | N/A | N/A |
| **RMF (Ours)** | 1 | **0.35** | 3.33 | 0.60 | 0.24 | 0.76 |

**Main results.** In Table 3, we report results across different numbers of function evaluations (NFE). Our method maintains high designability in the few-step regime: at 5 steps, it achieves $82\%$ designability (where designability is defined as the percentage of structures that have an RMSD $<2$Å when refolded using ESMFold (Lin et al., 2023)), while baselines drop sharply (e.g., $9\%$ for FrameDiff and $4\%$ for FrameFlow). Even in the extreme one-step setting, our method still generates $35\%$ designable samples, whereas baseline methods completely collapse. Since we optimize for designability, we observe a mild reduction in novelty, but maintain competitive diversity relative to baselines.

**Faster inference.** In Fig. 5a, we compare scRMSD against wall-clock time across NFEs. Notably, a RMF with small-sized model consistently outperforms prior methods across all step counts, achieving better designability than baselines at comparable cost. The RMF with larger model achieves the best designability in the low-step regime, although it has a larger per-step cost. More generally, increasing the model size significantly improves designability when using 1–10 sampling steps, while for 100 function evaluations, the gap

*Table 4.* Reward guidance increases the percentage of amino acids assigned to a target secondary structure composition. In each setting, 100 sequences of length 128 were generated using the RMF/S model. $\zeta_t$ set to "—" corresponds to no guidance.

| Reward | NFE | $\zeta_t$ | Mean ($\uparrow$) | Top-10 mean ($\uparrow$) | Max ($\uparrow$) | Frac. improved ($\uparrow$) |
|---|---|---|---|---|---|---|
| $\beta$-sheet | 5 | — | $0.18_{\pm 0.12}$ | $0.41_{\pm 0.06}$ | 0.51 | — |
| | | $\nabla r(x_t)$ | $0.18_{\pm 0.12}$ | $0.41_{\pm 0.04}$ | 0.48 | 0.28 |
| | | $\nabla r(\Phi_{t,1}^{\theta}(x_t))$ | $\mathbf{0.24}_{\pm 0.14}$ | $\mathbf{0.49}_{\pm 0.03}$ | **0.55** | **0.63** |
| | 10 | — | $0.20_{\pm 0.13}$ | $0.45_{\pm 0.04}$ | 0.52 | — |
| | | $\nabla r(x_t)$ | $0.20_{\pm 0.13}$ | $0.45_{\pm 0.04}$ | 0.52 | 0.37 |
| | | $\nabla r(\Phi_{t,1}^{\theta}(x_t))$ | $\mathbf{0.26}_{\pm 0.14}$ | $\mathbf{0.52}_{\pm 0.05}$ | **0.61** | **0.61** |
| $\alpha$-helix | 5 | — | $0.29_{\pm 0.20}$ | $0.68_{\pm 0.08}$ | 0.80 | — |
| | | $\nabla r(x_t)$ | $0.29_{\pm 0.20}$ | $0.68_{\pm 0.08}$ | 0.80 | 0.17 |
| | | $\nabla r(\Phi_{t,1}^{\theta}(x_t))$ | $\mathbf{0.39}_{\pm 0.22}$ | $\mathbf{0.76}_{\pm 0.04}$ | **0.83** | **0.75** |
| | 10 | — | $0.30_{\pm 0.20}$ | $0.70_{\pm 0.07}$ | 0.80 | — |
| | | $\nabla r(x_t)$ | $0.29_{\pm 0.20}$ | $0.70_{\pm 0.07}$ | 0.80 | 0.19 |
| | | $\nabla r(\Phi_{t,1}^{\theta}(x_t))$ | $\mathbf{0.45}_{\pm 0.23}$ | $\mathbf{0.81}_{\pm 0.04}$ | **0.86** | **0.75** |

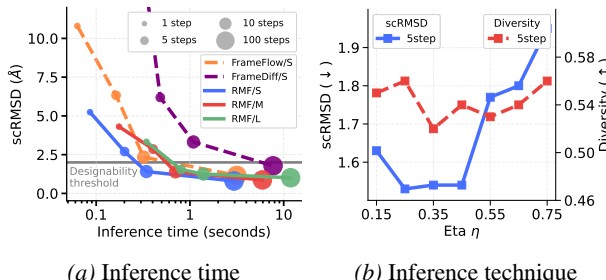

*(a)* Inference time       *(b)* Inference technique

*Figure 5.* **(a)** RMF consistently outperforms baselines in designability across inference steps. **(b)** Intermediate noise levels ($\eta \approx 0.25$–$0.45$) yield the best performance.

diminishes and all models perform on-par.

**Effect of inference techniques.** We apply a low-noise inference scheme at sampling time, a common technique in protein backbone generation (Yim et al., 2023a; Bose et al., 2023; Xie et al., 2025). We control the amount of injected noise using $\eta$, where $\eta = 1$ corresponds to full noise and $\eta = 0$ to no noise. In Fig. 5b, setting $\eta$ to intermediate values ($\eta \approx 0.25$–$0.45$) gave the best outcomes of low scRMSD and high diversity. We therefore used $\eta = 0.45$ in all following protein experiments. Additional details of the inference scheme are provided in App. F.2.

**Reward guidance results.** In protein design settings, it may be important to control the composition of structural motifs, as these motifs can influence protein function. Furthermore, generative models tend to oversample $\alpha$-helices (Faltings et al., 2025) and undersample other structural motifs (Lu et al., 2025). We consider secondary structure optimization as an illustrative task for controlling protein design (Hartman et al., 2025); we developed a differentiable secondary structure reward to guide towards higher compositions of $\beta$-sheets or $\alpha$-helices using $x_1$ look-ahead (see App. E.4), and evaluated final structures using DSSP (Kabsch & Sander, 1983). Table 4 shows that reward guidance using $\nabla_{x_t} r(\Phi_{t,1}^{\theta}(x_t))$ increases composition of both target secondary structures, while using $\nabla_{x_t} r(x_t)$ is similar to not doing guidance. A visual example is shown in Fig. 6a.

*Table 5.* Reward guidance can preserve target motifs in final generations of unconditional protein models. For each setting, 100 sequences of length 90 were generated using the RMF/S model with NFE = 50 and $\lambda = 1000$. $\zeta_t$ set to "—" corresponds to no guidance. Success rate is defined as the percentage of generations having motif RMSD < 1Å and backbone RMSD < 2Å, following (Zheng et al., 2024).

| Reward | $\zeta_t$ | Motif scRMSD ($\downarrow$) | Full scRMSD ($\downarrow$) | Success Rate ($\uparrow$) |
|---|---|---|---|---|
| 2KL8 | — | 3.07 | 12.88 | 0.00% |
| | $\nabla r(\Phi_{t,1}^{\theta}(x_t))$ | **1.52** | **7.26** | **3.00%** |

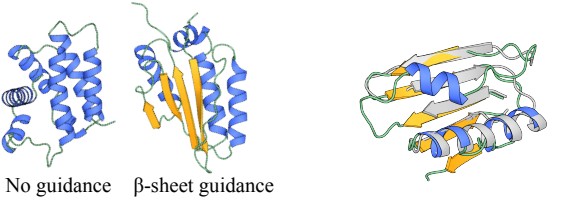

No guidance    β-sheet guidance

*(a)* Secondary structure      *(b)* Scaffold generation

*Figure 6.* **(a)** Example of protein generation from the same initial state using the base model (left) and reward-guided inference (right) toward higher $\beta$-sheet content with 10 NFE. $\beta$-sheets are highlighted in yellow, and $\alpha$-helices in blue. **(b)** Example of a generated protein scaffold that preserves the target motif (overlayed in grey) using reward guidance.

Finally, we apply our framework to motif scaffolding, where the goal is to generate protein backbones that incorporate a fixed target motif, a common requirement in functional protein design. Existing methods typically rely on explicit motif conditioning during training (Wang et al., 2022) or on importance sampling via sequential Monte Carlo applied to unconditional models (Trippe et al., 2023). We explore reward-guided inference as a test-time alternative with unconditional models. The reward is defined as the RMSD between the target motif and the corresponding residues of the generated structure at fixed indices, an approach similar to Yim et al. (2024). As a proof of concept, we apply this approach to the 2KL8 motif from Scaffold-Lab (Zheng et al., 2024) and show that $x_1$-based reward guidance yields successful motif-scaffolded generations (Table 5 and Fig. 6b).

## 5. Related Work

**Consistency models and flow-map learning.** Consistency models (CMs) reduce inference costs by mapping noise directly to data (Song et al., 2023; Song & Dhariwal, 2023; Lu & Song, 2024; Geng et al., 2024), but often lack explicit finite-time transport. Recent Euclidean flow map learning regresses *average velocities* for integration-free generation between arbitrary time points (Boffi et al., 2025; Geng et al., 2025; Guo et al., 2025; Zhou et al., 2025), which enhances multi-step robustness (Sabour et al., 2025b). However, these formulations assume flat geometry and do not naturally extend to Riemannian manifolds.

**Generative modeling on manifolds.** Manifold-aware

frameworks based on normalizing flows (Lou et al., 2020; Mathieu & Nickel, 2020), diffusion (Huang et al., 2022; De Bortoli et al., 2022), and flow matching (Chen & Lipman, 2023) have been shown to work in scientific domains such as biological sequence generation (Stark et al., 2024; Davis et al., 2024; Yim et al., 2023a;b; Bose et al., 2023). However, these methods often require computationally expensive numerical ODE/SDE integration during inference, which can be prohibitive in high dimensions.

**Few-step generation on manifolds.** Riemannian Consistency Models (Cheng et al., 2025) adapt CM objectives to manifolds but lack explicit flow map modeling. Most related to our work is Generalized Flow Map (GFM) (Davis et al., 2025), which appeared during the preparation of this manuscript and independently derives analogous flow-map identities on Riemannian manifolds. Despite this conceptual similarity, the two methods differ in a critical design choice: GFM applies stop-gradient to only some of the JVP-related terms, necessitating higher-order derivative evaluations and incurring greater memory cost and potential training instability in high-dimensional settings. RMF avoids these issues by construction. We illustrate this with the Eulerian objectives in Sec. 3, detail the theoretical connection to GFM in App. C.2, and provide empirical comparisons in App. D.

## 6. Conclusion

In this work, we introduce RMF, a principled framework for one- and few-step generative modeling on manifolds, informed by a systematic study of intrinsic manifold identities, parameterizations, and stabilization techniques. We find that aligning all these three components is important for scaling few-step generation to the high-dimensional geometries present in biological domains. Empirically, RMF matches the performance of strong multi-step baselines with up to $10\times$ fewer function evaluations, enabling efficient reward-guided exploration. Overall, our work establishes the theoretical foundation and practical guidelines for fast sampling on manifolds, which we envision will facilitate AI-aided discoveries in natural sciences.

## Impact Statement

This paper presents Riemannian MeanFlow, a principled framework for efficient generative modeling on manifolds. Our work contributes to the advancement of machine learning with specific applications in scientific domains such as protein backbone generation and DNA sequence design. While the advancement of these generative capabilities offers significant benefits for medicine and biotechnology, we recognize that the increased efficiency of biological design tools underscores the importance of maintaining rigorous ethical oversight and bio-security screening protocols. This work establishes a principled foundation for fast sampling on manifolds, which may lead to broader implications in scientific design pipelines where generative models serve as proposal mechanisms for property-guided optimization.

## Acknowledgements

We thank the authors of Davis et al. (2025) for valuable feedback that helped improve the presentation and comparison between our approaches.

This work was supported by the GRDC (Global Research Development Center) Cooperative Hub Program through the National Research Foundation of Korea (NRF) grant funded by the Ministry of Science and ICT (MSIT) (No. RS-2024-00436165); the Institute for Information & communications Technology Planning & Evaluation (IITP) grant funded by the Korea government (MSIT) (RS-2019-II190075, Artificial Intelligence Graduate School Program (KAIST)); the Advanced GPU Utilization Support Program (Beta Service) funded by the Government of the Republic of Korea (Ministry of Science and ICT); the "Advanced GPU Utilization Support Program" funded by the Government of the Republic of Korea (Ministry of Science and ICT); and the National Research Foundation of Korea (NRF) grant funded by the Ministry of Science and ICT (MSIT) (No. RS-2022-NR072184).

The research was also enabled in part by computational resources provided by the Digital Research Alliance of Canada (`https://alliancecan.ca`) and Mila (`https://mila.quebec`), and was partially sponsored by Google through the Google & Mila projects program. In addition, KN was supported by IVADO and Institut Courtois, and MS was supported by the IVADO 2025 Postdoctoral Research Funding Program. This project was undertaken thanks to funding from IVADO and the Canada First Research Excellence Fund.

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

# A. Tutorial on Riemannian Manifolds

This appendix provides a self-contained introduction to Riemannian geometry, covering the essential concepts needed to understand flow-based generative models on manifolds. We aim to build intuition while maintaining mathematical rigor.

## A.1. Smooth Manifolds

A **smooth manifold** $\mathcal{M}$ of dimension $d$ is a topological space that locally resembles Euclidean space $\mathbb{R}^d$. Formally, every point $x \in \mathcal{M}$ has a neighborhood that can be mapped smoothly and bijectively to an open subset of $\mathbb{R}^d$. These local maps are called *charts*, and a collection of charts covering $\mathcal{M}$ is called an *atlas*.

*Example* A.1 (The $d$-dimensional sphere). The unit sphere $\mathbb{S}^d = \{x \in \mathbb{R}^{d+1} : \|x\|_2 = 1\}$ is a $d$-dimensional manifold embedded in $\mathbb{R}^{d+1}$. Although it lives in $(d+1)$-dimensional space, it has only $d$ degrees of freedom.

*Example* A.2 (The probability simplex). The probability simplex $\Delta^{d-1} = \{p \in \mathbb{R}^d : p_i \geq 0, \sum_i p_i = 1\}$ represents discrete probability distributions over $d$ categories. This is the natural space for DNA sequences (with $d = 4$ for nucleotides A, C, G, T).

*Example* A.3 (Special Euclidean group SE(3)). The group $\mathrm{SE}(3) = \mathrm{SO}(3) \ltimes \mathbb{R}^3$ describes rigid body transformations in 3D space, consisting of rotations and translations. For protein backbone generation, we work on the product manifold $\mathrm{SE}(3)^N$ representing $N$ residue frames.

## A.2. Tangent Spaces and Tangent Bundles

At each point $x \in \mathcal{M}$, the **tangent space** $T_x\mathcal{M}$ is a vector space of the same dimension as $\mathcal{M}$. Intuitively, $T_x\mathcal{M}$ collects all possible infinitesimal directions in which one can move on the manifold starting from $x$.

**Definition A.1** (Tangent vector). A tangent vector $v \in T_x\mathcal{M}$ can be defined as the velocity of a smooth curve $\gamma : (-\epsilon, \epsilon) \to \mathcal{M}$ with $\gamma(0) = x$:

$$v = \dot{\gamma}(0) = \left. \frac{d\gamma(t)}{dt} \right|_{t=0}. \tag{22}$$

Under this definition, a tangent vector captures only the *first-order behavior* of a curve at the point $x$; different curves may represent the same tangent vector as long as they induce the same instantaneous change at $t = 0$.

**Tangent vectors and differentials.** Tangent vectors describe infinitesimal changes at a point. Differentials describe how such changes propagate through functions. Concretely, let $f : \mathcal{M} \to \mathcal{N}$ be a smooth map, and let $v \in T_x\mathcal{M}$ be represented by a curve $\gamma(t)$ with $\dot{\gamma}(0) = v$. Composing the curve with $f$ yields a new curve $f \circ \gamma(t)$ in $\mathcal{N}$, whose velocity at $t = 0$ defines the induced change in the output:

$$df_x(v) := \left. \frac{d}{dt} f(\gamma(t)) \right|_{t=0} \in T_{f(x)}\mathcal{N}. \tag{23}$$

Thus, a tangent vector specifies *what infinitesimal change is applied at the input*, while the differential specifies *how the function responds to that change*.

For manifolds embedded in $\mathbb{R}^D$, the tangent space can often be realized as a linear subspace of the ambient space. For example, for the sphere $\mathbb{S}^d \subset \mathbb{R}^{d+1}$, the tangent space at $x$ consists of all vectors orthogonal to $x$:

$$T_x\mathbb{S}^d = \{v \in \mathbb{R}^{d+1} : \langle v, x \rangle = 0\}. \tag{24}$$

The collection of all tangent spaces forms the **tangent bundle** as follows:

$$T\mathcal{M} = \bigsqcup_{x \in \mathcal{M}} T_x\mathcal{M}, \tag{25}$$

which itself has the structure of a smooth manifold of dimension $2d$. Points in $T\mathcal{M}$ are pairs $(x, v)$ consisting of a base point and a tangent vector attached to it.

**Definition A.2** (Projection onto tangent space). For embedded manifolds, ambient vectors in $\mathbb{R}^D$ can be mapped to the tangent space via orthogonal projection. The projection $\text{Proj}_x : \mathbb{R}^D \to T_x\mathcal{M}$ extracts the tangential component of a vector. For the sphere, it is given by

$$\text{Proj}_x(v) = v - \langle v, x \rangle x. \tag{26}$$

## A.3. Riemannian Metrics

A **Riemannian metric** $g$ assigns to each point $x \in \mathcal{M}$ an inner product $\langle \cdot, \cdot \rangle_x$ on the tangent space $T_x\mathcal{M}$, varying smoothly with $x$. This metric allows us to measure lengths, angles, and volumes on the manifold.

**Definition A.3** (Riemannian manifold). A Riemannian manifold $(\mathcal{M}, g)$ is a smooth manifold $\mathcal{M}$ equipped with a Riemannian metric $g$. The induced norm on $T_x\mathcal{M}$ is $\|v\|_x = \sqrt{\langle v, v \rangle_x}$.

*Example* A.4 (Euclidean metric). For $\mathbb{R}^d$, the standard Euclidean metric is $\langle u, v \rangle_x = u^\top v$, independent of $x$.

*Example* A.5 (Induced metric on spheres). For $\mathbb{S}^d \subset \mathbb{R}^{d+1}$, the induced metric is simply the restriction of the Euclidean inner product: $\langle u, v \rangle_x = u^\top v$ for $u, v \in T_x\mathbb{S}^d$.

*Example* A.6 (Fisher-Rao metric on the simplex). On the probability simplex $\Delta^{d-1}$, the Fisher-Rao metric is:

$$\langle u, v \rangle_p = \sum_{i=1}^{d} \frac{u_i v_i}{p_i}, \tag{27}$$

where $u, v \in T_p\Delta^{d-1}$ satisfy $\sum_i u_i = \sum_i v_i = 0$. This metric is natural for statistical manifolds and is used in DNA sequence modeling.

## A.4. Geodesics

A **geodesic** is a curve that locally minimizes distance on a Riemannian manifold—the generalization of straight lines to curved spaces.

**Definition A.4** (Geodesic). A smooth curve $\gamma : [0, 1] \to \mathcal{M}$ is a geodesic if it satisfies the geodesic equation:

$$\nabla_{\dot{\gamma}} \dot{\gamma} = 0, \tag{28}$$

where $\nabla$ is the Levi-Civita connection (defined below). Intuitively, this means the velocity vector undergoes parallel transport along the curve.

*Example* A.7 (Geodesics on the sphere). On the unit sphere $\mathbb{S}^d$, geodesics are great circles. The geodesic from $x$ to $y$ (assuming they are not antipodal) lies in the plane spanned by $x$, $y$, and the origin.

The **geodesic distance** $d_g(x, y)$ between two points is the length of the shortest geodesic connecting them:

$$d_g(x, y) = \inf_\gamma \int_0^1 \|\dot{\gamma}(t)\|_{\gamma(t)} \, dt, \tag{29}$$

where the infimum is over all smooth curves $\gamma$ with $\gamma(0) = x$ and $\gamma(1) = y$.

## A.5. Exponential and Logarithmic Maps

The exponential and logarithmic maps provide a way to move between the tangent space and the manifold, which is essential for defining interpolations and flow maps.

**Definition A.5** (Exponential map)**.** The exponential map $\exp_x : T_x\mathcal{M} \to \mathcal{M}$ maps a tangent vector $v$ to the endpoint of the geodesic starting at $x$ with initial velocity $v$, evaluated at time $t = 1$:

$$\exp_x(v) = \gamma(1), \quad \text{where } \gamma(0) = x,\ \dot\gamma(0) = v. \tag{30}$$

More generally, $\exp_x(tv)$ traces out the geodesic for $t \in [0, 1]$.

**Definition A.6** (Logarithmic map)**.** The logarithmic map $\log_x : \mathcal{M} \to T_x\mathcal{M}$ is the (local) inverse of the exponential map:

$$\log_x(y) = v \quad \text{such that} \quad \exp_x(v) = y. \tag{31}$$

The logarithmic map exists and is unique in a neighborhood of $x$ (within the *injectivity radius*).

**Proposition A.1** (Properties)**.** *The exponential and logarithmic maps satisfy:*

1. $\exp_x(\log_x(y)) = y$ *for y sufficiently close to* $x$

2. $\log_x(\exp_x(v)) = v$ *for* $\|v\|_x$ *sufficiently small*

3. $\|\log_x(y)\|_x = d_g(x, y)$ *(the norm of the log equals geodesic distance)*

4. $\exp_x(0) = x$ *and* $\log_x(x) = 0$

*Example* A.8 (Exponential and logarithmic maps on $\mathbb{S}^d$)**.** For the unit sphere, let $x \in \mathbb{S}^d$ and $v \in T_x\mathbb{S}^d$ with $\|v\| \neq 0$:

$$\exp_x(v) = \cos(\|v\|)x + \sin(\|v\|)\frac{v}{\|v\|}, \tag{32}$$

$$\log_x(y) = \frac{\theta}{\sin\theta}(y - \cos\theta \cdot x), \quad \theta = \arccos(\langle x, y\rangle). \tag{33}$$

When $v = 0$, we have $\exp_x(0) = x$. When $y = x$, we have $\log_x(x) = 0$.

### A.6. Geodesic Interpolation

Using the exponential and logarithmic maps, we can define geodesic interpolation between two points, which is fundamental for flow matching on manifolds.

**Definition A.7** (Geodesic interpolant)**.** Given $x_0, x_1 \in \mathcal{M}$, the geodesic interpolant at time $t \in [0, 1]$ is:

$$x_t = \exp_{x_0}(t \log_{x_0}(x_1)). \tag{34}$$

This traces out the geodesic from $x_0$ to $x_1$ with constant speed.

The velocity of this geodesic is constant along the path:

$$\dot{x}_t = \frac{d}{dt}x_t = P_{x_0 \to x_t}(\log_{x_0}(x_1)), \tag{35}$$

where $P_{x_0 \to x_t}$ denotes parallel transport (defined below).

### A.7. Covariant Derivatives and Connections

To rigorously formalize the geometric notions introduced above, we require the concept of a *connection*. In Euclidean space, geometric quantities such as velocity and acceleration along a curve are defined using ordinary derivatives: acceleration is simply the derivative of velocity. On a manifold, however, this notion breaks down. The velocity $\dot\gamma(t)$ of a curve $\gamma(t)$ lies in the tangent space $T_{\gamma(t)}\mathcal{M}$, which varies with $t$, so ordinary differentiation would attempt to compare vectors living in different tangent spaces.

A **connection** resolves this issue by prescribing a consistent way to differentiate vector fields on a manifold, thereby enabling meaningful notions of acceleration and variation of tangent vectors.

### A.7.1. VECTOR FIELDS AS DIFFERENTIAL OPERATORS

Before introducing connections, it is useful to recall a fundamental viewpoint from differential geometry: *vector fields act as differential operators on functions*. This operator perspective will be central to the definition of covariant derivatives.

**Definition A.8** (Directional derivative along a vector field). Let $X$ be a vector field on $\mathcal{M}$ and $f : \mathcal{M} \to \mathbb{R}$ a smooth function. The **directional derivative** of $f$ along $X$ is the function $Xf : \mathcal{M} \to \mathbb{R}$ defined by

$$(Xf)(x) = df_x(X(x)), \tag{36}$$

where $df_x : T_x\mathcal{M} \to \mathbb{R}$ denotes the differential of $f$ at $x$.

Equivalently, if $\gamma(t)$ is any curve satisfying $\gamma(0) = x$ and $\dot\gamma(0) = X(x)$, then

$$(Xf)(x) = \left.\frac{d}{dt}\right|_{t=0} f(\gamma(t)), \tag{37}$$

which represents the rate of change of $f$ in the direction $X(x)$.

**Proposition A.2** (Properties of directional derivatives). *For vector fields $X, Y$, functions $f, g$, and constant $c$:*

1. **Linearity in $X$:** $(X + Y)f = Xf + Yf$ *and* $(cX)f = c(Xf)$.

2. **Linearity in $f$:** $X(f + g) = Xf + Xg$ *and* $X(cf) = c(Xf)$.

3. **Leibniz rule:** $X(fg) = (Xf)g + f(Xg)$.

4. **Constants:** $X(c) = 0$ *for any constant function c.*

These properties show that $X$ acts as a *derivation* on the algebra of smooth functions. In fact, one may equivalently define vector fields as derivations satisfying these properties.

### A.7.2. AFFINE CONNECTIONS

Directional derivatives allow us to differentiate scalar functions, but do not provide a way to differentiate vector fields themselves. An affine connection fills this gap.

**Definition A.9** (Affine connection). An **affine connection** $\nabla$ on a smooth manifold $\mathcal{M}$ assigns to each pair of vector fields $X, Y$ a new vector field $\nabla_X Y$, called the *covariant derivative of $Y$ in the direction $X$*, such that for all vector fields $X, Y, Z$ and smooth functions $f, g$:

1. **Linearity in $X$:** $\nabla_{fX+gZ}Y = f\nabla_X Y + g\nabla_Z Y$.

2. **Linearity in $Y$:** $\nabla_X(Y + Z) = \nabla_X Y + \nabla_X Z$.

3. **Leibniz rule:** $\nabla_X(fY) = (Xf)Y + f\nabla_X Y$.

The Leibniz rule reflects the operator nature of $\nabla_X$: when differentiating $fY$, the derivative acts both on the scalar coefficient $f$ and on the vector field $Y$ itself.

Intuitively, $\nabla_X Y$ measures how the vector field $Y$ changes as one moves in the direction $X$, with the connection specifying how to compare vectors in neighboring tangent spaces.

### A.7.3. METRIC COMPATIBILITY

On a Riemannian manifold $(\mathcal{M}, g)$, it is natural to require the connection to interact consistently with the metric structure.

**Definition A.10** (Metric compatibility). A connection $\nabla$ is **metric-compatible** if for all vector fields $X, Y, Z$,

$$X\langle Y, Z\rangle = \langle \nabla_X Y, Z\rangle + \langle Y, \nabla_X Z\rangle. \tag{38}$$

This condition is a product rule for the Riemannian inner product, ensuring that differentiation commutes with taking inner products. It is a local compatibility requirement between the connection and the metric, and by itself does not uniquely determine the connection.

A.7.4. THE LEVI–CIVITA CONNECTION

Metric compatibility alone does not specify a unique connection. An additional natural requirement is the absence of torsion, which enforces symmetry of differentiation and generalizes the commutativity of partial derivatives in Euclidean space.

A classical result in Riemannian geometry shows that these two conditions together uniquely determine the connection.

**Theorem A.9** (Fundamental theorem of Riemannian geometry). *On any Riemannian manifold $(\mathcal{M}, g)$, there exists a unique affine connection $\nabla$, called the **Levi–Civita connection**, satisfying:*

1. ***Torsion-free:***
$$\nabla_X Y - \nabla_Y X = [X, Y],$$

2. ***Metric-compatible:***
$$X\langle Y, Z \rangle = \langle \nabla_X Y, Z \rangle + \langle Y, \nabla_X Z \rangle.$$

**Intuition.** The Levi–Civita connection is the canonical choice of differentiation that depends only on the Riemannian metric. It generalizes ordinary derivatives in Euclidean space and provides a consistent notion of differentiation intrinsic to the manifold geometry.

A.7.5. COVARIANT DERIVATIVE ALONG A CURVE

While an affine connection defines differentiation between vector fields, in dynamical and flow-based settings we primarily require differentiation *along a given trajectory* on the manifold.

**Definition A.11** (Covariant derivative along a curve). Let $\gamma : [0, 1] \to \mathcal{M}$ be a smooth curve and $V(t) \in T_{\gamma(t)}\mathcal{M}$ a vector field along $\gamma$. The **covariant derivative** of $V$ along $\gamma$ is defined as

$$D_t V := \nabla_{\dot{\gamma}(t)} V. \tag{39}$$

The operator $D_t$ provides an intrinsic notion of time differentiation for vector-valued quantities whose ambient space varies along the curve. In particular, $D_t V$ can be interpreted as the *acceleration* of $V(t)$ along $\gamma(t)$.

Importantly, $D_t V$ depends only on the values of $V$ along $\gamma$, and not on how $V$ is extended to a neighborhood of the curve.

For embedded submanifolds $\mathcal{M} \subset \mathbb{R}^D$, the covariant derivative admits a simple expression:

$$D_t V = \mathrm{Proj}_{\gamma(t)}\left( \frac{dV}{dt} \right), \tag{40}$$

where $\mathrm{Proj}_{\gamma(t)}$ denotes the orthogonal projection onto the tangent space $T_{\gamma(t)}\mathcal{M}$.

> *Example* A.10 (Covariant derivative on $\mathbb{S}^d$). For the unit sphere $\mathbb{S}^d \subset \mathbb{R}^{d+1}$, let $V(t)$ be a vector field along a curve $\gamma(t)$. Then
> $$D_t V = \frac{dV}{dt} - \left\langle \frac{dV}{dt}, \gamma(t) \right\rangle \gamma(t), \tag{41}$$
> which subtracts the normal component of the Euclidean derivative to ensure tangency.

**Remark.** In our framework, $D_t$ will serve as the intrinsic analogue of time derivatives appearing in flow-map dynamics, enabling consistent definitions of velocity and acceleration fields on curved spaces.

**A.8. Parallel Transport**

The covariant derivative allows us to interpret $D_t V$ as the *acceleration* of a vector field transported along a curve. A vector field satisfying $D_t V = 0$ is said to be *parallel* along $\gamma$; such fields change as little as possible while remaining tangent to the manifold. This notion of parallel transport provides a canonical way to compare tangent vectors at different points on $\mathcal{M}$ and will play a central role in defining consistent dynamics and flow-based constructions on curved spaces.

**Definition A.12** (Parallel vector field and parallel transport). Let $\gamma : [0, 1] \to \mathcal{M}$ be a smooth curve and let $V(t) \in T_{\gamma(t)}\mathcal{M}$ be a vector field along $\gamma$. We say that $V$ is *parallel* along $\gamma$ if it satisfies

$$D_t V(t) = 0 \qquad \text{for all } t \in [0, 1]. \tag{42}$$

Given an initial vector $v \in T_{\gamma(0)}\mathcal{M}$, there exists a unique parallel vector field $V(t)$ along $\gamma$ with $V(0) = v$. The *parallel transport* along $\gamma$ is the linear map

$$P_{\gamma,\, 0 \to t} : T_{\gamma(0)}\mathcal{M} \to T_{\gamma(t)}\mathcal{M}, \qquad P_{\gamma,\, 0 \to t}(v) := V(t), \tag{43}$$

where $V$ is the unique solution of $D_t V = 0$ with $V(0) = v$.

### A.9. Connection to Flow Map Learning

In our Eulerian MeanFlow objective, we encounter the covariant derivative $D_s u_{s,t}^{\theta}(x_s)$, where $u_{s,t}^{\theta}$ is the learned average velocity and $x_s$ moves along an integral curve. This derivative measures how the predicted average velocity changes as we move along the flow.

For embedded manifolds, this can be computed as:

$$D_s u_{s,t}^{\theta}(x_s) = \text{Proj}_{x_s} \left( \frac{d}{ds} u_{s,t}^{\theta}(x_s) \right), \tag{44}$$

where the total derivative $\frac{d}{ds}$ includes both explicit dependence on $s$ and implicit dependence through $x_s$:

$$\frac{d}{ds} u_{s,t}^{\theta}(x_s) = \partial_s u_{s,t}^{\theta}(x_s) + d u_{s,t}^{\theta}(x_s)[v_s(x_s)]. \tag{45}$$

In practice, this is computed efficiently using forward-mode automatic differentiation (Jacobian-vector products), followed by projection onto the tangent space.

### A.10. Differentials of the Exponential and Logarithmic Maps

For flow map learning, we need to differentiate through the exponential and logarithmic maps. Let $f : \mathcal{M} \to \mathcal{M}$ be a smooth map. The **differential** $df_x : T_x \mathcal{M} \to T_{f(x)} \mathcal{M}$ is defined by:

$$df_x(v) = \left. \frac{d}{dt} \right|_{t=0} f(\gamma(t)), \tag{46}$$

where $\gamma$ is any curve with $\gamma(0) = x$ and $\dot{\gamma}(0) = v$.

**Definition A.13** (Derivatives of the logarithmic map). For the logarithmic map $\log : \mathcal{M} \times \mathcal{M} \to T\mathcal{M}$, we denote:

- $\nabla_v^1 \log_x(y)$: derivative with respect to the first argument $x$ in direction $v$
- $d(\log_x)_y[w]$: derivative with respect to the second argument $y$ in direction $w$

These derivatives appear in our Eulerian and Lagrangian MeanFlow objectives. For many manifolds of interest (spheres, Lie groups, symmetric spaces), these derivatives have closed-form expressions.

### A.11. Product Manifolds

Many applications involve product manifolds $\mathcal{M} = \mathcal{M}_1 \times \mathcal{M}_2 \times \cdots \times \mathcal{M}_N$.

**Proposition A.3** (Geometry of product manifolds). *For a product manifold $\mathcal{M} = \prod_{i=1}^{N} \mathcal{M}_i$:*

- *Tangent space: $T_x \mathcal{M} = \prod_{i=1}^{N} T_{x_i} \mathcal{M}_i$*
- *Metric: $\langle u, v \rangle_x = \sum_{i=1}^{N} \langle u_i, v_i \rangle_{x_i}$*
- *Exponential map: $\exp_x(v) = (\exp_{x_1}(v_1), \ldots, \exp_{x_N}(v_N))$*
- *Logarithmic map: $\log_x(y) = (\log_{x_1}(y_1), \ldots, \log_{x_N}(y_N))$*

This decomposition is used for protein backbone generation on $SE(3)^N$, where each factor represents a residue frame.

# B. Derivation of Riemannian MeanFlow Identities and Objectives

## B.1. Riemannian MeanFlow Identities

In this section, we derive equivalent characterizations of the average velocity field on a Riemannian manifold. This appendix provides detailed proofs for Propositions Props. 2.1 to 2.3 stated in Sec. 2.3.

---

**Proposition B.1** (Riemannian MeanFlow identities). *A vector field $u_{s,t} : \mathcal{M} \to T\mathcal{M}$ is the average velocity associated with a time-dependent vector field $v_t$ if and only if one (and hence all) of the following conditions holds:*

1. *(**Eulerian condition**). For any integral curve $(x_t)_{t\in[0,1]}$ of $v_t$ and any $s, t \in [0, 1]$,*

$$u_{s,t}(x_s) = (t - s) \, D_s u_{s,t}(x_s) - \nabla^1_{v_s} \log_{x_s} x_t. \tag{47}$$

2. *(**Lagrangian condition**). For any integral curve $(x_t)_{t\in[0,1]}$ and any $s, t \in [0, 1]$,*

$$u_{s,t}(x_s) = \mathrm{d}(\log_{x_s})_{x_t}[v_t] - (t - s) \, \partial_t u_{s,t}(x_s). \tag{48}$$

3. *(**Semigroup condition**). For any $x_s \in \mathcal{M}$ and any $s, t \in [0, 1]$, we have $u_{s,s} = v_s$ and*

$$u_{s,t}(x_s) = \frac{1}{t - s} \log_{x_s}\big(\Phi_{r,t}\big(\Phi_{s,r}(x_s)\big)\big), \quad s \neq t, \tag{49}$$

*where $\Phi_{s,t}(x) := \exp_x\big((t - s)u_{s,t}(x)\big)$ denotes the flow map induced by $u_{s,t}$.*

---

*Proof.* Throughout the proof, we assume that $u_{s,t}$ is smooth. Under this assumption, any vector field satisfying the defining relation

$$(t - s) \, u_{s,t}(x_s) = \log_{x_s} x_t. \tag{50}$$

coincides with the average velocity.

For each identity, we first derive the identity from the defining relation of the average velocity, and then argue that each condition uniquely recovers the true average velocity field.

($\Rightarrow$) **Eulerian identity.** Both sides of Eq. (50) define vector fields along the curve $s \mapsto x_s$, so we apply the covariant derivative $D_s$:

$$-u_{s,t}(x_s) + (t - s) \, D_s u_{s,t}(x_s) = D_s(\log_{x_s} x_t). \tag{51}$$

Define the vector field $f : \mathcal{M} \to T\mathcal{M}$ by $f(x) := \log_x x_t$. By definition of the covariant derivative along a curve,

$$D_s(\log_{x_s} x_t) = \nabla_{\dot{x}_s} f(x_s) = \nabla_{v_s} f(x_s) =: \nabla^1_{v_s} \log_{x_s} x_t.$$

Rearranging terms yields Eq. (47).

($\Leftarrow$) **Converse.** For the proof of converse, assume that Eq. (47) holds along every integral curve. Fix an integral curve $(x_s)_s$ and define a vector field along it by

$$X(s) := (t - s)u_{s,t}(x_s) - \log_{x_s} x_t \in T_{x_s}\mathcal{M}.$$

Differentiating along the curve yields

$$D_s X(s) = -u_{s,t}(x_s) + (t - s)D_s u_{s,t}(x_s) - D_s(\log_{x_s} x_t) = 0,$$

where the last equality follows from Eq. (47). Hence $X$ is parallel along $s \mapsto x_s$. Since

$$X(t) = (t - t)u_{t,t}(x_t) - \log_{x_t} x_t = 0,$$

uniqueness of solutions to the parallel transport equation $D_s X = 0$ implies $X(s) \equiv 0$ for all $s$. Therefore,

$$(t - s)u_{s,t}(x_s) = \log_{x_s} x_t,$$

which is exactly the defining relation Eq. (50).

($\Rightarrow$) **Lagrangian identity.** Differentiating Eq. (50) with respect to $t$, both sides lie in the fixed vector space $T_{x_s}\mathcal{M}$, so ordinary differentiation applies:

$$u_{s,t}(x_s) + (t - s)\,\partial_t u_{s,t}(x_s) = \frac{\mathrm{d}}{\mathrm{d}t}\big(\log_{x_s} x_t\big). \tag{52}$$

By the chain rule,

$$\frac{\mathrm{d}}{\mathrm{d}t}\big(\log_{x_s} x_t\big) = \mathrm{d}(\log_{x_s})_{x_t}[v_t],$$

which gives Eq. (48).

($\Leftarrow$) **Converse.** Assume that Eq. (48) holds along every integral curve. Fix $s \in [0, 1]$ and an integral curve $(x_t)_t$. Define a curve in the fixed tangent space $T_{x_s}\mathcal{M}$ by

$$X(t) := (t - s)u_{s,t}(x_s) - \log_{x_s} x_t.$$

Differentiating with respect to $t$ yields

$$\frac{\mathrm{d}}{\mathrm{d}t}X(t) = u_{s,t}(x_s) + (t - s)\partial_t u_{s,t}(x_s) - \mathrm{d}(\log_{x_s})_{x_t}[v_t] = 0,$$

where the last equality follows from Eq. (48). Since $X(s) = 0$, we conclude that $X(t) \equiv 0$ for all $t$. Therefore,

$$(t - s)u_{s,t}(x_s) = \log_{x_s} x_t,$$

which is exactly the defining relation Eq. (50).

($\Rightarrow$) **Semigroup identity.** Let $\Phi_{s,t}$ denote the true flow map induced by $v_t$. By the semigroup property of the flow,

$$\Phi_{r,t}(\Phi_{s,r}(x_s)) = x_t.$$

Substituting this into the right-hand side of Eq. (49) recovers $\log_{x_s} x_t$, which is equivalent to Eq. (50). Hence the true average velocity satisfies the semigroup condition.

($\Leftarrow$) **Converse.** Assume that the semigroup condition Eq. (49) holds for every $x \in \mathcal{M}$, and that $u_{s,s} = v_s$. We show that the induced map

$$\Phi_{s,t}(x) := \exp_x\big((t - s)u_{s,t}(x)\big)$$

coincides with the true flow map of the time-dependent vector field $v_t$.

By definition,

$$\Phi_{s,s}(x) = \exp_x 0 = x,$$

so $\Phi_{s,s} = \mathrm{Id}_{\mathcal{M}}$. Moreover, by the chain rule,

$$\begin{aligned}
\frac{\mathrm{d}}{\mathrm{d}t}\Phi_{s,t}(x)\Big|_{t=s} &= \mathrm{d}(\exp_x)_{(t-s)u_{s,t}(x)}\big[u_{s,t}(x) + (t - s)\partial_t u_{s,t}(x)\big]\big|_{t=s} \\
&= \mathrm{d}(\exp_x)_0\,[u_{s,s}(x)] \\
&= v_s(x),
\end{aligned}$$

where we used the boundary condition $u_{s,s} = v_s$ and the identity $\mathrm{d}(\exp_x)_0 = \mathrm{Id}_{T_x\mathcal{M}}$. Thus,

$$\frac{\mathrm{d}}{\mathrm{d}t}\Phi_{s,t}(x)\Big|_{t=s} = v_s(x). \tag{53}$$

Next, the semigroup condition implies that for any fixed $r$,

$$\Phi_{s,t} = \Phi_{r,t} \circ \Phi_{s,r}.$$

Differentiating both sides with respect to $t$ yields

$$\frac{\mathrm{d}}{\mathrm{d}t}\Phi_{s,t}(x) = \frac{\mathrm{d}}{\mathrm{d}t}\Phi_{r,t}(\Phi_{s,r}(x)).$$

Evaluating this identity at $r = t$ and using Eq. (53), we obtain

$$\frac{\mathrm{d}}{\mathrm{d}t}\Phi_{s,t}(x) = v_t\big(\Phi_{s,t}(x)\big).$$

Therefore, $\Phi_{s,t}$ satisfies the ODE associated with $v_t$ with initial condition $\Phi_{s,s} = \mathrm{Id}_{\mathcal{M}}$, and hence coincides with the true flow map. Consequently, the corresponding average velocity $u_{s,t}$ satisfies the defining relation Eq. (50) and is the true average velocity. $\qquad\square$

## B.2. Riemannian MeanFlow Objectives

Now, we give the proof of the validity of the training objective (i.e., the minimizer with each proposed training objective is the average velocity).

**Proposition B.2** (Riemannian MeanFlow objectives). *Let $u_{s,t}^\theta : \mathcal{M} \to T\mathcal{M}$ be a parameterized average velocity, and define the induced flow map*

$$\Phi_{s,t}^\theta(x) := \exp_x\big((t - s)\, u_{s,t}^\theta(x)\big).$$

*Consider objectives of the form*

$$\mathcal{L}(\theta) = \mathbb{E}\Big[\big\|u_{s,t}^\theta(\hat{x}_s) - \mathrm{sg}(\hat{u}_{\mathrm{tgt}})\big\|_g^2\Big], \tag{54}$$

*where $\mathrm{sg}(\cdot)$ denotes the stop-gradient operator and the components are defined as follows:*

1. ***Eulerian RMF:*** $\mathbb{E} = \mathbb{E}_{x_s,s,t}$, $\hat{x}_s = x_s$, *and*

$$\hat{u}_{\mathrm{tgt}} = (t - s)\, D_s u_{s,t}^\theta(x_s) - \nabla_{v_s(x_s)}^1 \log_{x_s} \Phi_{s,t}^\theta(x_s).$$

2. ***Lagrangian RMF:*** $\mathbb{E} = \mathbb{E}_{x_t,s,t}$, $\hat{x}_s = \Phi_{t,s}^\theta(x_t)$, *and*

$$\hat{u}_{\mathrm{tgt}} = \mathrm{d}(\log_{\hat{x}_s})_{x_t}[v_t(x_t)] - (t - s)\, \partial_t u_{s,t}^\theta(\hat{x}_s).$$

*To promote invertibility of the induced flow maps, we additionally introduce a cycle-consistency regularizer*

$$\mathcal{L}_{\mathrm{cycle}}(\theta) = \mathbb{E}_{x_t,s,t}\Big[d_g\big(\Phi_{s,t}^\theta(\Phi_{t,s}^\theta(x_t)),\, x_t\big)^2\Big],$$

*which encourages $\Phi_{t,s}^\theta$ to act as an approximate inverse of $\Phi_{s,t}^\theta$ on the data distribution.*

3. ***Semigroup RMF:*** $\mathbb{E} = \mathbb{E}_{x_s,s,r,t}$, $\hat{x}_s = x_s$, *and*

$$\hat{u}_{\mathrm{tgt}} = \frac{1}{t - s} \log_{x_s} \Phi_{r,t}^\theta\big(\Phi_{s,r}^\theta(x_s)\big) \quad (t \neq s),$$

*while $\hat{u}_{\mathrm{tgt}} = v_s(x_s)$ for $t = s$.*

*Then any global minimizer of Eq. (54) satisfies the corresponding identity and hence recovers the average velocity.*

*Proof.* Assume that the sampling distribution of $\hat{x}_s$ has full support on $\mathcal{M}$. Then any global minimizer of Eq. (54) satisfies

$$u_{s,t}^\theta(x_s) = \hat{u}_{\mathrm{tgt}} \quad \text{for all } x_s \in \mathcal{M},\ s, t \in [0, 1].$$

The stop-gradient operator does not affect the set of global minimizers and can therefore be omitted in the analysis. For the semigroup RMF, the equality $u_{s,t}^\theta(x_s) = \hat{u}_{\mathrm{tgt}}$ directly yields the semigroup identity, which implies that $u_{s,t}^\theta$ recovers the average velocity. We therefore focus on the Eulerian and Lagrangian formulations.

**Eulerian RMF.**   Assume that

$$u_{s,t}^{\theta}(x_s) = (t-s)\, D_s u_{s,t}^{\theta}(x_s) - \nabla_{v_s(x_s)}^{1} \log_{x_s} \Phi_{s,t}^{\theta}(x_s) \tag{55}$$

holds for all $x_s \in \mathcal{M}$ and $s,t \in [0,1]$. We show that the induced flow map $\Phi_{s,t}^{\theta}(x_s)$ is independent of $s$ along any integral curve $(x_s)_s$ of $v_s$. This is sufficient since $\Phi_{t,t}^{\theta}(x_t) = x_t$, implying $\Phi_{s,t}^{\theta}(x_s) = x_t$ for all $s$.

Rearranging Eq. (55) yields

$$-u_{s,t}^{\theta}(x_s) + (t-s)\, D_s u_{s,t}^{\theta}(x_s) = \nabla_{v_s(x_s)}^{1} \log_{x_s} \Phi_{s,t}^{\theta}(x_s). \tag{56}$$

By the product rule, the left-hand side can be written as

$$-u_{s,t}^{\theta}(x_s) + (t-s)\, D_s u_{s,t}^{\theta}(x_s) = D_s\big((t-s) u_{s,t}^{\theta}(x_s)\big). \tag{57}$$

Moreover, by definition of the induced flow map,

$$\log_{x_s} \Phi_{s,t}^{\theta}(x_s) = (t-s) u_{s,t}^{\theta}(x_s). \tag{58}$$

Combining Eqs. (57) and (58) with Eq. (56), we obtain

$$D_s\big(\log_{x_s} \Phi_{s,t}^{\theta}(x_s)\big) = \nabla_{v_s(x_s)}^{1} \log_{x_s} \Phi_{s,t}^{\theta}(x_s). \tag{59}$$

Applying the chain rule to the left-hand side of Eq. (59) yields

$$D_s(\log_{x_s} \Phi_{s,t}^{\theta}(x_s)) = \nabla_{v_s(x_s)}^{1} \log_{x_s} \Phi_{s,t}^{\theta}(x_s) + \mathrm{d}(\log_{x_s})_{\Phi_{s,t}^{\theta}(x_s)}\left[\frac{\mathrm{d}}{\mathrm{d}s}\Phi_{s,t}^{\theta}(x_s)\right]. \tag{60}$$

Comparing Eqs. (59) and (60), the terms involving $\nabla^1 \log$ cancel, leaving

$$\mathrm{d}(\log_{x_s})_{\Phi_{s,t}^{\theta}(x_s)}\left[\frac{\mathrm{d}}{\mathrm{d}s}\Phi_{s,t}^{\theta}(x_s)\right] = 0. \tag{61}$$

Since the logarithmic map is a local diffeomorphism, its differential is invertible. Therefore,

$$\frac{\mathrm{d}}{\mathrm{d}s}\Phi_{s,t}^{\theta}(x_s) = 0 \in T_{\Phi_{s,t}^{\theta}(x_s)}\mathcal{M}, \tag{62}$$

and $\Phi_{s,t}^{\theta}(x_s)$ is constant with respect to $s$. This completes the proof.

**Lagrangian RMF.**   Assume that the Lagrangian identity

$$u_{s,t}^{\theta}(\hat{x}_s) = \mathrm{d}(\log_{\hat{x}_s})_{x_t}[v_t(x_t)] - (t-s)\, \partial_t u_{s,t}^{\theta}(\hat{x}_s), \qquad \hat{x}_s := \Phi_{t,s}^{\theta}(x_t), \tag{63}$$

holds for all $x_t \in \mathcal{M}$ and $s,t \in [0,1]$. We additionally assume a (local) invertibility condition on the induced flow maps,

$$\Phi_{s,t}^{\theta}\big(\Phi_{t,s}^{\theta}(x_t)\big) = x_t, \tag{64}$$

which is encouraged in practice by the cycle-consistency regularizer $\mathcal{L}_{\mathrm{cycle}}(\theta) = \mathbb{E}_{x_t,s,t}\big[d_g(\Phi_{s,t}^{\theta}(\Phi_{t,s}^{\theta}(x_t)), x_t)^2\big]$.

Rearranging Eq. (63) yields

$$u_{s,t}^{\theta}(\hat{x}_s) + (t-s)\, \partial_t u_{s,t}^{\theta}(\hat{x}_s) = \mathrm{d}(\log_{\hat{x}_s})_{x_t}[v_t(x_t)].$$

Since the base point $\hat{x}_s$ is fixed when taking $\partial_t$, the left-hand side can be written using the product rule as

$$\frac{\mathrm{d}}{\mathrm{d}t}\big((t-s)\, u_{s,t}^{\theta}(x)\big)\bigg|_{x=\hat{x}_s}.$$

On the other hand, by definition of the induced flow map,

$$\log_x \Phi_{s,t}^{\theta}(x) = (t-s)\, u_{s,t}^{\theta}(x).$$

Differentiating this identity with respect to $t$ while holding $x$ fixed and evaluating at $x = \hat{x}_s$ gives

$$\frac{\mathrm{d}}{\mathrm{d}t} \log_x \Phi^\theta_{s,t}(x) \bigg|_{x=\hat{x}_s} = \mathrm{d}(\log_{\hat{x}_s})_{\Phi^\theta_{s,t}(\hat{x}_s)} \left[\frac{\mathrm{d}}{\mathrm{d}t} \Phi^\theta_{s,t}(\hat{x}_s)\right].$$

Combining the two expressions above, we obtain

$$\mathrm{d}(\log_{\hat{x}_s})_{\Phi^\theta_{s,t}(\hat{x}_s)} \left[\frac{\mathrm{d}}{\mathrm{d}t} \Phi^\theta_{s,t}(\hat{x}_s)\right] = \mathrm{d}(\log_{\hat{x}_s})_{x_t} [v_t(x_t)].$$

Using the invertibility assumption Eq. (64), we have $\Phi^\theta_{s,t}(\hat{x}_s) = x_t$, so both differentials of the logarithmic map are evaluated at the same point. Since the logarithmic map is a local diffeomorphism (away from the cut locus), its differential is invertible, which implies

$$\frac{\mathrm{d}}{\mathrm{d}t} \Phi^\theta_{s,t}(\hat{x}_s) = v_t(x_t) = v_t\big(\Phi^\theta_{s,t}(\hat{x}_s)\big).$$

Finally, since $\hat{x}_s$ ranges over $\mathcal{M}$ as $x_t$ does under the invertibility assumption, we may relabel $\hat{x}_s$ as an arbitrary $x_s \in \mathcal{M}$ to conclude that

$$\frac{\mathrm{d}}{\mathrm{d}t} \Phi^\theta_{s,t}(x_s) = v_t\big(\Phi^\theta_{s,t}(x_s)\big).$$

This is precisely the defining ODE of the true flow map associated with $v_t$. Together with $\Phi^\theta_{s,s} = \mathrm{Id}$, this implies that $\Phi^\theta_{s,t}$ coincides with the true flow map, and hence $u^\theta_{s,t}$ recovers the average velocity.

**Practical remark.** In practice, we find that the Lagrangian objective often trains stably even without explicitly enforcing $\mathcal{L}_{\text{cycle}}$; empirically, the learned maps can become approximately cycle-consistent over the data distribution. We therefore treat $\mathcal{L}_{\text{cycle}}$ as an optional regularizer that can be enabled when stronger invertibility is desired. $\square$

**Marginal velocity approximation with conditional velocity.** Here we justify that, in the training objectives Eq. (54), the marginal velocity field can be replaced by a conditional velocity without affecting the solution characterized by the objective.

Here, $X_t$ denotes the random variable induced by the interpolant process used to couple samples at different times. Specifically, $X_t$ is obtained by sampling a data point and evolving it according to the chosen interpolant between times $0$ and $1$, as in Riemannian flow matching (Chen & Lipman, 2023). The marginal velocity field is defined as the conditional expectation

$$v_t(x) := \mathbb{E}\Big[\dot{X}_t \,\Big|\, X_t = x\Big], \tag{65}$$

interpreted as a tangent vector at $x$.

Fix any smooth operator $L_{x,y} : T_y\mathcal{M} \to T_x\mathcal{M}$ that is *linear* in its input tangent vector, such as $\mathrm{d}(\log_x)_y$ or, more generally, the map $w \mapsto \nabla^1_w \log_x(y)$ for fixed $(x, y)$. By linearity of $L_{x,y}$ and the tower property of conditional expectation, we have

$$\mathbb{E}\Big[L_{X_t,y}\big(\dot{X}_t\big) \,\Big|\, X_t\Big] = L_{X_t,y}\Big(\mathbb{E}\Big[\dot{X}_t \,\Big|\, X_t\Big]\Big) = L_{X_t,y}\big(v_t(X_t)\big), \tag{66}$$

where the first equality follows from linearity of $L_{x,y}$, and the second from the definition Eq. (65). Taking expectations of both sides yields

$$\mathbb{E}\Big[L_{X_t,y}\big(\dot{X}_t\big)\Big] = \mathbb{E}\big[L_{X_t,y}\big(v_t(X_t)\big)\big]. \tag{67}$$

Therefore, in objectives where the velocity field appears only through such linear operators—as is the case for the Eulerian and Lagrangian RMF objectives—the marginal velocity $v_t(X_t)$ can be replaced by the unbiased estimator of the conditional velocity sample $\dot{X}_t$ without changing the expected regression target; the replacement only affects its variance.

## C. Theoretical Connections to Existing Flow-map Learning Methods

### C.1. Eulerian RMF as a Riemannian Generalization of MeanFlow

In this section, we show that the proposed Eulerian RMF reduces exactly to the Euclidean MeanFlow objective when the underlying manifold is $\mathbb{R}^d$. This establishes Eulerian RMF as a direct Riemannian generalization of MeanFlow.

Recall that the Eulerian RMF regression target is given by

$$\hat{u}_{\text{tgt}} = (t - s)\, D_s u_{s,t}^\theta(x_s) - \nabla_{v_s(x_s)}^1 \log_{x_s}\!\big(\Phi_{s,t}^\theta(x_s)\big), \tag{68}$$

where $D_s$ denotes the covariant derivative along the integral curve $x_s$ and $\nabla^1$ denotes the covariant derivative of the logarithmic map with respect to its first (base-point) argument.

We now specialize to the Euclidean setting $\mathcal{M} = \mathbb{R}^d$. In this case, the Levi–Civita connection is flat, and the covariant derivative reduces to the ordinary derivative. In particular, we have

$$D_s u_{s,t}^\theta(x_s) = \frac{d}{ds} u_{s,t}^\theta(x_s). \tag{69}$$

Moreover, the logarithmic map in Euclidean space is given by $\log_x(y) = y - x$, and therefore its derivative with respect to the base point satisfies

$$\nabla_{v_s(x_s)}^1 \log_{x_s}\!\big(\Phi_{s,t}^\theta(x_s)\big) = -v_s(x_s). \tag{70}$$

Substituting these identities into (68), the Eulerian RMF target reduces to

$$\hat{u}_{\text{tgt}} = (t - s)\, \frac{d}{ds} u_{s,t}^\theta(x_s) + v_s(x_s), \tag{71}$$

which is exactly the regression target used in Euclidean MeanFlow (Geng et al., 2025). This shows that Eulerian RMF recovers MeanFlow in the Euclidean case, while providing a principled extension to general Riemannian manifolds through intrinsic geometric operators.

### C.2. Connection to Generalized Flow Map

In this section, we detail a theoretical connection between Riemannian MeanFlow and Generalized Flow Map (GFM) objectives. We first derive our objectives from GFM self-distillation objectives and show that our objective can be viewed as GFM objectives with a properly applied stop-gradient operation, thereby entirely avoiding backpropagation through Jacobian-vector products (JVPs). This suggests that our derivation reveals a further, practically important design space that is not focused on GFM objectives.

#### C.2.1. BRIEF DERIVATION OF GFM OBJECTIVES

We begin by briefly reviewing the objectives proposed in GFM. Their derivation starts from the defining relation of the flow map introduced in Definition 2.2:

$$\Phi_{s,t}(x_s) = x_t, \tag{72}$$

which holds for any integral curve $(x_t)_{t \in [0,1]}$ and any $s, t \in [0, 1]$. To construct learning objectives, GFM differentiates this identity with respect to the time variables $s$ and $t$. Differentiation with respect to the source time $s$ yields Eulerian-type objectives, while differentiation with respect to the target time $t$ yields Lagrangian-type objectives.

Specifically, differentiating (72) with respect to $s$ gives the generalized Eulerian characterization

$$\frac{d}{ds} \Phi_{s,t}(x_s) = \partial_s \Phi_{s,t}(x_s) + d\Phi_{s,t}(x_s)[v_s] = 0, \tag{73}$$

where $v_s$ denotes the velocity along the integral curve. GFM enforces this identity via the regression objective

$$\mathcal{L}_{\text{G-ESD}}(\theta) = \mathbb{E}_{x_s, s, t}\left[\left\|\partial_s \Phi_{s,t}^\theta(x_s) + d\Phi_{s,t}^\theta(x_s)\big[v_s^\theta(x_s)\big]\right\|_g^2\right], \tag{74}$$

referred to as the *G-ESD* objective.

Likewise, differentiating (72) with respect to $t$ yields the Lagrangian characterization

$$\partial_t \Phi_{s,t}(x_s) = v_t(x_t) = v_t(\Phi_{s,t}(x_s)),$$ (75)

which is precisely the defining property of an integral curve, i.e., $\Phi_{s,t}(x_s)$ solves the ODE $\frac{d}{dt}x_t = v_t(x_t)$. Enforcing this identity via regression leads to the *G-LSD* objective,

$$\mathcal{L}_{\text{G-LSD}}(\theta) = \mathbb{E}_{x_s,s,t}\left[\left\|\partial_t \Phi_{s,t}^\theta(x_s) - v_t^\theta\big(\Phi_{s,t}^\theta(x_s)\big)\right\|_g^2\right].$$ (76)

Finally, GFM introduces a semigroup objective that enforces the semigroup property of the flow map, $\Phi_{r,t}(\Phi_{s,r}(x_s)) = \Phi_{s,t}(x_s)$, via

$$\mathcal{L}_{\text{G-PSD}}(\theta) = \mathbb{E}_{x_s,s,r,t}\left[d_g^2\big(\Phi_{s,t}^\theta(x_s), \Phi_{r,t}^\theta\big(\Phi_{s,r}^\theta(x_s)\big)\big)\right].$$ (77)

For practical optimization, GFM applies the stop-gradient operator to obtain the following losses:

$$\mathcal{L}_{\text{G-ESD}}(\theta) = \mathbb{E}_{x_s,s,t}\left[\left\|\partial_s \Phi_{s,t}^\theta(x_s) + \text{sg}\big(d\Phi_{s,t}^\theta(x_s)\big[v_s^\theta(x_s)\big]\big)\right\|_g^2\right],$$ (78)

$$\mathcal{L}_{\text{G-LSD}}(\theta) = \mathbb{E}_{x_s,s,t}\left[\left\|\partial_t \Phi_{s,t}^\theta(x_s) - \text{sg}\big(v_t^\theta\big(\Phi_{s,t}^\theta(x_s)\big)\big)\right\|_g^2\right],$$ (79)

$$\mathcal{L}_{\text{G-PSD}}(\theta) = \mathbb{E}_{x_s,s,r,t}\left[d_g^2\big(\Phi_{s,t}^\theta(x_s), \text{sg}\big(\Phi_{r,t}^\theta\big(\Phi_{s,r}^\theta(x_s)\big)\big)\big)\right],$$ (80)

where $\text{sg}$ denotes the stop-gradient operator.

Importantly, the objectives G-ESD, G-LSD, and G-PSD enforce only *flow-map consistency*. This consistency alone does not guarantee that the velocity field $v_s^\theta$ corresponds to the true data-generating dynamics. To recover the desired flow map, GFM therefore augments the above objectives with a flow-matching loss that explicitly trains $v_s^\theta$.

### C.2.2. Summary of Key Differences Between RMF and GFM

In the following, we analyze these objectives in more detail and clarify the role of the stop-gradient operator in GFM. In prior works on consistency models (Song et al., 2023) and MeanFlow (Geng et al., 2025), the stop-gradient operator is primarily introduced to avoid computing expensive higher-order derivatives (e.g., gradients through Jacobian–vector products) and to improve computational efficiency and optimization stability. In GFM, stop-gradient is likewise employed in the differential objectives; however, due to the structure of these objectives, it blocks higher-order derivatives only partially and therefore does not fully eliminate the associated computational overhead. From this perspective, our formulation yields differential objectives in which higher-order derivatives are avoided by construction, resulting in improved computational efficiency and empirical stability.

Finally, for the semigroup objective, we show that the G-PSD objective and our corresponding formulation can be heuristically related through loss weighting with respect to the length of the time interval. We empirically demonstrate that our formulation leads to superior performance in App. D.

### C.2.3. From G-ESD to Eulerian RMF

We show that the Eulerian RMF objective arises as a first-order expansion of the GFM Eulerian self-distillation (G-ESD) objective under an exponential-map parameterization of the flow map. Recall that the G-ESD objective is based on the Eulerian consistency residual

$$\Delta_{\text{G-ESD}}(x) := \partial_s \Phi_{s,t}^\theta(x) + d\Phi_{s,t}^\theta(x)[v_s(x)],$$ (81)

where $\partial_s$ denotes the partial derivative with respect to the flow-map parameter $s$.

We parameterize the flow map using the average velocity field as

$$\Phi_{s,t}^\theta(x) = \exp_x\big((t-s)\,u_{s,t}^\theta(x)\big),$$ (82)

and define $\xi(x) := (t-s)\,u_{s,t}^\theta(x)$. In what follows, we evaluate all expressions along the interpolant $x = x_s$.

**Expansion of the time derivative.** Using the chain rule for the exponential map, we obtain

$$\partial_s \Phi_{s,t}^\theta(x_s) = d_2 \exp_{x_s}(\xi_s)\Big[ - u_{s,t}^\theta(x_s) + (t - s)\,\partial_s u_{s,t}^\theta(x_s)\Big], \tag{83}$$

where $d_2 \exp$ denotes the differential of $\exp$ with respect to its second (tangent-vector) argument.

**Expansion of the pushforward term.** Writing $\Phi_{s,t}^\theta(x) = \exp_x(\xi(x))$, the differential with respect to the base point $x$ yields

$$d\Phi_{s,t}^\theta(x_s)[v_s(x_s)] = d_1 \exp_{x_s}(\xi_s)[v_s(x_s)] + d_2 \exp_{x_s}(\xi_s)\big[\nabla_{v_s(x_s)}\xi(x_s)\big], \tag{84}$$

where $d_1 \exp$ denotes the differential with respect to the base point.

Combining (83) and (84), the G-ESD residual becomes

$$\Delta_{\text{G-ESD}}(x_s) = d_1 \exp_{x_s}(\xi_s)[v_s(x_s)] + d_2 \exp_{x_s}(\xi_s)\big[\nabla_{v_s(x_s)}\xi(x_s) - u_{s,t}^\theta(x_s) + (t - s)\,\partial_s u_{s,t}^\theta(x_s)\big]. \tag{85}$$

**Pull-back to the tangent space.** To express the residual in $T_{x_s}\mathcal{M}$, we pull it back via the differential of the log map at $x_s$. Let $y_s := \Phi_{s,t}^\theta(x_s) = \exp_{x_s}(\xi_s)$ and define

$$\widehat{\Delta}_{\text{G-ESD}}(x_s) := d(\log_{x_s})_{y_s}[\Delta_{\text{G-ESD}}(x_s)]. \tag{86}$$

Within a normal neighborhood, the identities

$$d(\log_{x_s})_{y_s} \circ d_2 \exp_{x_s}(\xi_s) = \text{Id}_{T_{x_s}\mathcal{M}}, \qquad d(\log_{x_s})_{y_s}\big[d_1 \exp_{x_s}(\xi_s)[v]\big] = -\nabla_v^1 \log_{x_s}(y_s) \tag{87}$$

hold, where $\nabla^1$ denotes the covariant derivative with respect to the first argument of the log map.

Applying these identities to (85), we obtain

$$\widehat{\Delta}_{\text{G-ESD}}(x_s) = -\nabla_{v_s(x_s)}^1 \log_{x_s}\big(\Phi_{s,t}^\theta(x_s)\big) - u_{s,t}^\theta(x_s) + (t - s)\Big(\partial_s u_{s,t}^\theta(x_s) + \nabla_{v_s(x_s)} u_{s,t}^\theta(x_s)\Big). \tag{88}$$

Introducing the covariant derivative along the interpolant,

$$D_s u_{s,t}^\theta(x_s) := \partial_s u_{s,t}^\theta(x_s) + \nabla_{v_s(x_s)} u_{s,t}^\theta(x_s), \tag{89}$$

the residual can be written compactly as

$$\widehat{\Delta}_{\text{G-ESD}}(x_s) = -\nabla_{v_s(x_s)}^1 \log_{x_s}\big(\Phi_{s,t}^\theta(x_s)\big) - u_{s,t}^\theta(x_s) + (t - s)\, D_s u_{s,t}^\theta(x_s). \tag{90}$$

Setting (90) to zero yields

$$u_{s,t}^\theta(x_s) = (t - s)\, D_s u_{s,t}^\theta(x_s) - \nabla_{v_s(x_s)}^1 \log_{x_s}\big(\Phi_{s,t}^\theta(x_s)\big), \tag{91}$$

which exactly recovers the Eulerian RMF regression target used in our method.

**Relation between G-ESD and Eulerian RMF objectives.** The above derivation shows that the G-ESD and Eulerian RMF objectives enforce the same Eulerian consistency condition, differing primarily in the space in which the residual is represented. G-ESD minimizes the residual $\Delta_{\text{G-ESD}}(x_s) \in T_{\Phi_{s,t}^\theta(x_s)}\mathcal{M}$, defined at the transported point $\Phi_{s,t}^\theta(x_s)$. In contrast, Eulerian RMF applies an invertible change of coordinates given by the log-map differential $d(\log_{x_s})_{\Phi_{s,t}^\theta(x_s)}$, which pulls the residual back to the reference tangent space $T_{x_s}\mathcal{M}$. The two residuals are related by

$$\widehat{\Delta}_{\text{G-ESD}}(x_s) = d(\log_{x_s})_{\Phi_{s,t}^\theta(x_s)}\big[\Delta_{\text{G-ESD}}(x_s)\big],$$

so both objectives share the same zero set of the consistency constraint.

This perspective also clarifies the practical effect of stop-gradient. In Eulerian RMF, stop-gradient is applied to the entire regression target. In contrast, G-ESD applies stop-gradient at the level of the Eulerian residual; under the pull-back above, this induces a *partial* stop-gradient in the Eulerian RMF form, acting only on the geometric transport terms $\nabla_{v_s(x_s)}^1 \log_{x_s}\big(\Phi_{s,t}^\theta(x_s)\big)$ and $\nabla_{v_s(x_s)} u_{s,t}^\theta(x_s)$, while leaving the explicit time-derivative term $\partial_s u_{s,t}^\theta(x_s)$ differentiable.

### C.2.4. FROM G-LSD TO LAGRANGIAN RMF.

We now establish an analogous connection between the Lagrangian objectives of GFM and our Lagrangian RMF. Recall that the GFM Lagrangian self-distillation (G-LSD) objective is defined as

$$\mathcal{L}_{\text{G-LSD}}(\theta) = \mathbb{E}_{x_s,s,t}\left[\left\|\partial_t \Phi_{s,t}^\theta(x_s) - v_t^\theta\left(\Phi_{s,t}^\theta(x_s)\right)\right\|_g^2\right], \tag{92}$$

which enforces the Lagrangian consistency condition $\partial_t \Phi_{s,t}(x_s) = v_t(\Phi_{s,t}(x_s))$ along transported particles.

As in the Eulerian case, we adopt the average-velocity parameterization

$$\Phi_{s,t}^\theta(x) = \exp_x\left((t-s)\, u_{s,t}^\theta(x)\right), \tag{93}$$

and evaluate all quantities along the interpolant $x = x_s$. Differentiating with respect to $t$ yields

$$\partial_t \Phi_{s,t}^\theta(x_s) = d_2 \exp_{x_s}(\xi_s)\left[u_{s,t}^\theta(x_s) + (t-s)\,\partial_t u_{s,t}^\theta(x_s)\right], \qquad \xi_s := (t-s)u_{s,t}^\theta(x_s). \tag{94}$$

To express the velocity term in a compatible form, we use the identity $\exp_{x_s}(\log_{x_s} y) = y$, which implies

$$v_t\left(\Phi_{s,t}^\theta(x_s)\right) = d_2 \exp_{x_s}(\xi_s)\left[d_2 \log_{x_s}\left(\Phi_{s,t}^\theta(x_s)\right)\left[v_t(\Phi_{s,t}^\theta(x_s))\right]\right]. \tag{95}$$

Substituting (94) and (95) into (92), and using the invertibility of $d_2 \exp_{x_s}(\xi_s)$ within a normal neighborhood, the G-LSD objective reduces to an equivalent regression in $T_{x_s}\mathcal{M}$:

$$\mathcal{L}_{\text{G-LSD}}(\theta) \equiv \mathbb{E}_{x_s,s,t}\left[\left\|u_{s,t}^\theta(x_s) + (t-s)\,\partial_t u_{s,t}^\theta(x_s) - d_2 \log_{x_s}\left(\Phi_{s,t}^\theta(x_s)\right)\left[v_t(\Phi_{s,t}^\theta(x_s))\right]\right\|_g^2\right]. \tag{96}$$

This expression coincides with the Lagrangian RMF objective up to the treatment of stop-gradient. In particular, while Lagrangian RMF applies stop-gradient to the entire regression target, the stop-gradient G-LSD objective—defined at the flow-map level—induces a *partial* stop-gradient under the above pull-back, affecting the transport and log-map terms while leaving the explicit time-derivative $\partial_t u_{s,t}^\theta(x_s)$ differentiable. As in the Eulerian case, this shows that Lagrangian RMF can be interpreted as a geometrically equivalent reparameterization of G-LSD, expressed in the tangent space of the current state.

### C.2.5. GENERALISED MEANFLOW VS. EULERIAN RMF.

Davis et al. (2025) propose a generalization of the MeanFlow objective to Riemannian manifolds. However, as the authors acknowledge, this is not straightforward:

> "Generalising Mean Flows directly is non-trivial as it requires defining the integral of a vector field on a manifold properly, which would involve parallel transport and therefore derivatives thereof. Also, Mean Flows operate on the vector field level as opposed to Flow Map Matching which operates on the level of the flow map. It is difficult to go from one level to another directly, as it will involve nontrivial curvature terms."

As a result, Davis et al. (2025) propose a heuristic objective by substituting Riemannian differential operators where appropriate:

> "Instead, we propose to heuristically follow our derivations in Appendix A, in the 'stopgradients' section. Indeed, we can see that our loss involves the instantaneous vector field of the modelled flow map, $v_{t,t}^\theta$, as opposed to the ideal flow $\partial_t I_t$; hence the use of the Levi-Civita connection along $v_s(I_s)$ instead of the differential evaluated at $v_{s,s}^\theta$, which indeed recovers the Euclidean case as a special case."

$$\widehat{\mathcal{L}}_{\text{G-MF}}(\theta) = \mathbb{E}_{t,s,x_s}\left[\left\|u_{s,t}^\theta(x_s) - \text{stopgrad}\left(v_s - (t-s)\nabla_{v_s} u_{s,t}^\theta(x_s)\right)\right\|_g^2\right]. \tag{97}$$

In contrast, our Eulerian RMF is derived from a different starting point. Rather than relying on an integral formulation of the average velocity field, we base the derivation on identities satisfied by the average velocities along integral curves of the flow.

This perspective entirely avoids the need to define integrals of vector fields on the manifold. As a result, the characterizing identity of the average velocity can be differentiated directly using covariant derivatives, yielding Eulerian learning objectives that are intrinsic and well-defined on Riemannian manifolds. By grounding the derivation in integral-curve-based identities, Eulerian RMF provides a direct and principled generalization of Euclidean MeanFlow (Geng et al., 2025), effectively replacing the heuristically constructed G-MF objective.

### C.2.6. SEMIGROUP RMF VS. G-PSD.

We compare the Semigroup RMF and G-PSD objectives in the Euclidean setting and show that they differ only by a time-dependent loss weighting. In Euclidean space, the Semigroup RMF per-sample loss is given by

$$L_{\text{S-RMF}}(\theta) = \left\| u_{s,t}^\theta(x_s) - \text{sg}\left( \frac{r-s}{t-s} u_{s,r}^\theta(x_s) + \frac{t-r}{t-s} u_{r,t}^\theta(\hat{x}_r) \right) \right\|_2^2, \tag{98}$$

with $\hat{x}_r = \Phi_{s,r}^\theta(x_s) = x_s + (r-s)u_{s,r}^\theta(x_s)$. In the same setting, the G-PSD objective reduces to

$$L_{\text{G-PSD}}(\theta) = \left\| (t-s) u_{s,t}^\theta(x_s) - \text{sg}\big((r-s) u_{s,r}^\theta(x_s) + (t-r) u_{r,t}^\theta(\hat{x}_r)\big) \right\|_2^2. \tag{99}$$

The two objectives are related by

$$L_{\text{G-PSD}}(\theta) = (t-s)^2 L_{\text{S-RMF}}(\theta), \tag{100}$$

indicating that their difference can be fully characterized by a time-dependent loss weighting $w(s,t) = (t-s)^2$.

On general Riemannian manifolds, this equivalence no longer holds exactly due to curvature-dependent effects. In practice, we observe that Semigroup RMF achieves slightly better performance than G-PSD.

# D. Empirical Comparison to GFM

In this section, we empirically compare our approach to the concurrent Generalized Flow Matching (GFM) method (Davis et al., 2025) using the geospatial Earth dataset and a high-dimensional DNA promoter design task. Our results indicate that while GFM struggles to scale reliably in the DNA task, our method remains stable and performs better due to our proposed stabilization techniques. Furthermore, we analyze the computational overhead of both methods, specifically comparing training time, memory usage, and NFEs per iteration. These comparisons highlight the superior optimization behavior and scaling properties of our framework over GFM.

## D.1. Toy Earth Datasets

To provide a quantitative comparison between our proposed objectives and GFM (Davis et al., 2025), we evaluate our method on the geospatial Earth events benchmark on $\mathbb{S}^2$ (Mathieu & Nickel, 2020), following the evaluation protocols established in Chen & Lipman (2023) and Davis et al. (2025). In these experiments, we fix the parameterization to $v$-prediction to isolate and investigate the impact of different training objectives.

**Metric.** We report the empirical Maximum Mean Discrepancy (MMD) between the test data and generated samples, consistent with Davis et al. (2025). For the MMD computation, we employ a geodesic-based RBF kernel, $k(x, y) = \exp(-d_g(x,y)^2/(2\kappa^2))$, with a bandwidth $\kappa = 1$. We omit the test Negative Log-Likelihood (NLL) because exact NLL evaluation is intractable unless the flow map is strictly invertible or satisfies specific regularity conditions (Rehman et al., 2025).

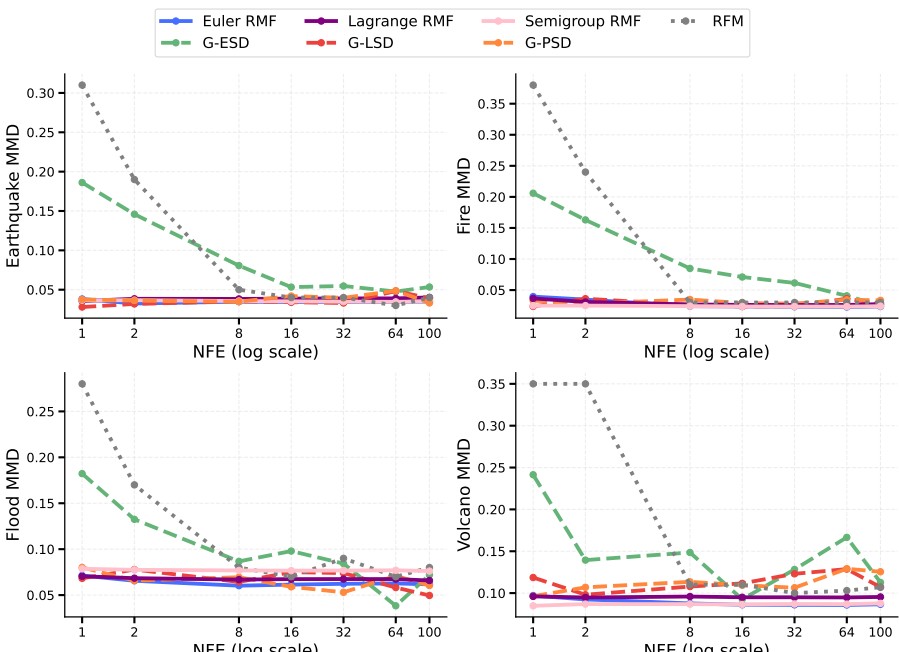

*Figure A1.* **Inference steps vs. MMD on the Earth dataset.** Empirical MMD (lower is better) as a function of the number of inference steps for four datasets (Volcano, Earthquake, Flood, and Fire).

**Results.** As illustrated in Fig. A1, our Riemannian MeanFlow objectives achieve competitive or superior MMD values on the Earth dataset compared to the GFM baseline. Notably, all of our objective variants (Eulerian, Lagrangian, and Semigroup) yield comparable results with 100-step sampling. In contrast, GFM-Eulerian exhibits a significantly higher MMD in its 1-step performance. We hypothesize that this discrepancy arises because GFM's formulation does not entirely bypass backpropagation through the Jacobian-vector product (JVP), which potentially destabilizes the optimization process compared to our JVP-free objectives.

*Table A1.* One-step performance on test-set of promoter DNA sequence.

| Method | | MSE ($\downarrow$) | $k$-mer corr. ($\uparrow$) |
|---|---|---|---|
| G-ESD | $v$-pred | 0.055 | 0.13 |
| G-LSD | $v$-pred | 0.046 | 0.73 |
| G-PSD | $v$-pred | 0.035 | 0.82 |
| Euler RMF | $x_1$-pred | $0.030_{\pm 0.000}$ | $0.96_{\pm 0.01}$ |
| | $v$-pred | $0.031_{\pm 0.001}$ | $0.96_{\pm 0.00}$ |
| Lagrange RMF | $x_1$-pred | $0.027_{\pm 0.001}$ | $0.88_{\pm 0.00}$ |
| | $v$-pred | $0.027_{\pm 0.001}$ | $0.85_{\pm 0.01}$ |
| Semigroup RMF | $x_1$-pred | $0.030_{\pm 0.001}$ | $0.84_{\pm 0.03}$ |
| | $v$-pred | $0.030_{\pm 0.001}$ | $0.93_{\pm 0.02}$ |

*Table A2.* Training cost of G-ESD and Eulerian MF. We measure the memory consumption and training time per iteration on the DNA task.

| | NFE/it | Memory (GB) $\downarrow$ | Training time (s/it) $\downarrow$ |
|---|---|---|---|
| G-ESD | 3 | 17.7 | 0.40 |
| E-RMF | 1 | 9.5 | 0.15 |
| E-RMF self-distillation | 2 | 9.5 | 0.16 |

### D.2. Promoter DNA Datasets

To demonstrate the scalability of our objectives compared to GFM (Davis et al., 2025), we evaluate both methods on a high-dimensional DNA promoter design task. For fair comparison, we fix the model architecture, optimizer, time sampling scheme, batch size, and the total number of training iterations. The experimental setup for these evaluations is consistent with the protocols described in Sec. 4; further implementation details can be found in Apps. E and F.

**Results.** As summarized in Table A1, we observe a significant performance gap between the two frameworks. Specifically, GFM's differential objectives—Eulerian (G-ESD) and Lagrangian (G-LSD)—fail to achieve meaningful convergence, resulting in high MSE. We hypothesize that this failure stems from optimization instabilities caused by the high variance of the network output's derivatives in high-dimensional spaces. In contrast, all variants of our Riemannian MeanFlow (Eulerian, Lagrangian, and Semigroup) maintain robust stability throughout training. Our methods consistently outperform GFM across all metrics.

### D.3. Computational Efficiency Analysis

To further evaluate the practical utility of our proposed objectives, we conduct a comparative analysis of the computational costs between our Eulerian RMF (Eulerian RMF) and the Generalized Flow Map Eulerian (G-ESD) objective (Davis et al., 2025). As summarized in Table A2, our method demonstrates superior efficiency in terms of both memory consumption and training speed.

**Number of function evaluations (NFEs).** A significant advantage of our formulation is the reduction in the number of network evaluations per training iteration. G-ESD requires 3 NFEs per iteration: because the stop-gradient operator is applied only to the spatial derivative $d\Phi^{\theta}_{s,t}$, the time-derivative $\partial_s \Phi^{\theta}_{s,t}$ and the spatial JVP term must be evaluated separately. In contrast, our Eulerian RMF evaluates both the average velocity $u^{\theta}_{s,t}$ and its time derivative $D_s u^{\theta}_{s,t}$ within a single forward pass, requiring only 1 NFE. Even in our self-distillation variant, which utilizes a learned instantaneous velocity, the cost remains at 2 NFEs, still lower than that of the standard GFM Eulerian objective.

**Memory usage and optimization.** All memory and throughput numbers are measured on an NVIDIA RTX 3090 GPU. As shown in Table A2, G-ESD incurs substantially higher memory usage (17.7 GB) compared to Eulerian RMF (9.5 GB). This disparity arises because G-ESD does not fully avoid backpropagation through the JVP output $\partial_s \Phi^{\theta}_{s,t}$. The resulting computational graph for G-ESD is more complex, requiring approximately twice the memory of our method. Our JVP-free formulation not only reduces the memory footprint but also contributes to the optimization stability observed in high-dimensional tasks like DNA promoter design, where G-ESD failed to converge (Table A1).

**Training throughput.** The combination of fewer NFEs and lower memory overhead leads to a marked improvement in training speed. Eulerian RMF achieves a training time of 0.15 s/it, which is approximately 2.6× faster than G-ESD's 0.40 s/it. The self-distillation variant of Eulerian RMF also maintains high throughput (0.16 s/it), demonstrating that the efficiency gains are robust across different variants of our objective.

# E. Evaluation details

## E.1. DNA Promoter Design

**Task description.** Promoters are critical DNA sequences that dictate the initiation and magnitude of gene transcription (Haberle & Stark, 2018). The objective of this task is to generate promoter sequences conditioned on a desired transcription signal profile. Successful generation of these sequences enables precise control over the expression levels of synthetic genes, which is essential for applications such as controlled gene expression and synthetic biology. For DNA promoter design, we use MSE for evaluation following Davis et al. (2024); Stark et al. (2024), and additionally report $k$-mer correlation to assess whether local sequence patterns are preserved.

**MSE.** Following prior work, we evaluate promoter activity using mean squared error (MSE) between transcription signal profiles predicted by a pretrained Sei model from generated sequences and those predicted from the corresponding test-set reference sequences under the same condition. Specifically, for each conditioning signal in the test set, we compute Sei-predicted profiles for both the generated and reference sequences, measure their MSE, and report the average over the test set. This metric quantifies how closely the generated sequences match the regulatory activity of the reference sequences.

**$k$-mer correlation.** In addition to MSE, we measure the $k$-mer correlation between the generated sequences and the empirical test distribution. This metric evaluates whether the generated promoter sequences preserve local sequence patterns, capturing compositional similarity beyond global activity prediction. Concretely, we aggregate the generated sequences into a single $k$-mer frequency vector, aggregate the test-set sequences into another $k$-mer frequency vector, and report the Pearson correlation between the two vectors.

## E.2. DNA Promoter Reward Guidance

**Reward function.** For optimizing the regulatory footprint of a DNA sequence based on a target profile, we use the following reward function:

$$r(x) = -\frac{1}{|\mathcal{N}|} \sum_{n \in \mathcal{N}} \left( \mathrm{Sei}(x)[n] - \mathrm{Sei}(x_{\mathrm{target}})[n] \right)^2, \tag{101}$$

where $\mathcal{N}$ corresponds to the promoter-related features from Sei. Note that lower MSE is better, so we negate the MSE metric to maximize it.

**Ablations on reward guidance scale.** We perform a grid search over the guidance scale ($\lambda$) in Eq. (21), evaluating values $\lambda \in \{1, 10, 100, 1000\}$ and show the performance in Table A3. For each type of reward guidance at each NFE, we report the best performance in Table 2.

*Table A3.* Ablation over the guidance scale $\lambda$ in Eq. (21) for reward-guided promoter DNA generation. We report mean squared error (MSE) $\pm$ standard deviation across 60 batches of 128 samples for two reward evaluations: the naive approach based on the current state ($\nabla r(x_t)$) and using $x_1$ look-ahead ($\nabla r(\Phi^{\theta}_{t,1}(x_t))$).

| NFE | $\lambda$ | $\nabla r(x_t)$ | $\nabla r(\Phi^{\theta}_{t,1}(x_t))$ |
|---|---|---|---|
| 1 | 1 | $0.033_{\pm 0.015}$ | $0.026_{\pm 0.011}$ |
| | 10 | $0.033_{\pm 0.015}$ | $0.025_{\pm 0.011}$ |
| | 100 | $0.033_{\pm 0.015}$ | $0.049_{\pm 0.033}$ |
| | 1000 | $0.033_{\pm 0.015}$ | $0.068_{\pm 0.053}$ |
| 5 | 1 | $0.031_{\pm 0.014}$ | $0.021_{\pm 0.008}$ |
| | 10 | $0.031_{\pm 0.014}$ | $0.013_{\pm 0.005}$ |
| | 100 | $0.026_{\pm 0.011}$ | $0.024_{\pm 0.013}$ |
| | 1000 | $0.017_{\pm 0.009}$ | $0.048_{\pm 0.039}$ |
| 10 | 1 | $0.031_{\pm 0.013}$ | $0.017_{\pm 0.007}$ |
| | 10 | $0.031_{\pm 0.013}$ | $0.008_{\pm 0.003}$ |
| | 100 | $0.025_{\pm 0.010}$ | $0.016_{\pm 0.007}$ |
| | 1000 | $0.008_{\pm 0.002}$ | $0.036_{\pm 0.029}$ |

### E.3. Protein Backbone Design

We mainly follow the evaluation pipeline and metric definition of FrameFlow (Yim et al., 2023a), FrameDiff (Yim et al., 2023b), and La Proteina (Geffner et al., 2025a). We sample 10 backbone structures for every length between 60 and 128, and measure three metrics for the generated samples: designability, diversity, and novelty. For Table 3, we have reproduced FrameDiff and FrameFlow, and have report metrics below.

**Designability.** We assess the designability with the *self-consistency* evaluation from Trippe et al. (2023), measuring how closely a generated backbone can be recovered by sequence design and refolding. To be specific, we use ProteinMPNN (Dauparas et al., 2022) to achieve 8 sequences for each backbone structure and re-fold, i.e., predict their backbone structures using ESMfold (Lin et al., 2023). Afterwards, we compute the root-mean-square-distance (scRMSD) between the backbone structures and the generated backbone structure. We also report the designable fraction with a threshold of scRMSD $< 2.0$ Å.

**Diversity.** Diversity measures how many distinct structural conformations the model generates. For each length, we cluster all the generated backbones using MaxCluster (Herbert & Sternberg, 2008), and report the number of clusters divided by the total number of designable samples. Additionally, we also report the pairwise scTM, which quantifies the structural similarity between the generated backbones. A lower pairwise scTM indicate higher diversity, as they reflect larger structural deviation between the generated samples . The MaxCluster command used to compute this metric is given by

```
maxcluster -l <pdb file list> -C 3 -Tm 0.8 -noalign
```

where `<pdb file list>` is the path for a text file containing the list of paths to PDB files.

**Novelty.** Novelty evaluates how different the generated backbones are from known protein structures. For each designable sample, we use FoldSeek (Van Kempen et al., 2024) to search the PDB database and compute the highest TM-score to any matching chain (Zhang & Skolnick, 2005, pdbTM). Afterwards, we report the average pdbTM across all samples. The FoldSeek command used to compute this metric is given by

```
foldseek easy-search <path sample> <reference database path> <result file>
<tmp path> --format-output query,target,alntmscore
```

where `<path sample>` is the path for PDB files containing the generated structure, and `<reference database path>` is the path of the dataset used as reference, for which we use the Protein Data Bank (PDB).

### E.4. Protein Reward Guidance

**Reward function.** We design a differentiable reward function as based on PyDSSP[1], an open-source implementation of the Define Secondary Structure of Proteins (DSSP) algorithm (Kabsch & Sander, 1983), which is the standard method secondary structure assignment to protein residues. PyDSSP implements DSSP in PyTorch.

PyDSSP first computes a hydrogen-bond energy map using the DSSP electrostatic model, where hydrogen bond energies are calculated from interatomic distances between backbone atoms (O–N, C–H, O–H, and C–N), and then uses a smooth, differentiable approximation to determine hydrogen bond presence. Secondary structure elements are then identified from this map following DSSP-style rules: turns are detected from diagonal hydrogen bonds between residues separated by three to five positions, helices are defined by consecutive turns, and $\beta$-bridges are identified via parallel and antiparallel hydrogen-bond patterns using a local unfolding window. However, in order to identify these structural elements, non-differentiable boolean tensors are produced and gradients are broken.

Instead, we construct a soft score for each each structural class independently, which we call DiffDSSP. First, we compute pairwise electrostatic H-bond energies between all residue pairs using the DSSP energy formula:

$$e = 0.084 \cdot \left( \frac{1}{\mathrm{d_{ON}}} + \frac{1}{\mathrm{d_{CH}}} - \frac{1}{\mathrm{d_{OH}}} - \frac{1}{\mathrm{d_{CN}}} \right) \cdot 332 \tag{102}$$

Then, we pass energies through a sigmoid:

---

[1] https://github.com/ShintaroMinami/PyDSSP

$$\text{hydrogen\_bond\_map} = \texttt{sigmoid}(-(e - 0.5 + \text{margin})) \tag{103}$$

where $-0.5$ is the cutoff value for hydrogen-bond presence. We extract differentiable pattern scores from the hydrogen-bond map corresponding to $\alpha$-helices and $\beta$-sheets. These scores quantify the extent to which each residue participates in helix- or strand-like hydrogen-bond patterns, without requiring hard assignments or binary decisions as in the original DSSP algorithm. We compute the helix and strand fractions by averaging the per-residue helix and strand scores across residues to get scalar fractions in [0,1], denoted as $\text{DiffDSSP}_\alpha(x)$ and $\text{DiffDSSP}_\beta(x)$. Finally, we compute the mean-squared error between the predicted fractions and the target fractions ($w_\alpha$ or $w_\beta$), and return the negative value so the reward can be maximized:

$$r(x) = -\left((w_\alpha - \text{DiffDSSP}_\alpha(x))^2 + (w_\beta - \text{DiffDSSP}_\beta(x))^2\right) , \tag{104}$$

When steering toward increased $\beta$-sheet content, we set $w_\beta = 1$ and $w_\alpha = -1$, while for $\alpha$-helix guidance, we set $w_\beta = -1$ and $w_\alpha = 1$.

**Final evaluation.** To evaluate the final generated sequences, we used the original DSSP algorithm (Kabsch & Sander, 1983), which is the standard way for assigning structural classes to a protein. We used $\texttt{dssp-2.0.4-linux-amd64}$[2] with the following command:

```
dssp -i <pdb file list>
```

**Ablations on reward guidance scale.** In Table A4 and Table A5, we compute the $\alpha$-helix and $\beta$-sheet content, respectively, of generated proteins at different guidance scales ($\lambda$) for each reward state ($\zeta_t$) and each NFE. In Table A5, we ablate scaling the reward by $t$ (i.e., $r_t(x) = t \cdot r(x)$), similar to (Sabour et al., 2025a), although we find this generally does not help in our ad-hoc setting. We keep the settings with the best top-10 mean for each $\zeta_t$ and NFE, and report these values in Table 4. We also evaluated the performance at 1 NFE, although there was no significant difference compared to no guidance.

---

[2]https://swift.cmbi.umcn.nl/gv/dssp/

*Table A4.* Ablation over the guidance scale $\lambda$ in Eq. (21) for guiding $\alpha$-helix generation in proteins. We report mean squared error (MSE) $\pm$ standard deviation across 100 samples of length 128 for two reward evaluations, the naive approach based on the current state ($\nabla r(x_t)$) and using $x_1$ look-ahead ($\nabla r(\Phi^\theta_{t,1}(x_t))$), as well as no guidance (—).

| NFE | $\zeta_t$ | Guidance scale ($\lambda$) | Mean ($\uparrow$) | Top-10 mean ($\uparrow$) | Max ($\uparrow$) | Frac. improved ($\uparrow$) |
|---|---|---|---|---|---|---|
| 5 | — | — | $0.29_{\pm 0.20}$ | $0.68_{\pm 0.08}$ | 0.80 | — |
| 5 | $\nabla r(x_t)$ | 1 | $0.29_{\pm 0.20}$ | $0.69_{\pm 0.08}$ | 0.80 | 0.02 |
| 5 | $\nabla r(x_t)$ | 10 | $0.29_{\pm 0.20}$ | $0.68_{\pm 0.08}$ | 0.80 | 0.05 |
| 5 | $\nabla r(x_t)$ | 100 | $0.29_{\pm 0.20}$ | $0.68_{\pm 0.08}$ | 0.80 | 0.17 |
| 5 | $\nabla r(x_t)$ | 1000 | $0.27_{\pm 0.19}$ | $0.66_{\pm 0.07}$ | 0.77 | 0.24 |
| 5 | $\nabla r(x_t)$ | 10000 | $0.14_{\pm 0.12}$ | $0.40_{\pm 0.08}$ | 0.62 | 0.03 |
| 5 | $\nabla r(\Phi^\theta_{t,1}(x_t))$ | 1 | $0.30_{\pm 0.20}$ | $0.69_{\pm 0.09}$ | 0.85 | 0.32 |
| 5 | $\nabla r(\Phi^\theta_{t,1}(x_t))$ | 10 | $0.33_{\pm 0.21}$ | $0.71_{\pm 0.06}$ | 0.81 | 0.65 |
| 5 | $\nabla r(\Phi^\theta_{t,1}(x_t))$ | 100 | $0.39_{\pm 0.22}$ | $0.76_{\pm 0.04}$ | 0.83 | 0.75 |
| 5 | $\nabla r(\Phi^\theta_{t,1}(x_t))$ | 1000 | $0.24_{\pm 0.17}$ | $0.59_{\pm 0.06}$ | 0.70 | 0.39 |
| 5 | $\nabla r(\Phi^\theta_{t,1}(x_t))$ | 10000 | $0.03_{\pm 0.03}$ | $0.09_{\pm 0.01}$ | 0.11 | 0.08 |
| 10 | — | — | $0.30_{\pm 0.20}$ | $0.70_{\pm 0.07}$ | 0.80 | — |
| 10 | $\nabla r(x_t)$ | 1 | $0.30_{\pm 0.20}$ | $0.70_{\pm 0.07}$ | 0.80 | 0.05 |
| 10 | $\nabla r(x_t)$ | 10 | $0.30_{\pm 0.20}$ | $0.70_{\pm 0.07}$ | 0.80 | 0.06 |
| 10 | $\nabla r(x_t)$ | 100 | $0.29_{\pm 0.20}$ | $0.70_{\pm 0.07}$ | 0.80 | 0.19 |
| 10 | $\nabla r(x_t)$ | 1000 | $0.27_{\pm 0.19}$ | $0.65_{\pm 0.08}$ | 0.79 | 0.23 |
| 10 | $\nabla r(x_t)$ | 10000 | $0.16_{\pm 0.13}$ | $0.41_{\pm 0.07}$ | 0.59 | 0.06 |
| 10 | $\nabla r(\Phi^\theta_{t,1}(x_t))$ | 1 | $0.30_{\pm 0.20}$ | $0.70_{\pm 0.07}$ | 0.80 | 0.36 |
| 10 | $\nabla r(\Phi^\theta_{t,1}(x_t))$ | 10 | $0.35_{\pm 0.21}$ | $0.74_{\pm 0.05}$ | 0.82 | 0.68 |
| 10 | $\nabla r(\Phi^\theta_{t,1}(x_t))$ | 100 | $0.45_{\pm 0.22}$ | $0.80_{\pm 0.03}$ | 0.84 | 0.82 |
| 10 | $\nabla r(\Phi^\theta_{t,1}(x_t))$ | 1000 | $0.45_{\pm 0.23}$ | $0.81_{\pm 0.04}$ | 0.86 | 0.75 |
| 10 | $\nabla r(\Phi^\theta_{t,1}(x_t))$ | 10000 | $0.03_{\pm 0.03}$ | $0.08_{\pm 0.01}$ | 0.09 | 0.05 |

*Table A5.* Ablation over the guidance scale $\lambda$ in Eq. (21) for guiding $\beta$-sheet generation in proteins. We report mean squared error (MSE) $\pm$ standard deviation across 100 samples of length 128 for two reward evaluations, the naive approach based on the current state ($\nabla r(x_t)$) and using $x_1$ look-ahead ($\nabla r(\Phi^\theta_{t,1}(x_t))$), as well as no guidance (—). We also ablate over time-dependent reward guidance, where $r_t(x) = t \cdot r(x)$ as in (Sabour et al., 2025a).

| NFE | $\zeta_t$ | Guidance scale ($\lambda$) | Time-dependent reward? | Mean (↑) | Top-10 mean (↑) | Max (↑) | Frac. improved (↑) |
|---|---|---|---|---|---|---|---|
| 5 | — | — | — | $0.18_{\pm 0.12}$ | $0.41_{\pm 0.06}$ | 0.51 | — |
| 5 | $\nabla r(x_t)$ | 1 | No | $0.18_{\pm 0.12}$ | $0.41_{\pm 0.05}$ | 0.51 | 0.01 |
| 5 | $\nabla r(x_t)$ | 10 | No | $0.18_{\pm 0.12}$ | $0.41_{\pm 0.05}$ | 0.51 | 0.01 |
| 5 | $\nabla r(x_t)$ | 100 | No | $0.18_{\pm 0.12}$ | $0.41_{\pm 0.05}$ | 0.51 | 0.04 |
| 5 | $\nabla r(x_t)$ | 1000 | No | $0.18_{\pm 0.12}$ | $0.41_{\pm 0.04}$ | 0.48 | 0.28 |
| 5 | $\nabla r(x_t)$ | 10000 | No | $0.16_{\pm 0.11}$ | $0.37_{\pm 0.04}$ | 0.44 | 0.31 |
| 5 | $\nabla r(x_t)$ | 10 | No | $0.18_{\pm 0.12}$ | $0.41_{\pm 0.05}$ | 0.51 | 0.01 |
| 5 | $\nabla r_t(x_t)$ | 10 | Yes | $0.18_{\pm 0.12}$ | $0.41_{\pm 0.06}$ | 0.51 | 0.01 |
| 5 | $\nabla r(x_t)$ | 100 | No | $0.18_{\pm 0.12}$ | $0.41_{\pm 0.05}$ | 0.51 | 0.04 |
| 5 | $\nabla r_t(x_t)$ | 100 | Yes | $0.18_{\pm 0.12}$ | $0.41_{\pm 0.06}$ | 0.51 | 0.01 |
| 5 | $\nabla r(x_t)$ | 1000 | No | $0.18_{\pm 0.12}$ | $0.41_{\pm 0.04}$ | 0.48 | 0.28 |
| 5 | $\nabla r_t(x_t)$ | 1000 | Yes | $0.18_{\pm 0.12}$ | $0.41_{\pm 0.06}$ | 0.51 | 0.15 |
| 5 | $\nabla r(\Phi^\theta_{t,1}(x_t))$ | 1 | No | $0.18_{\pm 0.12}$ | $0.41_{\pm 0.05}$ | 0.48 | 0.26 |
| 5 | $\nabla r(\Phi^\theta_{t,1}(x_t))$ | 10 | No | $0.19_{\pm 0.12}$ | $0.44_{\pm 0.04}$ | 0.52 | 0.42 |
| 5 | $\nabla r(\Phi^\theta_{t,1}(x_t))$ | 100 | No | $0.23_{\pm 0.12}$ | $0.46_{\pm 0.04}$ | 0.53 | 0.60 |
| 5 | $\nabla r(\Phi^\theta_{t,1}(x_t))$ | 1000 | No | $0.24_{\pm 0.14}$ | $0.49_{\pm 0.03}$ | 0.55 | 0.63 |
| 5 | $\nabla r(\Phi^\theta_{t,1}(x_t))$ | 10000 | No | $0.12_{\pm 0.12}$ | $0.40_{\pm 0.05}$ | 0.49 | 0.29 |
| 5 | $\nabla r(\Phi^\theta_{t,1}(x_t))$ | 10 | No | $0.19_{\pm 0.12}$ | $0.44_{\pm 0.04}$ | 0.52 | 0.42 |
| 5 | $\nabla r_t(\Phi^\theta_{t,1}(x_t))$ | 10 | Yes | $0.18_{\pm 0.12}$ | $0.41_{\pm 0.06}$ | 0.51 | 0.11 |
| 5 | $\nabla r(\Phi^\theta_{t,1}(x_t))$ | 100 | No | $0.23_{\pm 0.12}$ | $0.46_{\pm 0.04}$ | 0.53 | 0.60 |
| 5 | $\nabla r_t(\Phi^\theta_{t,1}(x_t))$ | 100 | Yes | $0.18_{\pm 0.12}$ | $0.42_{\pm 0.06}$ | 0.53 | 0.36 |
| 5 | $\nabla r(\Phi^\theta_{t,1}(x_t))$ | 1000 | No | $0.24_{\pm 0.14}$ | $0.49_{\pm 0.03}$ | 0.55 | 0.63 |
| 5 | $\nabla r_t(\Phi^\theta_{t,1}(x_t))$ | 1000 | Yes | $0.20_{\pm 0.13}$ | $0.44_{\pm 0.05}$ | 0.54 | 0.48 |
| 10 | — | — | — | $0.20_{\pm 0.13}$ | $0.45_{\pm 0.04}$ | 0.52 | — |
| 10 | $\nabla r(x_t)$ | 1 | No | $0.20_{\pm 0.13}$ | $0.45_{\pm 0.04}$ | 0.52 | 0.05 |
| 10 | $\nabla r(x_t)$ | 10 | No | $0.20_{\pm 0.13}$ | $0.45_{\pm 0.04}$ | 0.52 | 0.04 |
| 10 | $\nabla r(x_t)$ | 100 | No | $0.20_{\pm 0.13}$ | $0.44_{\pm 0.05}$ | 0.52 | 0.15 |
| 10 | $\nabla r(x_t)$ | 1000 | No | $0.20_{\pm 0.13}$ | $0.44_{\pm 0.05}$ | 0.56 | 0.30 |
| 10 | $\nabla r(x_t)$ | 10000 | No | $0.20_{\pm 0.13}$ | $0.45_{\pm 0.04}$ | 0.52 | 0.37 |
| 10 | $\nabla r(x_t)$ | 10 | No | $0.20_{\pm 0.13}$ | $0.45_{\pm 0.04}$ | 0.52 | 0.04 |
| 10 | $\nabla r_t(x_t)$ | 10 | Yes | $0.20_{\pm 0.13}$ | $0.45_{\pm 0.04}$ | 0.52 | 0.01 |
| 10 | $\nabla r(x_t)$ | 100 | No | $0.20_{\pm 0.13}$ | $0.44_{\pm 0.05}$ | 0.52 | 0.15 |
| 10 | $\nabla r_t(x_t)$ | 100 | Yes | $0.20_{\pm 0.13}$ | $0.45_{\pm 0.04}$ | 0.52 | 0.03 |
| 10 | $\nabla r(x_t)$ | 1000 | No | $0.20_{\pm 0.13}$ | $0.44_{\pm 0.05}$ | 0.56 | 0.30 |
| 10 | $\nabla r_t(x_t)$ | 1000 | Yes | $0.20_{\pm 0.13}$ | $0.45_{\pm 0.04}$ | 0.52 | 0.22 |
| 10 | $\nabla r(\Phi^\theta_{t,1}(x_t))$ | 1 | No | $0.20_{\pm 0.13}$ | $0.45_{\pm 0.04}$ | 0.52 | 0.24 |
| 10 | $\nabla r(\Phi^\theta_{t,1}(x_t))$ | 10 | No | $0.22_{\pm 0.13}$ | $0.45_{\pm 0.05}$ | 0.55 | 0.43 |
| 10 | $\nabla r(\Phi^\theta_{t,1}(x_t))$ | 100 | No | $0.25_{\pm 0.13}$ | $0.47_{\pm 0.03}$ | 0.52 | 0.65 |
| 10 | $\nabla r(\Phi^\theta_{t,1}(x_t))$ | 1000 | No | $0.26_{\pm 0.14}$ | $0.52_{\pm 0.05}$ | 0.61 | 0.61 |
| 10 | $\nabla r(\Phi^\theta_{t,1}(x_t))$ | 10000 | No | $0.27_{\pm 0.16}$ | $0.51_{\pm 0.02}$ | 0.55 | 0.59 |
| 10 | $\nabla r(\Phi^\theta_{t,1}(x_t))$ | 10 | No | $0.22_{\pm 0.13}$ | $0.45_{\pm 0.05}$ | 0.55 | 0.43 |
| 10 | $\nabla r_t(\Phi^\theta_{t,1}(x_t))$ | 10 | Yes | $0.20_{\pm 0.13}$ | $0.44_{\pm 0.04}$ | 0.52 | 0.19 |
| 10 | $\nabla r(\Phi^\theta_{t,1}(x_t))$ | 100 | No | $0.25_{\pm 0.13}$ | $0.47_{\pm 0.03}$ | 0.52 | 0.65 |
| 10 | $\nabla r_t(\Phi^\theta_{t,1}(x_t))$ | 100 | Yes | $0.21_{\pm 0.12}$ | $0.44_{\pm 0.04}$ | 0.52 | 0.36 |
| 10 | $\nabla r(\Phi^\theta_{t,1}(x_t))$ | 1000 | No | $0.26_{\pm 0.14}$ | $0.52_{\pm 0.05}$ | 0.61 | 0.61 |
| 10 | $\nabla r_t(\Phi^\theta_{t,1}(x_t))$ | 1000 | Yes | $0.23_{\pm 0.13}$ | $0.46_{\pm 0.03}$ | 0.53 | 0.58 |

# F. Experiment Details

## F.1. Promoter DNA Design

We model promoter DNA sequences as continuous, relaxed representations of length 1024, i.e., arrays in $\mathbb{R}^{1024 \times 4}$ with support restricted to the positive orthant, and interpret them as points on a product of spheres. We learn a time-dependent velocity field on this manifold using the average-velocity parameterization of Riemannian MeanFlow. For $n$-step evaluation, we discretize the time horizon $[0, 1]$ into $n$ uniform sub-intervals and apply flow-map inference sequentially over this grid. All experiments are conducted on a single NVIDIA RTX 3090 GPU. Code is available at `https://github.com/dongyeop3813/Riemannian-MeanFlow`.

**Dataset.** We use the FANTOM5 (Hon et al., 2017) dataset consisting of 100,000 transcription start sites (TSSs), following the same preprocessing and train/validation/test splits as prior work (Stark et al., 2024) (88,470/3,933/7,497). During training, we apply a random offset of up to $\pm 10$ bp around each TSS, while validation and test splits use fixed windows.

**Architecture.** We adopt the same 1D CNN backbone as in prior promoter models (Stark et al., 2024; Davis et al., 2024), consisting of an initial embedding layer followed by 20 residual convolutional blocks (kernel size 9) with progressively increasing dilation. The model conditions on two time variables by embedding both the absolute time $s$ and the time gap $(t - s)$ using Gaussian Fourier features, which are concatenated and injected into all time-conditioned layers. This doubles the time-embedding dimension and results in a modest parameter increase (13.27M $\rightarrow$ 14.65M). Depending on the parameterization, the output is either projected onto the tangent space ($v$-prediction) or mapped back to the manifold ($x_1$-prediction).

**Training and objectives.** Models are trained for 200 epochs with batch size 128 (138,400 steps total) using AdamW with learning rate $10^{-3}$, zero weight decay, and global-norm gradient clipping at 1.0. For evaluation, we maintain an exponential moving average (EMA) of the model parameters with decay 0.9999 to reduce the variance induced by differential objectives. For the Eulerian and Lagrangian RMF objectives, we apply adaptive loss weighting with exponent $p = 0.5$ and clip neural-network derivatives with threshold 100.0; for this task, Lagrangian RMF is trained without the cyclic consistency loss. For the semigroup RMF objective, we set $w_{\text{semigroup}} = 5.0$ and use adaptive loss weighting ($p = 0.5$), except for the $s = r = t$ (flow-matching) case where adaptive weighting is disabled. For $x_1$-prediction, we down-weight losses near $s = 1$ using a time clipping threshold $\epsilon = 0.1$. The best checkpoint is selected based on validation MSE between the true promoter signal and the signal predicted by the Sei model conditioned on generated sequences.

**Time sampling and interpolant.** We sample time points from a log-normal distribution with parameters $\mu = -0.4$ and $\sigma = 1.0$. For objectives requiring two time points, samples are ordered accordingly, with 75% boundary samples following Geng et al. (2025). For the semigroup objective, we sample $(s, t)$ as above and draw the intermediate time $r$ uniformly from $[s, t]$. We use a linear geodesic interpolant throughout, following prior work (Davis et al., 2024).

**Training cost and resources.** Table A6 summarizes the training resources used for the promoter DNA experiments. Across objectives, RMF uses the same single-GPU setup and peak memory as the reproduced Fisher FM baseline, with additional wall-clock time mainly coming from the extra derivative or consistency terms in the RMF objectives.

*Table A6.* Training resource summary for promoter DNA design. RMF uses the same single-GPU setup as Fisher FM, with comparable memory usage and modest additional wall-clock time depending on the objective.

| Model | Wall-clock time | Optimizer steps | Peak memory (GB) | # GPUs | GPU |
|---|---|---|---|---|---|
| Fisher FM | 20 hours | 140K | 20 | 1 | RTX 3090 |
| Eulerian RMF | 22 hours | 140K | 20 | 1 | RTX 3090 |
| Lagrangian RMF | 27 hours | 140K | 20 | 1 | RTX 3090 |
| Semigroup RMF | 44 hours | 140K | 20 | 1 | RTX 3090 |

## F.2. Protein Backbone Design

We provide code for the protein backbone experiments at `https://github.com/dongyeop3813/Protein-RMF`.

**SCOPe dataset.** The Structural Classification of Proteins—extended (SCOPe) (Chandonia et al., 2022) organizes protein structures from the Protein Data Bank (PDB) (Berman et al., 2000; Burley et al., 2021) into domains according to structural similarity and evolutionary relationships, providing curated coordinate files and hierarchical labels (e.g., class, fold,

superfamily, and family). Following Yim et al. (2023a), we use experimentally determined single-chain backbones with sequence lengths between 60 and 128 residues (3,938 examples) and evaluate on the protein monomer generation setting.

**Baselines.** We compare against three prior methods for backbone generation: GENIE (Lin & AlQuraishi, 2023), FrameDiff (Yim et al., 2023b), and FrameFlow (Yim et al., 2023a).

**Model architecture.** We adopt the FrameDiff architecture used in FrameFlow (Yim et al., 2023a), an SE(3)-equivariant model built around IPA blocks. We report the architecture variants (S/M/L) in Table A7. To adapt FrameDiff into a

*Table A7.* Architectural differences across IPA models.

| Model | Total # params | # of blocks | Node emb. dim | Edge emb. dim | Attn. heads |
|-------|---------------|-------------|---------------|---------------|-------------|
| RMF/S | 16.3 M | 6 | 256 | 128 | 4 |
| RMF/M | 91.7 M | 12 | 512 | 128 | 8 |
| RMF/L | 437.4 M | 16 | 768 | 384 | 12 |

flow-map parameterization, we condition each IPA normalization layer via adaptive layer-normalization (AdaLN) scaling applied to the node embeddings.

**Training setup.** For both translations and rotations, we use a linear noise schedule with a geodesic interpolant. Training minimizes a weighted combination of (i) flow-matching losses on translations and rotations, (ii) a semigroup consistency loss, and (iii) auxiliary geometric losses on backbone atom coordinates and local $C\alpha$–$C\alpha$ distances, as commonly used in protein backbone generation (Yim et al., 2023a;b; Bose et al., 2023). Auxiliary losses are applied only at late times ($t > 0.75$). The final objective uses adaptive loss reweighting with exponent $p = 0.5$. Unless otherwise stated, we set the semigroup loss weight to 1.0, the translation loss weight to 2.0, and use $x_1$-prediction down-weighting with $\epsilon = 0.1$. We maintain an exponential moving average (EMA) of model parameters with decay 0.9999 and use the EMA weights for evaluation. We train with AdamW (learning rate $10^{-4}$, no weight decay) for up to 1000 epochs (up to 1000K optimization steps), with gradient clipping and no learning-rate scheduling.

For flow-map training, we sample time points from the beta–uniform mixture proposed by Geffner et al. (2025b): $p(t) = 0.02\,\mathcal{U}[0, 1] + 0.98\,\mathcal{B}(1.9, 1.0)$. To form a time interval, we draw two time points i.i.d. from $p(t)$ and sort them to obtain $s < t$. For the intermediate time $r$, we use the midpoint $r = (s + t)/2$.

**Batching.** To accommodate variable-length proteins, we construct mini-batches dynamically by enforcing both a maximum batch size of 80 sequences and a global complexity constraint $\sum_i L_i^2 \leq 4 \times 10^5$, where $L_i$ is the length of sequence $i$. This yields stable memory usage across batches.

**Training cost and resources.** Table A8 summarizes the wall-clock time, optimizer steps, peak memory, number of GPUs, and GPU type used for the protein backbone experiments. RMF/S is trained on NVIDIA RTX 3090 GPUs, while RMF/M and RMF/L are trained on NVIDIA B200 GPUs. Compared with FrameFlow, RMF uses more training compute, but this cost is offset at inference time by achieving strong sample quality with substantially fewer NFEs.

*Table A8.* Training resource summary for protein backbone generation. RMF requires additional training compute compared with FrameFlow, but reduces inference cost by supporting high-quality generation with up to $10\times$ fewer NFEs.

| Model | Wall-clock time | Optimizer steps | Peak memory (GB) | # GPUs | GPU |
|-------|-----------------|-----------------|------------------|--------|-----|
| FrameFlow | 4 days | 500K | 21 | 4 | RTX 3090 |
| RMF/S | 8 days | 1.1M | 21 | 4 | RTX 3090 |
| RMF/M | 7 days | 1.1M | 70 | 2 | B200 |
| RMF/L | 8 days | 600K | 56 | 8 | B200 |

**Low-noise inference techniques with the flow map.** In protein backbone generation with flow matching or diffusion models, inference-time heuristics are commonly used to improve designability. For example, FrameFlow and FoldFlow (Yim et al., 2023a; Bose et al., 2023) apply velocity scaling to the rotational components during sampling. Similarly, Proteina (Geffner et al., 2025b) reports that reducing the magnitude of injected noise at each diffusion step can be beneficial. Motivated by these findings, we adopt the low-noise inference technique of Xie et al. (2025). Instead of strictly following the flow map

$\Phi_{s,t}^{\theta}$, we first recover an estimate of the data point and then re-introduce a controlled amount of noise:

$$x_t = t\hat{x}_1 + \eta(1-t)\epsilon, \quad \text{where } \hat{x}_1 = \Phi_{s,1}^{\theta}(x_s) \text{ and } \epsilon \sim \mathcal{N}(0, I). \tag{105}$$

When $\eta = 1$, this update resembles DDIM-style inference, whereas $\eta = 0$ yields a deterministic path with no added noise. In our experiments, we set $\eta = 0.45$ for rotations and $\eta = 1.0$ for translations; following Geffner et al. (2025b); Xie et al. (2025), $\eta = 0.45$ provided robust performance. Overall, this heuristic improves designability, with the largest gains in few-step sampling and for smaller model architectures.

### F.3. Reward Guidance Implementation

For reward-guided inference, we compute the Riemannian gradient by projecting the ambient Euclidean gradient onto the tangent space:

$$\nabla r(x_t) = \text{Proj}_{x_t}\left(\bar{\nabla} r(x_t)\right), \qquad \nabla r\left(\Phi_{t,1}^{\theta}(x_t)\right) = \text{Proj}_{x_t}\left(\bar{\nabla} r\left(\Phi_{t,1}^{\theta}(x_t)\right)\right). \tag{106}$$

where $\bar{\nabla}$ denotes the Euclidean gradient in the ambient space. We perform a grid search over the number of function evaluations (NFE) and the guidance scale.

# G. Additional Results

## G.1. Empirical Evidence on Training Stabilization Techniques

In this section, we provide empirical evidence for the stabilization techniques introduced in Sec. 3.3. While the effectiveness of adaptive loss weighting was discussed in Sec. 4.1, here we focus on two other critical factors: time-sampling distributions and time-derivative control.

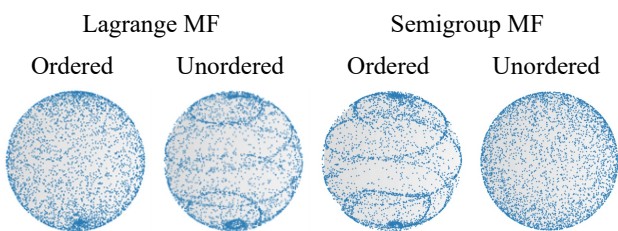

Lagrange MF      Semigroup MF

Ordered    Unordered     Ordered    Unordered

*Figure A2.* Effect of time sampling order. Comparison of Lagrangian (left) and Semigroup (right) objectives. Lagrangian RMF requires unordered intervals for convergence, whereas Semigroup RMF becomes unstable when trained with unordered intervals.

*Figure A3.* Training instability of Eulerian and Lagrangian RMF: (Left) Variance of the Eulerian RMF regression target. (Right) Reducing the Fourier frequency to $\omega = 0.02$ lowers the regression-target variance.

**Effect of time sampling distributions.** As illustrated in Fig. A2, the choice of time-sampling distribution is critical for the the stability and convergence of the learned flow map. While Eulerian MF remains largely insensitive to the temporal ordering of samples—typically using ordered pairs $s \leq t$, which cover half of the unit square—the other two objectives exhibit strict and contrasting requirements:

1. **Lagrangian MF** necessitates unordered time intervals. Training exclusively with ordered pairs ($s < t$) fails to capture the full dynamics, resulting in failure to converge to a valid flow map. This suggests that the Lagrangian formulation relies on the bidirectional information provided by sampling both $s < t$ and $t < s$ to regularize the path.

2. **Semigroup MF** shows the opposite behavior, where stability is tied to the sequential structure of the triplets. When trained with unordered intervals, the objective becomes highly unstable. The inclusion of an intermediate time $r$ ($s < r < t$) reinforces the compositionality of the flow, and departing from this ordered structure leads to significant performance degradation observed in our sphere visualizations.

**Time-derivative control.** Differential objectives, such as the Eulerian and Lagrangian formulations, are particularly susceptible to instability arising from time-derivative terms. Fig. A3 (left) elucidates this phenomenon: the norm of the regression target explodes at certain time steps (highlighted by the red circle). This instability stems from the uncontrolled magnitude of the neural network's time derivative, $D_s u_{s,t}^{\theta}$, which significantly destabilizes the optimization process.

To mitigate this, we bound the derivative magnitude by adjusting the Fourier time embeddings. Since the derivative of a periodic embedding $\frac{d}{dt} \sin(\omega t) = \omega \cos(\omega t)$ scales linearly with the frequency $\omega$, using high frequencies (e.g., $\omega = 30$) leads to high-variance regression targets. By adopting a lower frequency (e.g., $\omega = 0.02$), we effectively stabilize the training process. As shown in Fig. A3 (right), this modification drastically reduces the variance of the regression target, a finding that mirrors observations in consistency model training (Lu & Song, 2024) and proves essential for robust flow map learning.

**Ablation on training stabilization techniques.** We ablate the stabilization techniques on the promoter DNA task using the Eulerian RMF objective with $x_1$-prediction and one-step sampling. Table A9 removes the proposed components cumulatively from the full training recipe, complementing the identity and parameterization ablations in Figs. 2 and 3. Removing the additional weighting near $s = 1$ causes only a small degradation, increasing MSE from $0.030$ to $0.034$ while preserving the $k$-mer correlation at $0.90$. In contrast, removing adaptive loss weighting has a larger effect, worsening MSE to $0.041$ and reducing the $k$-mer correlation to $0.84$. The largest degradation occurs when time-embedding frequency control is removed: MSE increases to $0.058$ and the $k$-mer correlation collapses to $-0.12$. These results are consistent with the analysis above: differential objectives produce high-variance regression targets, and both adaptive loss weighting and frequency control are needed to control this variance in high-dimensional sequence generation.

**Hyperparameter sensitivity analysis on training stabilization.** We further evaluate whether the method is sensitive to the precise hyperparameter values used by these stabilization mechanisms. As shown in Table A10, all tested settings converge successfully across a range of adaptive weighting powers $p$, time-embedding frequencies $\omega$, and clipping thresholds $\epsilon$, as long as the corresponding stabilization mechanism is present. Varying $p$ from the default 0.5 to 0.25 or 0.75 yields similar MSE values (0.032–0.033) and maintains high $k$-mer correlation. Likewise, low-frequency time embeddings with $\omega \in \{0.01, 0.1, 0.2\}$ all match or slightly improve the default setting. The clipping ablation shows a mild but consistent benefit from explicit clipping: removing clipping keeps the MSE competitive but reduces the $k$-mer correlation from 0.90 to 0.74. Importantly, we use the same defaults, $p = 0.5$, $\omega = 0.02$, and $\epsilon = 0.1$, across the spherical helix, DNA, and protein experiments without task-specific tuning. Overall, RMF training is more sensitive to the presence of the stabilization principles than to fine tuning their exact hyperparameters.

*Table A9.* Ablation study of stabilization techniques for Eulerian RMF on the DNA promoter dataset (NFE=1). Each row additionally removes one component from the row above. Both adaptive loss weighting and frequency control are essential for stable training.

| Configuration | MSE ($\downarrow$) | $k$-mer corr. ($\uparrow$) |
|---|---|---|
| Full | **0.030** | **0.90** |
| — weighting near $s{=}1$ | 0.034 | 0.90 |
| — adaptive loss weighting | 0.041 | 0.84 |
| — time-emb. frequency control | 0.058 | $-0.12$ |

*Table A10.* Sensitivity analysis of training stabilization hyperparameters for Eulerian RMF on the DNA promoter dataset. We vary $p \in \{0.25, 0.5, 0.75\}$, $\omega \in \{0.01, 0.02, 0.1, 0.2\}$, and $\epsilon \in \{0.05, 0.1, \text{none}\}$. Given that each stabilization technique is enabled, performance is robust to the specific hyperparameter values.

| Configuration | MSE ($\downarrow$) | $k$-mer corr. ($\uparrow$) |
|---|---|---|
| Default ($p{=}0.5$, $\omega{=}0.02$, $\epsilon{=}0.1$) | **0.030** | 0.90 |
| Varying $p$ (Adaptive loss weight) | | |
| $\quad p{=}0.25$ | 0.032 | 0.87 |
| $\quad p{=}0.75$ | 0.033 | 0.85 |
| Varying $\omega$ (Time-embedding frequency) | | |
| $\quad \omega{=}0.01$ | 0.029 | 0.92 |
| $\quad \omega{=}0.1$ | 0.031 | 0.92 |
| $\quad \omega{=}0.2$ | 0.031 | 0.93 |
| Varying $\epsilon$ (Clipping threshold) | | |
| $\quad \epsilon{=}0.05$ | 0.032 | 0.92 |
| $\quad$ no-clipping | 0.034 | 0.74 |

## G.2. Effect of Cycle-Consistency Regularizer on Lagrangian Objective

In this section, we study when the cycle-consistency regularizer is useful for the Lagrangian RMF objective. As discussed in Sec. 3.1, the Lagrangian objective evaluates the regression loss at a model-predicted input $\hat{x}_s = \Phi_{t,s}^\theta(x_t)$. Therefore, if the learned forward and backward flow maps are not approximately inverse to each other, errors in the predicted input can propagate directly into the regression target. The cycle-consistency regularizer is designed to mitigate this issue by encouraging weak invertibility of the learned flow map:

$$\mathcal{L}_{\text{cyc}}(\theta) = \mathbb{E}_{x_t, s, t}\left[d_g(\Phi_{s,t}^\theta(\Phi_{t,s}^\theta(x_t)), x_t)^2\right]. \tag{107}$$

We compare Lagrangian RMF trained *with* and *without* this regularizer while keeping all other training configurations fixed. We first consider the spherical helix benchmark introduced in Sec. 4.1, where the data lie on a low-dimensional manifold embedded in a high-dimensional ambient space. As shown in Fig. A5, cycle consistency improves the learned transport in this setting: without the regularizer, accumulated invertibility errors lead to visibly poorer samples, whereas the regularized model better preserves the target helical structure. This suggests that cycle consistency can be important when small flow-map errors directly affect the geometry of the generated samples.

We then repeat the same ablation on the promoter DNA generation task. In contrast to the spherical helix result, Table A11 shows that removing the cycle-consistency regularizer does not degrade 1-NFE generation quality on DNA, yielding comparable MSE and $k$-mer correlation to the regularized model. We attribute this robustness to the final $\arg\max$ discretization step, which maps continuous model outputs to discrete nucleotide sequences and can absorb small continuous-space invertibility errors before evaluation.

Taken together, these results indicate that the cycle-consistency regularizer is a task-dependent stabilization mechanism rather than a universally required component. It is beneficial when generation quality is sensitive to geometric errors in the continuous flow map, as in the high-dimensional spherical helix benchmark. However, for tasks such as DNA sequence generation, where the output is ultimately discretized and the evaluation metrics are less sensitive to small continuous deviations, the regularizer may be unnecessary.

*Table A11.* 1-NFE performance of Lagrangian RMF on promoter DNA generation with and without cycle-consistency regularization. Removing cycle-consistency does not degrade performance, which we attribute to the argmax discretization applied after generation absorbing minor invertibility errors.

| Method | MSE ($\downarrow$) | $k$-mer corr. ($\uparrow$) |
|---|---|---|
| With cycle-consistency | 0.028 | 0.85 |
| Without cycle-consistency | 0.027 | 0.88 |

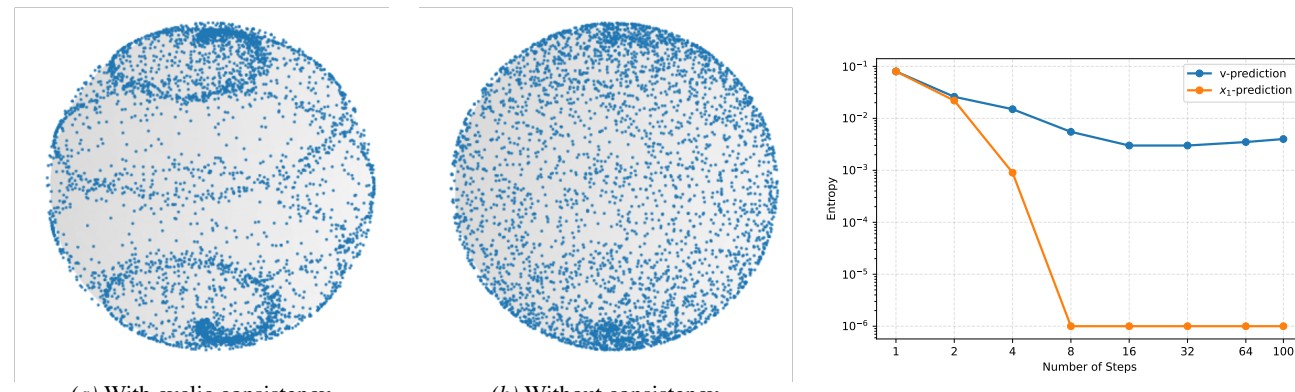

*(a)* With cyclic consistency      *(b)* Without consistency

*Figure A5.* Effect of cyclic consistency regularization. Samples generated by Lagrangian RMF with and without the proposed cycle-consistency regularizer on a high-dimensional spherical helix dataset ($D = 512$).

*Figure A6.* Inference steps vs. entropy. $x_1$-prediction achieves a significantly lower entropy compared to $v$-prediction.

### G.3. Parametrization Comparison in Promoter DNA Design

While $x_1$- and $v$-prediction achieve comparable standard metrics (e.g., MSE and $k$-mer correlation), they induce markedly different distributional sharpness. As shown in Fig. A6, $x_1$-prediction attains near-zero entropy (on the order of $10^{-6}$), whereas $v$-prediction saturates at a substantially higher level (approximately $3 \times 10^{-3}$), indicating that $x_1$-prediction produces outputs closer to one-hot distributions.

In discrete sequence design, this difference is largely collapsed by the final $\arg\max$ discretization step and can therefore be hidden by downstream metrics. However, such increased sharpness may be beneficial in settings where small distributional differences directly affect geometry or structure, for example, protein backbone modeling.

## G.4. Additional FoldFlow Comparison on Protein Backbone Generation

We additionally compare against FoldFlow (Bose et al., 2023). FoldFlow was not included in the main comparison because its released checkpoint was trained on PDB, whereas our protein backbone experiments use the SCOPe setting. To make the comparison more complete, Table A12 reports results for both the released FoldFlow PDB checkpoint and a FoldFlow model retrained on SCOPe, evaluated using the same self-consistency pipeline as the main protein backbone experiments.

The results show a clear distinction between many-step quality and few-step efficiency. At 100 NFE, FoldFlow achieves high designability: the PDB checkpoint attains the highest designable fraction and lowest scRMSD in this expanded comparison, while the SCOPe-retrained model is comparable to FrameFlow and RMF in designability. However, this performance does not transfer to the few-step regime. At 10 NFE and 5 NFE, both FoldFlow checkpoints produce no designable samples under the scRMSD < 2Å criterion, whereas RMF remains designable for 87% of samples at 10 NFE and 82% at 5 NFE. RMF also retains nontrivial one-step generation quality, with 35% designability at 1 NFE.

We further examine the secondary-structure distribution of the generated backbones in Table A13. Although FoldFlow attains strong self-consistency metrics at 100 NFE, both checkpoints exhibit a pronounced secondary-structure bias, generating approximately 90% $\alpha$-helices and essentially no $\beta$-sheets. This is reflected in the low MaxCluster diversity scores in Table A12. By contrast, RMF and FrameFlow produce secondary-structure compositions much closer to the PDB reference distribution.

*Table A12.* Protein backbone generation results. Rows are grouped by inference regime (NFE). We highlight in **bold** the best designability (<2Å) within each regime, our primary metric. We mark not applicable (N/A) when no designable samples are generated or when metrics are not reported in prior work.

| Model | NFE | Designability | | Diversity | | Novelty |
|---|---|---|---|---|---|---|
| | | <2Å ($\uparrow$) | scRMSD ($\downarrow$) | Max. Cluster ($\uparrow$) | Pairwise scTM ($\downarrow$) | Max. scTM ($\downarrow$) |
| Many-step regime (NFE $\geq$ 100) | | | | | | |
| FrameDiff | 100 | 0.74 | 1.78 | 1.74 | 0.34 | 0.51 |
| FrameFlow | 100 | 0.93 | 1.16 | 0.41 | 0.30 | 0.77 |
| FoldFlow (PDB) | 100 | **0.98** | **0.64** | 0.16 | 0.41 | 0.39 |
| FoldFlow (SCOPE) | 100 | 0.94 | 1.04 | 0.19 | 0.43 | 0.64 |
| **RMF (Ours)** | 100 | 0.94 | 1.01 | 0.55 | 0.27 | 0.89 |
| Moderate regime (10 $\leq$ NFE < 100) | | | | | | |
| FrameDiff | 10 | 0.47 | 3.32 | 0.42 | 0.28 | 0.52 |
| FrameFlow | 10 | 0.61 | 2.34 | 0.54 | 0.26 | 0.67 |
| FoldFlow (PDB) | 10 | 0.00 | 8.19 | N/A | N/A | N/A |
| FoldFlow (SCOPE) | 10 | 0.00 | 10.45 | N/A | N/A | N/A |
| **RMF (Ours)** | 10 | **0.87** | **1.25** | 0.55 | 0.27 | 0.87 |
| Few-step regime (NFE < 10) | | | | | | |
| FrameFlow | 5 | 0.04 | 6.53 | 0.68 | 0.22 | 0.74 |
| FrameDiff | 5 | 0.09 | 6.19 | 0.54 | 0.24 | 0.96 |
| FoldFlow (PDB) | 5 | 0.00 | 57.33 | N/A | N/A | N/A |
| FoldFlow (SCOPE) | 5 | 0.00 | N/A | N/A | N/A | N/A |
| **RMF (Ours)** | 5 | **0.82** | **1.54** | 0.54 | 0.27 | 0.85 |
| FrameFlow | 2 | 0.00 | N/A | N/A | N/A | N/A |
| **RMF (Ours)** | 1 | **0.35** | 3.33 | 0.60 | 0.24 | 0.76 |

*Table A13.* Secondary structure composition of generated protein backbones compared to the PDB reference distribution. FoldFlow (both PDB and SCOPE checkpoints) generates approximately 90% $\alpha$-helices and 0% $\beta$-sheets, indicating severe mode collapse. FrameFlow and RMF produce secondary structure distributions close to the PDB reference.

| Secondary structure | FoldFlow | | FrameFlow | RMF | PDB ref. |
|---|---|---|---|---|---|
| | PDB | SCOPE | | | |
| $\alpha$-helix % | 90 | 89 | 33 | 32 | 34 |
| $\beta$-sheet % | 0 | 0 | 27 | 28 | 26 |
| Coil % | 10 | 10 | 40 | 40 | 40 |

## G.5. Ablations on Protein Backbone Experiments

**Adaptive loss weighting.** We study the effect of adaptive loss weighting in the protein backbone task, keeping all model architectures, training schedules, and optimization hyperparameters fixed. All results in Fig. A7a are reported for the L model without inference-time techniques. In Fig. A7a, we compare $p = 0.0$ (no weighting) and $p = 0.5$. Across inference step counts, $p = 0.5$ consistently yields lower scRMSD, indicating that adaptive weighting improves generation quality.

**Model scaling without inference tricks.** To isolate the contribution of inference-time techniques and examine the effect of model scaling, we repeat the protein backbone evaluation without applying any inference-time heuristics and report the results in Fig. A7b.

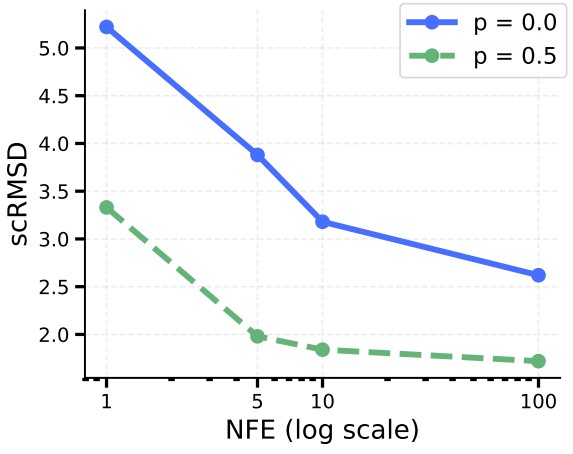

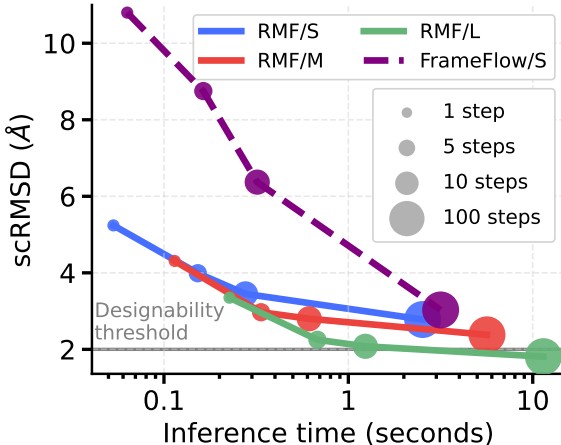

*(a)* Effect of adaptive loss weighting on generation quality.

*(b)* Generation quality vs. inference time without inference-time techniques.

*Figure A7.* **Ablation on protein backbone generation.** (a) Adaptive loss weighting improves scRMSD across inference steps. (b) Removing inference-time heuristics reveals Pareto improvements from model scaling.

Without inference tricks (e.g., inference-time rotation velocity scaling), the FrameFlow baseline exhibits a substantial performance drop, achieving only 40% designability with an scRMSD of 3.03 even at 100 inference steps. Under the same setting and model size, our flow-map-based model shows clear improvements at few-step regimes (1, 5, and 10 steps), but its multi-step performance remains limited without scaling.

This observation suggests that strong multi-step performance is crucial for effective few-step generation. Motivated by this, we scale the model to the L variant, which significantly improves multi-step sampling, achieving an scRMSD of 1.59 and 77% designability at 100 steps. As the multi-step performance improves, few-step generation quality also improves accordingly, reaching scRMSD values of 1.98 and 1.84 at 5 and 10 steps, respectively. Overall, evaluating all methods without inference-time tricks makes the impact of model scaling more apparent. As shown in Fig. A7b, increasing model capacity yields consistent improvements in both generation quality and designability across inference budgets, yielding a clear Pareto improvement. A promising direction for future work is to incorporate the effect of inference-time heuristics directly into the learned dynamics (or training objective), so that their benefits do not rely on sampling-time adjustments. Notably, in the Euclidean setting, MeanFlow (Geng et al., 2025) has explored integrating classifier guidance into the model dynamics; analogously, one could aim to internalize common inference-time tricks and potentially accelerate the resulting guided dynamics.

## G.6. Qualitative Results on Protein Backbone Generation

In Fig. A8, we visualize the protein backbones generated by our model across different inference steps and protein lengths. All samples were generated using the inference trick with a eta value of 0.45.

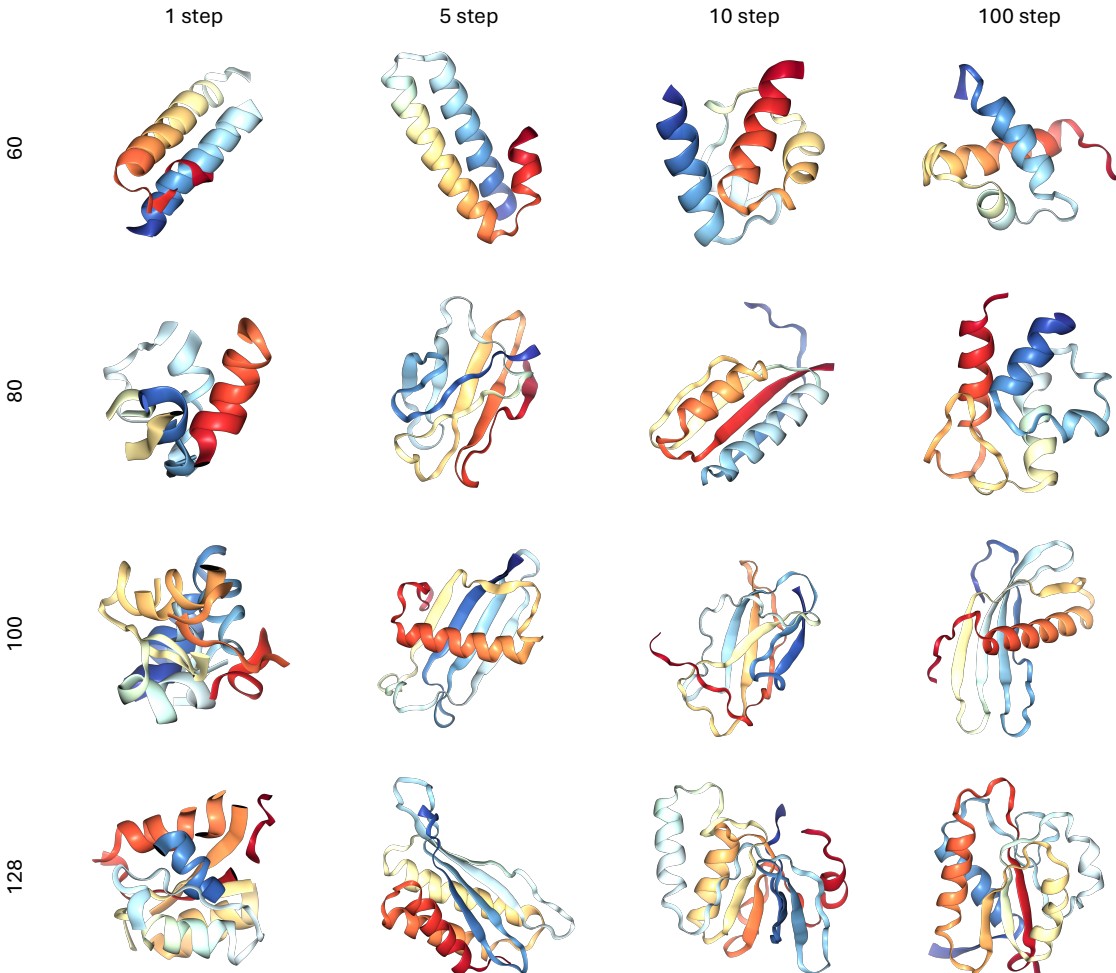

*Figure A8.* Visualization of generated protein backbones. We visualize the generated protein backbone from our model for steps of {1, 5, 10, 100} and sequence length {60, 80, 100, 128}.

