# OpenReview forum: "Riemannian MeanFlow"
_ICML.cc/2026/Conference — ICML 2026 regular_

### Official Review · Reviewer_qnnr · 2026-03-02

**Soundness:** 4
**Presentation:** 3
**Significance:** 3
**Originality:** 3
**Overall Recommendation:** 5
**Confidence:** 4

**Summary:**

The paper present an extension to the mean flow algorithm, which is originally defined for Euclidean spaces, i.e., flat spaces, to the case of more general manifolds, specifically those admitting a Riemannian structure. Such a structure is important since it allows to define what a flow is, and consequently, what a "constant velocity" flow on the manifold is supposed to be.

The authors derive three equivalent descriptions for the mean flow, which leads to three parametrization for the loss. In the experiments they show good results on some non-trivial datasets, like DNA promoter design and protein backbone generation.

**Compliance With Llm Reviewing Policy:**

Affirmed.

**Key Questions For Authors:**

- Is there any special property when the manifold is a Lie group? For instance, in this case, the mean velocity is an element of the Lie algebra, and I can imagine that the mean flow identity can be formulated generally through the property of the commutator of the Lie algebra.
-  Can this formalism be extended to the case of generative models defined for spaces of representation of Lie groups, rather than on the Lie group itself? ( as in Bertolini et al., "Diffusion Generative Modeling on Lie Group Representations")
- How do the authors explain the very large difference in performance between the different parametrizations (e.g., in Figure 2), even though they are supposed to be equivalent?

**Limitations:**

yes

**Strengths And Weaknesses:**

Strenghts:

- The paper is well structured and the problem is well-defined and motivated.
- The claims are clearly formulated and the theoretical results are intuitively introduced, in addition to the formal proofs. I also like the fact that the main text has sketches of the proofs.
- The authors performs ablation between the different formulations of the mean flow and different parametrization of the network on toy datasets.

Weaknesses:
- Some connections to the most important case of $\mathcal{M}=G$ a Lie group is missing in my opinion.
- The paper would benefit quite substantially if the authors could carry out an explicit example through the formalism of the paper. For example, in the case $\mathcal{M}=SO(2)$ or $\mathcal{M}=SO(3)$ it should be possible to explicitly write down all the definitions and prove explicitly that they are equivalent.

---

> ### Author Rebuttal · Authors · 2026-03-31
>
> We thank the reviewer for the thorough and constructive review. We are glad that the reviewer finds our theoretical framework well-structured, the proof sketches helpful, and the ablation studies informative. We address the weaknesses and questions below.
>
> ---
> **W1/Q1. Lack of connections to the Lie group. Can the identities be formulated through commutators?**
>
> The reviewer's intuition is correct. When $\mathcal{M}$ is a Lie group, the mean velocity admits a natural definition as a Lie algebra element: $u_{s,t}(x_s) = \frac{1}{t-s}\log(x_s^{-1}x_t)$, where $\log$ is the Lie group logarithm rather than the Riemannian log map. In this case, the differential of the logarithm can be expanded in terms of Lie bracket (commutator) series (e.g., the BCH formula). However, our proofs of the core identities do not require such expansion: since the differential of the log map is treated as a known analytic map, the identities follow directly without invoking the commutator structure. That said, investigating whether Lie group structure yields additional simplifications or tighter characterizations is an interesting direction for future work.
>
> **W2. An explicit worked example (e.g., SO(2) or SO(3)) through the full formalism would strengthen the paper.**
>
> Thank you for this constructive suggestion. We agree that a self-contained worked example with explicit closed-form expressions would aid readability. We will provide a self-contained walkthrough on $\textrm{SO}(3)$ covering: (1) the explicit forms of the exponential map, logarithm map, and geodesic interpolant using the Rodrigues formula; (2) the covariant derivative of the log map in closed form; (3) each of the three MeanFlow identities (Eulerian, Lagrangian, Semigroup) written out explicitly with these closed-form expressions; and (4) how these connect to the training loss computation in practice.
>
> **Q2. Can RMF extend to generative models on spaces of Lie group representations (Bertolini et al.)?**
>
> When the representation space itself carries a manifold structure, RMF applies directly as presented. The approach of Bertolini et al. is particularly interesting: rather than performing diffusion on the group manifold, they remain in Euclidean space and guide the dynamics using the differential of the representation map. In this Euclidean setting, the special case of our framework—Euclidean MeanFlow (Geng et al., 2025)—is directly applicable, potentially enabling few-step generation within their representation-guided diffusion framework. We consider this a promising direction for future work.
>
> **Q3. Why is the performance gap between equivalent parameterizations (Figure 2) so large?**
>
> The three parameterizations are equivalent in the sense that they characterize the same ground-truth average velocity, but the difficulty of the resulting optimization problem can differ substantially.
>
> **$x_1$-prediction.** In our setting, data lies on a two-dimensional manifold embedded in a high-dimensional ambient space. With $x_1$-prediction, the network only needs to capture this low-dimensional structure (by the manifold hypothesis), so even a small network ($h=256$) maintains good performance as $D$ increases to $2048$.
>
> **$v$-prediction.** The network must produce a full $D$-dimensional velocity vector whose norm and variance grow with the ambient dimension due to injected noise, making it intractable to learn with the same network capacity.
>
> **$x_t$-prediction.** This reduces to regression on the derivative of the neural network, leading to very unstable training dynamics in practice (as discussed in Sec. 4.1), making it infeasible to train with only the flow-matching objective.
>
> This is consistent with recent Euclidean findings: Li & He (2025) show that predicting clean data simplifies learning, and Lu et al. (2026) demonstrate that $x_{1}$-prediction enables one-step pixel-space generation where $v$-prediction cannot.
>
> [1] Li & He. "Back to Basics: Let Denoising Generative Models Denoise."
>
> [2] Lu et al. "One-step Latent-free Image Generation with Pixel Mean Flows."

---

> > ### Author Rebuttal · Reviewer_qnnr · 2026-04-02
> >
> > Thank you for the reply. I do not have any further questions.

---

> > > ### Author Response · Authors · 2026-04-03
> > >
> > > Thank you for the supportive review and valuable feedback. We're glad the concerns have been fully addressed.

---

### Official Review · Reviewer_R1vx · 2026-03-11

**Soundness:** 3
**Presentation:** 3
**Significance:** 3
**Originality:** 3
**Overall Recommendation:** 4
**Confidence:** 3

**Summary:**

The authors attempt to present a central theme of efficient, few-step generative modeling on Riemannian manifolds, addressing the inference bottleneck of standard diffusion and flow matching methods which typically require numerical integration. Broadly speaking, the paper evaluates a key concept: learning flow maps directly on manifolds (Riemannian MeanFlow, RMF) to transport prior distributions to data distributions in as few as one step.

The paper makes several specific contributions:
* **Theoretical Identities:** It derives three equivalent characterizations of the manifold average velocity—Eulerian, Lagrangian, and Semigroup—and formulates corresponding training objectives.
* **Parameterization & Stabilization:** Through ablation studies, it identifies that combining the Semigroup objective with an $x_1$-prediction parameterization (predicting the endpoint on the manifold rather than the tangent vector) yields the best stability and performance in high dimensions. It also introduces adaptive loss weighting and frequency control for time embeddings.
* **Applications:** The method is evaluated on high-dimensional biological tasks, specifically Promoter DNA design (product of simplices) and Protein Backbone generation ($SE(3)$). The authors demonstrate that RMF matches the performance of multi-step baselines (like FrameFlow) with significantly fewer function evaluations (NFE).
* **Reward Guidance:** The paper proposes a method for reward-guided inference using look-ahead prediction with the learned flow map.

**Compliance With Llm Reviewing Policy:**

Affirmed.

**Final Justification:**

Thanks for the rebuttal. You've cleared up my questions regarding the baseline comparisons and Table 4 metrics, so I've raised my score to a Weak Accept.

**Key Questions For Authors:**

1. **Missing Baselines:** In the protein backbone design experiments (Sec 4.3), you compare against FrameDiff and FrameFlow. Could you explain why FoldFlow was not included as a baseline, given it is cited as a highly relevant architecture? Including this comparison would significantly strengthen the evaluation of $SE(3)$ generative capabilities.
2. **Table 4 Interpretation:** In Table 4 ($\alpha$-helix, 10 NFE), the naive guidance results in the exact same Mean (0.30) and Max (0.80) scores as the unguided baseline, yet you report a "Frac. improved" of 0.27. Could you clarify how this fraction is calculated? Does the unchanged mean imply that the guidance was ineffective for the distribution as a whole?
3. **Cycle-Consistency:** You mention that the cycle-consistency loss ($\mathcal{L}_{cycle}$) is an "optional regularizer" for Lagrangian RMF. In what specific high-dimensional scenarios is this strictly necessary for convergence? Does the performance gap shown in Figure A5 (Spherical Helix) persist in the protein or DNA tasks if this regularizer is entirely removed?
4. **Inference-Time Noise Heuristics:** In Figure 5(b) and Appendix F.2, you note that injecting a controlled amount of noise during inference ($\eta$ between 0.25 and 0.45) yields the best outcomes for designability and diversity. Since the primary theoretical appeal of flow map models is taking direct, deterministic leaps along the trajectory, why does re-injecting noise at inference time remain so critical? Does this suggest the flow maps are still struggling to learn smooth, noise-free trajectories in high-dimensional spaces?

**Limitations:**

**Yes.**

**Strengths And Weaknesses:**

**Strengths:**
* **Originality & Theoretical Soundness:** The derivation of the three identities (Eulerian, Lagrangian, Semigroup) on Riemannian manifolds is mathematically grounded and insightful. The authors provide a rigorous extension of Euclidean concepts to curved spaces, handling complications like covariant derivatives and parallel transport carefully. The connection drawn between the proposed Eulerian RMF and the concurrent Generalized Flow Map (GFM) is valuable, showing how RMF avoids specific Jacobian-vector product (JVP) overheads.
* **Computational Efficiency vs. GFM:** The authors effectively demonstrate in Appendix D that their Eulerian RMF formulation uses roughly half the memory of GFM (9.5 GB vs 17.7 GB) and is about 2.6x faster per training iteration (0.15s vs 0.40s). This strongly supports their claim that bypassing JVPs substantially improves scalability and optimization stability in high-dimensional settings.
* **Practical Significance:** The focus on reducing inference NFE is highly relevant for scientific domains like drug discovery, where sampling speed is critical. The empirical findings regarding parameterization are useful for the community; specifically, the observation that $x_1$-prediction scales better to high dimensions ($D=2048$) than vector-field prediction is a strong practical insight.
* **Presentation:** The paper is generally well-organized. The progression from geometric definitions to objectives and then to experiments is logical, and Appendix A provides a helpful tutorial on Riemannian geometry.

**Weaknesses:**
* **Experimental Baselines (Protein Design):** While the paper compares against FrameDiff and FrameFlow, it lacks comparisons to other relevant $SE(3)$ flow matching baselines, specifically FoldFlow or its successors. Given that the paper cites FoldFlow as a related architecture that naturally predicts frames, excluding it from the performance benchmarks weakens the claim of state-of-the-art performance. A broader comparison would better contextualize RMF's standing in the protein design landscape.
* **Clarity on Metrics (Table 4):** The results in Table 4 regarding reward-guided inference are confusing. For the $\alpha$-helix task at 10 NFE, the naive guidance row ($Proj_{x_t}(\nabla r(x_t))$) reports a Mean of 0.30 and Max of 0.80, which are identical to the "No guidance" baseline. However, the "Frac. improved" column reports 0.27. It is unclear how 27% of the samples improved if the mean and maximum statistics remained completely stagnant. The "Frac. improved" metric needs a precise definition to be interpretable.
* **Complexity:** The method introduces several stabilization hyperparameters (adaptive loss weighting $p$, time-embedding frequency $\omega$, clipping $\epsilon$). While these are empirically justified, they suggest that training RMF might be somewhat brittle compared to standard flow matching, requiring careful tuning.

---

> ### Author Rebuttal · Authors · 2026-03-31
>
> We thank the reviewer for the thorough evaluation. We address each weakness and question below.
>
> **W1, Q1. Missing comparison with FoldFlow baseline.**
>
> We appreciate the reviewer for raising this important point, which strengthens our experimental evaluation. FoldFlow was not included in our original comparison because it was trained on a different dataset (PDB) than our setting (SCOPE), making direct comparison unfair. To address this concern, we evaluated both the released FoldFlow PDB checkpoint and a retrained FoldFlow on SCOPE using our evaluation pipeline (see [Table A11](https://imgur.com/u41EsaC), [Table A12](https://imgur.com/xFTwtBv)).
>
> In the few-step regime, FoldFlow's performance degrades substantially, whereas RMF maintains over 80% designability at both 10 and 5 NFE. In the many-step regime, FoldFlow achieves high designability but at the cost of severe mode collapse — its pairwise scTM is 0.41 (vs. 0.27 for RMF), and it generates nearly no $\beta$-sheet-containing proteins, unlike FrameFlow and RMF which produce ~28% $\beta$-sheets close to the training distribution.
>
> **W2, Q2. Clarity on metrics (Table 4)**
>
> We thank the reviewer for this insightful question. The "fraction of improved samples" is the proportion of samples for which the target secondary structure composition (measured by DSSP) increased compared to the unguided baseline from the same initial noise. In the $\alpha$-helix task at 10 NFE, naive guidance improves 27% of samples, yet the mean and max remain unchanged because a similar proportion degraded. As the reviewer correctly noted, this confirms that naive guidance is ineffective for the distribution as a whole. To make this clearer, we report the fraction of degraded samples alongside (see [Table A13](https://imgur.com/Vx9FK3h)). Our reward look-ahead guidance achieves substantially higher improvement rates with low degradation rates (0.75 vs. 0.22 for α-helix at 10 NFE), demonstrating that evaluating the reward on the predicted endpoint provides a far more reliable gradient signal.
>
> **W3. Training RMF may be brittle, requiring careful tuning of several stabilization hyperparameters.**
>
> We agree that the stabilization techniques are essential for stable training. However, we would like to clarify that the hyperparameters within these techniques do not require careful tuning. To validate this, we tested Eulerian RMF on the DNA dataset across various $p$, $\omega$, and $\epsilon$ values — all settings converged successfully (see [Table A14](https://imgur.com/xdA0a2B)).
>
> Furthermore, the same defaults ($p=0.5$, $\omega=0.02$, $\epsilon=0.1$) are used across all tasks (spherical helix, DNA, protein) without task-specific tuning. Similar stabilization techniques are also used in Euclidean few-step methods (Lu & Song, 2024; Geng et al., 2025), suggesting this is inherent to the few-step paradigm rather than specific to our method.
>
> **Q3. When is the cycle-consistency loss strictly necessary, and does removing it affect performance on protein or DNA tasks?**
>
> As shown in Figure A5, cycle-consistency is important for Lagrangian RMF on the high-dimensional spherical helix. However, we found that it is not necessary for the DNA task (nor for the Earth dataset in Appendix D). To clarify its effect, we report results with and without it (see [Table A15](https://imgur.com/X7Pjfg6)).
>
> We attribute this to the argmax discretization applied after generation, which absorbs minor invertibility errors in the continuous output. For the protein task, we were unable to isolate the effect of cycle-consistency, because Lagrangian RMF exhibits high training variance on SE(3) regardless of whether it is included, indicating that the primary challenge lies in the Lagrangian objective itself rather than the regularizer.
>
> **Q4. Inference-time controlled noise injection is critical for best designability and diversity. Does this suggest flow maps struggle to learn smooth trajectories in high dimensions?**
>
> The fact that noise injection improves inference-time performance is both well known and poorly understood in the field. However, it does not necessarily indicate a failure of the model in high dimensions. As shown in Figure A7(b), even without any inference-time heuristic, a sufficiently large model (RMF/L) already achieves strong designability at 5, 10, and 100 steps, demonstrating that our objective itself learns valid flow maps. Noise injection is a practical trick that enables smaller models (RMF/S, RMF/M) to achieve competitive performance with fewer parameters, rather than a requirement for valid generation.
>
> Importantly, this is not unique to flow map models — FrameFlow also degrades significantly without inference-time heuristics (scRMSD 3.03 at 100 NFE), suggesting that noise injection is a domain-level phenomenon in protein backbone generation, not a limitation specific to RMF.

---

> > ### Author Rebuttal · Reviewer_R1vx · 2026-04-03
> >
> > Thank you for the detailed rebuttal and the additional experiments. You have successfully addressed my initial concerns. I appreciate the effort put into this response, and I am raising my score to a Weak Accept.

---

> > > ### Author Response · Authors · 2026-04-06
> > >
> > > We sincerely thank the reviewer for the thorough evaluation and positive reassessment of our work. Your constructive feedback was instrumental in strengthening the paper, and we are grateful that our rebuttal and additional experiments successfully addressed the initial concerns.

---

### Official Review · Reviewer_68AG · 2026-03-11

**Soundness:** 4
**Presentation:** 2
**Significance:** 3
**Originality:** 4
**Overall Recommendation:** 4
**Confidence:** 3

**Summary:**

This paper proposes Riemannian MeanFlow (RMF), a framework for learning flow maps directly on manifolds rather than learning instantaneous vector fields and integrating them with many steps. The core idea is to define a manifold average velocity $u_{s, t}$ via the log map and recover the flow map through the exponential map, then train using one of three equivalent identities: Eulerian, Lagrangian, or semigroup. This paper also provides extensive theorectical derivation bridging the gap from Euclidean mean flow to Riemannian mean flow.

**Compliance With Llm Reviewing Policy:**

Affirmed.

**Final Justification:**

My concerns and questions have been resolved.

However, I would like to keep my score unchanged because (1) it is already on the positive side toward acceptance, (2) I gave my intial rating already based on the belief that the authors can address most of the concerns, and (3) to be honest, some of the mathematical aspects of the paper are beyond what I can confidently assess. I cannot confidently give a higher rating without fully understand the work.

**Key Questions For Authors:**

1. Personally I feel mean flow will lead to more "mode collapse" issue than regular flow matching. Mean flow generally experiences lower diversity from my experience. From Table 3, it seems that RMF has better diversity than regular flow matching models. What's your take on this?

2. Can the authors better quantify training cost versus baseline flow matching / diffusion models, including wall-clock and memory, not just inference NFE?

3. See weakness 3, can the authors provide a clarfication on the individual contribution of each components? Especially the contribution of  the new RMF identities themselves.

**Limitations:**

The authors do not explicitly discuss the limitations of this work. However, I am not sure whether there are any limitations that particularly need to be mentioned.

**Strengths And Weaknesses:**

# Strenths
1. The paper presents a natural manifold generalization of average-velocity-based flow-map learning. This is conceptually clean and mathematically well aligned with Riemannian geometry.
2. The derivation of three equivalent identities, Eulerian, Lagrangian, and semigroup, and the objective construction from these identities give the work a principled feel.
3. The results in Table 2 and Table 3 is strong. In the few-step regime, RMF appears substantially better than FrameDiff and FrameFlow, e.g. 0.82 designability at 5 steps versus 0.09/0.04 for the baselines, and still 0.35 at 1 step where baselines collapse. This is a meaningful practical result.

# Weaknesses
1. The practical contribution seems narrower than the theoretical framing. Although the paper presents three equivalent identities, the experiments strongly suggest that the semigroup objective is the only practically reliable one, while Eulerian and Lagrangian objectives appear mainly of conceptual interest.
2. Some theoretical claims rely on nontrivial local assumptions. The proofs of objective validity and flow recovery invoke local diffeomorphism / invertibility of the log map differential and, for the Lagrangian case, local invertibility / approximate inverse assumptions away from the cut locus. These assumptions are reasonable, but they weaken the force of the “complete characterization” message and deserve more prominent discussion in the main text.
3. The gains may come from a bundle of design choices rather than one isolated idea. RMF's success seems to depend on a combination of semigroup training, $x_1$-prediction, weighting near $s=1$, adaptive loss weighting, and time-embedding choices. It makes it harder to isolate how much of the improvement comes from the new RMF identities themselves.

---

> ### Author Rebuttal · Authors · 2026-03-31
>
> We thank the reviewer for the thorough evaluation and for recognizing the principled theoretical framework and the strong few-step generation results. We address each point below.
>
> **W1. Only the semigroup objective is practically reliable; Eulerian and Lagrangian appear mainly conceptual.**
>
> We want to clarify that our claim is not that the three objectives perform identically — rather, that Eulerian and Lagrangian RMF offer meaningful compute-efficiency gains while remaining competitive in sample quality. They are practical alternatives worth investigating, not merely conceptual. On both the DNA promoter and spherical helix tasks, Eulerian and Lagrangian RMF achieve comparable (and in some cases better) quality relative to the semigroup objective, with significantly less compute per iteration. See [Table A7, A8](https://imgur.com/yF0KqW0).
>
> In particular, Eulerian RMF with $x_1$-prediction achieves the best k-mer correlation (0.96) on the DNA task at 1 NFE (Table 1), matching the 100-step Fisher FM baseline. Nevertheless, we agree that for protein backbone generation, the semigroup objective is currently more reliable, as the differential terms in the Eulerian and Lagrangian objectives introduce additional variance in this high-dimensional setting. However, we note that the Eulerian formulation already underlies MeanFlow (Geng et al., 2025) and the Lagrangian formulation underlies TVM (Zhou et al., 2025), both scaling successfully to large-scale vision tasks. Our theoretical foundation ensures these objectives are well-defined on manifolds, ready for future advances in training stabilization.
>
> **W2-1. Some theoretical claims rely on nontrivial local assumptions.**
>
> Thank you for pointing this out. The key assumption in the Eulerian and Lagrangian proofs is that the log map differential is invertible, which holds excluding the cut locus. On compact Riemannian manifolds, the cut locus is a closed set of measure zero, so this assumption is violated only on a negligible subset of the manifold. For the specific manifolds considered in our experiments: the simplex with Fisher–Rao geometry and Euclidean space have globally invertible log maps (no cut locus), and for the SO(3) component of SE(3), the cut locus (rotations by $\pi$) has measure zero and does not affect training in practice.
>
> The Lagrangian proof additionally assumes flow map invertibility (Eq. 62). This is not a geometric assumption — the true flow map is a global diffeomorphism, so invertibility holds everywhere. The term "(local)" in the original proof conflates this with geometric locality; we will revise accordingly.
>
> **W2-2. These assumptions weaken the "complete characterization" message.**
>
> We agree that the scope should be stated more precisely. We will remove the "complete characterization" phrasing and instead state explicitly that our identities characterize the average velocity on its natural domain, where the log map differential is invertible, which excludes only a measure-zero set.
>
> **W3, Q3. Individual contribution of each component.**
>
> We appreciate this suggestion. The main paper already isolates identity choice (Figure 3) and parameterization (Figure 2). Here we further disentangle each stabilization technique by ablating from the full Eulerian + $x_1$-prediction configuration on DNA (see [Table A9](https://imgur.com/ExaumYQ)).
>
> Both variance-reduction techniques are essential: without frequency control, k-mer correlation collapses to −0.12; without adaptive loss weighting, MSE rises to 0.041. This is consistent with our analysis in Sec. 3.3 — differential objectives produce high-variance regression targets, and both techniques are needed to control this variance.
>
> **Q1. Why does RMF achieve better diversity than flow matching, given that mean flow methods typically have lower diversity?**
>
> This is an insightful observation; however, we would rather say that RMF's diversity is comparable to baseline methods. At 100 steps, RMF and FrameFlow show similar pairwise scTM (0.27 vs. 0.30), and at 10 steps, they are nearly identical (0.27 vs. 0.26). We view these differences as within the expected range.
>
> Regarding the concern about mode collapse in mean flow methods more generally, the average velocity field, if learned exactly, recovers the same flow map and target distribution as the underlying flow matching model. Mode collapse can arise in practice from approximation error, but it is not an inherent property of the formulation.
>
> **Q2. Training cost (wall-clock, memory) comparison with baselines?**
>
> We provide a full training cost comparison (see [Table A10](https://imgur.com/WfGAju0)). On the DNA task, RMF's training cost is nearly identical to Fisher FM. On the protein task, RMF requires approximately 2× longer wall-clock time; however, we believe this is a reasonable trade-off given the substantial gains in few-step inference efficiency (up to 10× fewer NFEs at test time).

---

> > ### Author Rebuttal · Reviewer_68AG · 2026-04-02
> >
> > Thank you for your response. My concerns and questions have been resolved.
> >
> > However, I would like to keep my score unchanged because (1) it is already on the positive side toward acceptance, and (2) to be honest, some of the mathematical aspects of the paper are beyond what I can confidently assess. I cannot confidently give a high rating without fully understand the work.

---

> > > ### Author Response · Authors · 2026-04-03
> > >
> > > We appreciate the reviewer's careful and constructive feedback, which helped improve the paper. We're glad our rebuttal resolved the concerns.

---

### Official Review · Reviewer_iR54 · 2026-03-13

**Soundness:** 4
**Presentation:** 4
**Significance:** 3
**Originality:** 2
**Overall Recommendation:** 4
**Confidence:** 3

**Summary:**

The paper proposes Riemannian Mean Flow (RMF), a generative modeling framework that learns flow maps directly on Riemannian manifolds to reduce the high inference cost of diffusion and flow models. Experiments on promoter DNA design and protein backbone generation show that RMF achieves comparable sample quality to prior diffusion/flow methods while requiring up to 10× fewer neural network evaluations.

**Compliance With Llm Reviewing Policy:**

Affirmed.

**Final Justification:**

While the authors address my concerns by providing additional information, the novelty of the work remains limited: it primarily extends existing mean flow architectures to the setting of Riemannian manifolds. Conceptually, it is analogous to prior extensions of flow matching methods to Riemannian manifolds. I lean toward acceptance; however, the limited novelty constrains the overall score.

**Key Questions For Authors:**

See weakness for questions.

**Limitations:**

Yes

**Strengths And Weaknesses:**

## Strengths

* The paper is well-written and provides sufficient background. Clear and standard derivations of theoretical results are provided.
* It introduces practical techniques for improving stability and training on manifolds, which might be useful for real-world applications.
* The experimental results are good, demonstrating significant reductions in function evaluations while maintaining competitive sample quality.


## Weaknesses

* The novelty is somewhat limited, as the method appears to be a relatively straightforward extension of prior flow map work by Boffi et al to the Riemannian manifold setting.
* While the extension is valuable and well-executed, the paper could better clarify the conceptual differences and new challenges introduced by the manifold setting compared to existing flow map approaches. Why existing flow map approaches may fail?

---

> ### Author Rebuttal · Authors · 2026-03-31
>
> We thank the reviewer for the careful evaluation and the positive assessment of our paper's clarity, theoretical derivations, and experimental results. We address the concern regarding novelty below.
>
> ---
> **W1 & W2. Limited novelty as a straightforward extension; clarify conceptual differences and why existing approaches may fail.**
>
> We thank the reviewer for this important question. While RMF shares the high-level goal of learning flow maps with Boffi et al. [1, 2], we respectfully argue that extending flow map learning to Riemannian manifolds is non-trivial, both mathematically and practically.
>
> **Mathematical challenges unique to the manifold setting.** On Riemannian manifolds, the average velocities and their derivatives used in Boffi et al. [1,2] need to be replaced by the logarithmic map, covariant derivatives, and differentials of the log map; these quantities introduce curvature-dependent terms that have no Euclidean counterpart. For example, the Eulerian identity (Prop 2.1) involves the term $\nabla_{v_{s}}^{1} \log_{x_{s}}x_{t}$ which reduces to $-v_s$ in the Euclidean setting. This term captures how the log map itself changes as one moves along $v_s$ on the manifold. Deriving the Eulerian, Lagrangian, and semigroup characterizations and proving that each is both necessary and sufficient for recovering the true average velocity (Appendix B.1) requires careful handling of these geometric quantities.
>
> **Average-velocity formulation enables efficient objective design.** Beyond extending the identities to manifolds, RMF differs from prior flow map methods in a practically important way. Prior works [1, 2] and their manifold extension GFM [3] define objectives at the flow-map level, requiring backpropagation through Jacobian-vector products (JVPs) involving geometric quantities, which becomes increasingly expensive and unstable in high dimensions. In contrast, our average-velocity formulation allows all differentiation-related terms to be placed entirely under a stop-gradient operator, bypassing JVP backpropagation by construction (see App. C.2 for the formal connection). This results in roughly half the memory (9.5 GB vs. 17.7 GB) and 2.6x faster training per iteration (Table A2).
>
> **Euclidean flow maps do not guarantee outputs on the manifold.** At a fundamental level, Euclidean flow maps produce outputs in $\mathbb{R}^d$ with no mechanism to enforce manifold constraints, leading to invalid samples — e.g., unnormalized probability vectors for DNA sequences or non-orthogonal rotation matrices for proteins. RMF resolves this by construction through the exponential map parameterization (Eq. 9), which maps tangent vectors to valid points on the manifold. This is not a minor implementation detail but a structural requirement: without it, the generative model cannot produce valid scientific objects.
>
> **Practical challenges of scaling to high dimensions**. Beyond the theoretical formulation, we empirically study multiple parameterizations and identify $x_1$-prediction as the most effective for high-dimensional manifolds (Sec. 4.1, Fig. 2). We also systematically investigate training stabilization techniques — adaptive loss weighting, time-embedding frequency control, and objective-specific time sampling strategies — that are essential for making flow map learning work at scale on manifolds. While some of these techniques have individually appeared in the Euclidean consistency model literature, they have not been employed in the flow map framework of Boffi et al., nor have they been systematically studied or consolidated in any prior work on manifold flow maps. Our ablation studies (Sec. 4.1 and App. G.1) demonstrate their critical role: without adaptive loss weighting, Eulerian RMF produces poor samples (Fig. 3); high-frequency time embeddings cause regression target explosion (Fig. A3); and semigroup and Lagrangian objectives require opposite time-sampling orderings (Fig. A2). We believe this systematic investigation and consolidation of practical guidelines constitutes a meaningful contribution to the community.
>
> **SOTA results on protein-backbone generation.** We furthermore highlight that achieving SOTA results in terms of speed and quality of generation brings its own merits to the field of drug design.
>
> [1] Boffi et al., "Flow map matching with stochastic interpolants."
>
> [2] Boffi et al., "How to build a consistency model."
>
> [3] Davis et al., "Generalised flow maps for few-step generative modelling on Riemannian manifolds."

---

> > ### Author Rebuttal · Reviewer_iR54 · 2026-03-31
> >
> > Thank you for the authors’ response. I will keep my score, which is already leaning toward acceptance.

---

> > > ### Author Response · Authors · 2026-04-03
> > >
> > > We're grateful that our rebuttal helped clarify your concerns. Thank you for your constructive feedback and continued support.

---

### Decision · Program_Chairs · 2026-04-30

**Decision:**

Accept (regular)

**Comment:**

The reviewers found this to be a technically strong and well-executed paper, with a mathematically principled extension of mean-flow ideas to Riemannian manifolds and clear derivations of the Eulerian, Lagrangian, and semigroup formulations. They also viewed the empirical results favorably, especially in the few-step regime, where RMF achieves competitive or strong sample quality while substantially reducing function evaluations on DNA and protein design tasks. Several reviewers further highlighted the paper’s practical value, including its useful analysis of parameterization and stabilization choices and its relevance for efficient scientific generation. The main weaknesses were concerns about limited novelty relative to prior flow-map methods, along with some questions about the scope of theoretical claims and the completeness and clarity of parts of the experimental evaluation. Overall, the reviewers converged on a positive assessment and viewed the paper as a solid contribution that merits acceptance despite these reservations.